# Asynchronous Decentralized Online Learning

**Jiyan Jiang**[*]
Tsinghua University
scjjy95@outlook.com

**Wenpeng Zhang**[*]
Ant Group
zhangwenpeng0@gmail.com

**Jinjie Gu**
Ant Group
jinjie.gujj@antgroup.com

**Wenwu Zhu**[†]
Tsinghua University
wwzhu@tsinghua.edu.cn

## Abstract

Most existing algorithms in decentralized online learning are conducted in the synchronous setting. However, synchronization makes these algorithms suffer from the straggler problem, i.e., fast learners have to wait for slow learners, which significantly reduces such algorithms' overall efficiency. To overcome this problem, we study decentralized online learning in the asynchronous setting, which allows different learners to work at their own pace. We first formulate the framework of Asynchronous Decentralized Online Convex Optimization, which specifies the whole process of asynchronous decentralized online learning using a sophisticated event indexing system. Then we propose the Asynchronous Decentralized Online Gradient-Push (AD-OGP) algorithm, which performs asymmetric gossiping communication and instantaneous model averaging. We further derive a regret bound of AD-OGP, which is a function of the network topology, the levels of processing delays, and the levels of communication delays. Extensive experiments show that AD-OGP runs significantly faster than its synchronous counterpart and also verify the theoretical results.

## 1 Introduction

Decentralized online learning has recently received increasing research attention due to its capability of processing large-scale streaming data in decentralized computational architectures (e.g., mobile networks and CPU/GPU clusters). Many different algorithms have been proposed [31, 32, 22, 15, 28, 16, 24] and successfully applied in various application scenarios, such as recommendation systems [29], energy management [5] and cooperative navigation [21].

Most of the existing decentralized online algorithms are investigated under the synchronous setting, where all the learners are restricted to communicating their updated information (e.g., model parameters, gradients, dual variables) with each other at the same pace. However, this synchronization mechanism incurs the straggler problem, which can markedly deteriorate the efficiency of the algorithms [30, 17, 3]. In more detail, at each iteration, the fast learners have to wait for the slow learners, i.e., the stragglers, to finish their update and communication. Consequently, their speeds will be dragged down by those of the stragglers, which may dramatically reduce the algorithms' overall efficiency as the speeds of different learners in a decentralized system usually vary widely owing to varying computing powers, storage capacities, network bandwidths, etc.

As is well acknowledged, designing the asynchronous counterpart of a synchronous algorithm is the most common way to resolve the straggler problem. However, compared to the well-studied

---

[*]Equal contributions.    [†]Corresponding author.

centralized online [9, 18] and decentralized stochastic [17, 12, 3, 23] settings, how to do asynchronous learning in the decentralized online setting remains largely unexplored. The major obstacle is its intrinsic hardness. In asynchronous decentralized online learning, each learner has five basic types of actions, namely prediction, weight update, message sending, message receiving, and model averaging. Due to asynchronization, each learner executes its own actions independently. Hence the orders of the actions of different learners are intermixed together, which results in very complex interaction dynamics. By far, even no formal framework exists that can accurately describe the whole process of asynchronous decentralized online learning, not to mention algorithm design or theoretical analysis.

In this paper, we conduct a systematic study of asynchronous decentralized online learning. To begin with, we formulate the framework of Asynchronous Decentralized Online Convex Optimization (AD-OCO). We first specify the action sequence of each learner using its own indices, then propose a novel event indexing system, which delicately characterizes the complete orders of actions of all learners by mapping them into a single time axis. To the best of our knowledge, this is the first formal framework that gives a complete characterization of the whole interaction dynamics of asynchronous decentralized online learning.

Based on the proposed AD-OCO framework, we develop the Asynchronous Decentralized Online Gradient-Push (AD-OGP) algorithm. AD-OGP consists of two key modules. The first module is a gossip-based communication scheme, in which each learner is only allowed to communicate with its immediate neighbors. In general, there are two major types of gossiping strategies in decentralized algorithms, namely symmetric gossiping [13, 17] and asymmetric gossiping [19, 3]. In *asymmetric gossiping*, the gossiping sender does not need to wait for the response message from the gossiping receiver after it sends out its message. Hence asymmetric gossiping does not require any local synchronization among learners, which is needed in symmetric gossiping. This makes asymmetric gossiping more suitable for asynchronous online learning. In particular, we adopt the push-sum strategy [14, 19], which is the most common way to realize asymmetric gossiping. The second module is a local update scheme, which builds upon online gradient descent (OGD) [33]. When combining push-sum with OGD, if we directly apply the standard projection operation, the resulting algorithm crashes as it may generate predictions outside of the feasible region. To resolve this problem, we devise a novel *weighted projection* operation, which makes its local update compatible with push-sum. Moreover, in AD-OGP, we allow each learner to perform model averaging as long as it is unoccupied and its receiving buffer is not empty, which effectively reduces the communication delays. We emphasize that the above innovations, namely weighted projection and *instantaneous model averaging*, are two special designs when tailoring asynchronization to the online setting.

We further conduct the first regret analysis of asynchronous decentralized online learning. In particular, we extend the graph augmentation technique [10] to the online setting to handle message delays, then disentangle the complex effects of predictions, weight update, and model averaging via our proposed event indexing system. We derive a non-trivial regret bound of AD-OGP, which captures the effects of the network structures, the levels of processing delays, and the levels of communication delays. Our bound is in the same order $O(\sqrt{T})$ as that of its synchronous counterpart [27]. In addition, we make an extra contribution to the convergence analysis of push-sum. Specifically, we reduce a factor of $\sqrt{n}$ ($n$ is the dimension of model parameters) in the convergence rate of push-sum compared to that in [2], and our final bound is now independent of $n$.

Experimental results on two large-scale real-world datasets show that AD-OGP runs substantially faster than its synchronous counterpart, with nearly negligible loss in performance. We further design elaborate experiments to verify the effectiveness of the two innovations of asymmetric gossiping and instantaneous model averaging in AD-OGP. Specifically, with either innovation removed or weakened, the reduced algorithm will be ineffective in some specific settings. The theoretical results of AD-OGP's regret bound on different graphs and delay levels are also well confirmed.

## 2 Asynchronous Decentralized Online Convex Optimization

In this section, we formally describe the AD-OCO framework. Our framework is defined over an undirected graph $\mathcal{G} = (V, E)$ with vertex set $V = \{1, \dots, m\}$ and edge set $E \subset V \times V$. Each node $i \in V$ represents an individual learner which maintains a local model $\boldsymbol{w}_i \in \mathcal{K}$, where $\mathcal{K}$ is a convex compact decision set. Each learner $i$ can only communicate with its immediate neighbors $\mathcal{N}(i) = \{j \in V \mid (i, j) \in E\}$.

We first illustrate the whole process of AD-OCO. As the process is very complex, we divide its description into two parts: (i) *Individual view*, which specifies the action sequence of each individual learner; (ii) *System view*, which characterizes the interaction dynamics among different learners.

**Individual View**   Each learner $i \in V$ is selected adversarially by the environment to make predictions at certain time points. We index the prediction rounds of learner $i$ as $\tau = 1, \ldots, N_i$. At each round $\tau$, learner $i$ *predicts* with its most recent model $\boldsymbol{w}_i^\tau$. Then the environment reveals a convex loss function $f_i^\tau : \mathcal{K} \to \mathbb{R}$, and learner $i$ suffers the loss $f_i^\tau(\boldsymbol{w}_i^\tau)$. When receiving the feedback, learner $i$ begins to calculate the gradient $\nabla f_i^\tau(\boldsymbol{w}_i^\tau)$. After finishing the calculation, it uses the gradient to *update* its model $\boldsymbol{w}_i \leftarrow \mathcal{U}(\boldsymbol{w}_i; \nabla f_i^\tau(\boldsymbol{w}_i^\tau))$, where $\mathcal{U}$ is the local update scheme. Then it selects a subset of neighbors, and *sends* a copy of its updated model to each of them. Besides sending out its own model copies, learner $i$ will also *receive* its neighbors' model copies at certain time points. These copies will be stored in its receiving buffer $\mathcal{B}_i$. Whenever learner $i$ is not occupied with gradient calculation or local update, and its buffer is not empty, it can *average* its model with the model copies stored in the buffer, i.e., $\boldsymbol{w}_i \leftarrow \mathcal{A}(\boldsymbol{w}_i; \mathcal{B}_i)$, where $\mathcal{A}$ is the model averaging scheme.

**System View**   Due to asynchronization, each learner executes its actions independently. Hence the orders of actions of all learners are intermixed. To precisely characterize the interaction dynamics among learners, we need to specify the complete orders of the actions of all learners by mapping them into a single time axis. To this end, we propose a novel event indexing system. We view either a prediction or a local update of any learner as an *event*, and use a virtual counter $t$ to index these events, which will increase by one each time some learner makes a prediction or completes a local update. Let $T = \sum_{i \in V} N_i$ be the total number of predictions of all learners, then the total number of local updates is also $T$. For each event $t \in \{1, \ldots, 2T\}$, we use $\delta_t \in \{0, 1\}$ to distinguish the event type: (i) $\delta_t = 0$ if it is a prediction event, and (ii) $\delta_t = 1$ if it is a local update event. We further use $i_t$ to denote the learner that executes the event $t$, and $f_t$ to denote the associated feedback; then there exists some $\tau \in \{1, \ldots, N_{i_t}\}$ such that $f_t = f_{i_t}^\tau$.

In particular, if event $t$ is a local update event (i.e., $\delta_t = 1$), we denote $l_t$ as the index of the associated prediction event[1], and $d^g(t) = t - l_t$ as the gap between these two indices. Since any prediction must occur before its corresponding local update, here we have $t \geq 2$, $l_t < t$ and $d^g(t) > 0$. Let $\mathrm{card}(A)$ denote the cardinality of set $A$. Now we can introduce a quantity of $f_t$ called the **processing delay** $d^p(t) = \mathrm{card}\{l_t < s < t \mid \delta_s = 1\}$, which measures the number of local updates that occur between the prediction and the local update associated with $f_t$. Note that our definition of processing delay is in line with those in other asynchronous frameworks for stochastic optimization [17, 3].

Recall that, after any local update event $t$, the learner $i_t$ that executes this event will send out its model copy to a subset $\mathcal{S}_t$ of its neighbors $\mathcal{N}(i_t)$. We focus on the copy sent to any neighbor $j \in \mathcal{S}_t$. We assume such copy is used to average learner $j$'s model between events $r_{tj}$ and $(r_{tj} + 1)$ for some $r_{tj} \geq t$. For the message sent from learner $i_t$ to learner $j$ after event $t$, we introduce a quantity termed the **message delay** $d_{i_t j}^m(t) = r_{tj} - t$, which measures the number of events that occur while this copy is in transmission or learner $j$'s receiving buffer. Note that in the asynchronous stochastic setting [3, 23], the message delay is measured by solely counting the number of local updates. In the online setting, we also count the predictions because whether model averaging occurs before or after a prediction will affect the gradient calculation, which further affects the corresponding weight update.

With the above event indexing system, we can now characterize the evolution of model $\boldsymbol{w}_i$. For each learner $i$, we denote $\boldsymbol{w}_i(t)$ as its most recent model before event $t$, and $\boldsymbol{u}_i(t)$ as its model immediately after event $t$. We first give the detailed characterization of local updates. Specifically, for each local update event $t$, it is executed by learner $i_t$. The associated gradient was computed w.r.t. learner $i_t$'s model $\boldsymbol{w}_{i_t}(l_t)$ at prediction event $l_t$. So the local update on $i_t$ can be characterized as

$$\boldsymbol{u}_{i_t}(t) = \mathcal{U}(\boldsymbol{w}_{i_t}(t); \nabla f_{l_t}(\boldsymbol{w}_{i_t}(l_t))).$$

The models of other learners do not change at $i_t$'s local update event, so $\boldsymbol{u}_j(t) = \boldsymbol{w}_j(t)$ for $j \neq i_t$.

We then give a full description of model averaging. Recall that, in our indexing system, model averaging operation is not regarded as an event. Hence we need to specify it using the indices of its nearby events. Concretely, for each learner $i$, we consider its model averaging during the interval

---

[1]Note that in our indexing system, when $\delta_t = 1$, the feedback and its associated learner can be represented by the index of either the prediction event $l_t$ or the local update event $t$, i.e., $f_{l_t} = f_t$ and $i_{l_t} = i_t$.

---

**Protocol 1** Asynchronous Decentralized Online Convex Optimization (**AD-OCO**)

1: **Input:** Time horizon $T$, convex set $\mathcal{K}$, local update scheme $\mathcal{U}$, and model averaging scheme $\mathcal{A}$.
2: **Initialize:** Local models $\boldsymbol{w}_j(1) \in \mathcal{K}, \forall j \in V$.
3: **for** $t = 1, \ldots, 2T$ **do**
4:     **if** $\delta_t = 0$ **then**                                                 *// for a prediction event*
5:         A learner $i_t \in V$ is selected by the environment adversarially to predict with $\boldsymbol{w}_{i_t}(t)$.
6:         Learner $i_t$ receives a convex function $f_t$ from environment, and suffers the loss $f_t(\boldsymbol{w}_{i_t}(t))$.
7:         Learner $i_t$ starts to compute the gradient $\nabla f_t(\boldsymbol{w}_{i_t}(t))$.
8:     **else**                                                         *// for a local update event*
9:         Learner $i_t$ performs a local update:

$$\boldsymbol{u}_{i_t}(t) = \mathcal{U}(\boldsymbol{w}_{i_t}(t); \nabla f_{l_t}(\boldsymbol{w}_{i_t}(l_t))).$$

10:         Learner $i_t$ selects a subset $\mathcal{S}_t$ of its neighbors $\mathcal{N}(i_t)$ and sends $\boldsymbol{u}_{i_t}(t)$ to each $j \in \mathcal{S}_t$.
11:     **end if**
12:     **for** $i \in V$ **do**
13:         Learner $i$ performs model averaging using model copies $\mathcal{B}_i^1(t), \ldots, \mathcal{B}_i^s(t)$ if it is *unoccupied*:

$$\boldsymbol{w}_i(t+1) = \begin{cases} \boldsymbol{u}_i(t), & s = 0; \\ \mathcal{A}(\cdots(\mathcal{A}(\mathcal{A}(\boldsymbol{u}_i(t), \mathcal{B}_i^1(t)), \mathcal{B}_i^2(t))\cdots), \mathcal{B}_i^s(t)), & s \geq 1. \end{cases}$$

14:     **end for**
15: **end for**

---

between event $t$ and $t+1$. If it is *not* occupied in this interval, we assume it performs $s$ model averaging operations for some $s \geq 0$, each time using copies in $\mathcal{B}_i^1(t), \ldots, \mathcal{B}_i^s(t)$. Recall that learner $i$'s models at the beginning and the end of such interval are $\boldsymbol{u}_i(t)$ and $\boldsymbol{w}_i(t+1)$ respectively. Hence the overall effect of model averaging on learner $i$ between event $t$ and $t+1$ can be expressed as

$$\boldsymbol{w}_i(t+1) = \begin{cases} \boldsymbol{u}_i(t), & s = 0; \\ \mathcal{A}(\cdots(\mathcal{A}(\mathcal{A}(\boldsymbol{u}_i(t), \mathcal{B}_i^1(t)), \mathcal{B}_i^2(t))\cdots), \mathcal{B}_i^s(t)), & s \geq 1. \end{cases}$$

**Regret Definition**   In AD-OCO, the regret w.r.t. an arbitrary reference learner $j \in V$ is defined as

$$\mathbf{Regret}_j = \sum_{t \in \mathcal{P}_T} f_t(\boldsymbol{w}_j(t)) - \sum_{t \in \mathcal{P}_T} f_t(\boldsymbol{w}^*),$$

where $\mathcal{P}_T = \{t \in \{1, \ldots, 2T\} \mid \delta_t = 0\}$ denotes the set of all prediction events, and $\boldsymbol{w}^* \in \operatorname{argmin}_{\boldsymbol{w} \in \mathcal{K}} \sum_{t \in \mathcal{P}_T} f_t(\boldsymbol{w})$ denotes the best fixed model in hindsight. The goal of the learners is to make a sequence of predictions $\{\boldsymbol{w}_i(t)\}_{i \in V, t \in \mathcal{P}_T}$, so that the regret w.r.t. any reference learner $j$ is sublinear in $T$, i.e., $\lim_{T \to \infty} \mathbf{Regret}_j / T = 0$. Note that our regret is in line with the regret defined in the synchronous setting [27, 31], where all the losses $\{f_t \mid t \in \mathcal{P}_T\}$ are evaluated by a fixed reference learner $j$.

## 3   Asynchronous Decentralized Online Gradient-Push

In this section, we first present the AD-OGP algorithm, then provide theoretical analysis for it.

### 3.1   The Algorithm

AD-OGP consists of two core modules: a local update scheme and a communication scheme. The local update scheme builds upon OGD [33]. The communication scheme is based on push-sum [19]. We introduce two novel algorithmic designs when combining push-sum with OGD, namely weighted projection and instantaneous model averaging. The complete pseudo-code of AD-OGP is given in Algorithm 1.

**Push-Sum Communication**   Before presenting the details of push-sum [14], we first give some background knowledge to help understand the rationale of it. In most decentralized algorithms,

---

**Algorithm 1** Asynchronous Decentralized Online Gradient-Push (**AD-OGP**)

---

1: **Input:** Time horizon $T$, convex set $\mathcal{K}$, and learning rate $\eta$.
2: **Initialize:** Push-sum model $\boldsymbol{x}_j(1) \leftarrow \boldsymbol{w}_0 \in \mathcal{K}$ and push-sum weight $y_j(1) \leftarrow 1, \forall j \in V$.
3: **for** $t = 1, \ldots, 2T$ **do**
4:     **if** $\delta_t = 0$ **then**                                    *// for a prediction event*
5:        Learner $i_t$ predicts with $\boldsymbol{w}_{i_t}(t) = \boldsymbol{x}_{i_t}(t)/y_{i_t}(t)$.
6:        Learner $i_t$ starts to calculate the gradient $\nabla f_t(\boldsymbol{w}_{i_t}(t))$.
7:     **else**                                                   *// for a local update event*
8:        Learner $i_t$ performs weighted projected online gradient descent:

$$\left(\boldsymbol{x}'_{i_t}(t), \ y'_{i_t}(t)\right) = \left(\gamma_{i_t} y_{i_t}(t)\Pi_{\mathcal{K}}\left(\frac{\boldsymbol{x}_{i_t}(t) - \eta \nabla f_{l_t}(\boldsymbol{w}_{i_t}(l_t))}{y_{i_t}(t)}\right), \ \gamma_{i_t} y_{i_t}(t)\right).$$

9:        Learner $i_t$ sends $\boldsymbol{x}'_{i_t}(t)$ and $y'_{i_t}(t)$ to each of its neighbors $j \in \mathcal{N}(i_t)$.
10:    **end if**
11:    **for** $i \in V$ **do**
12:       Learner $i$ performs model averaging using model copies $\mathcal{B}_i^1(t), \ldots, \mathcal{B}_i^s(t)$ if it is *unoccupied*:

$$\left(\boldsymbol{x}_i(t+1), \ y_i(t+1)\right) = \left(\boldsymbol{x}'_i(t) + \sum_{r=1}^{s} \sum_{(\boldsymbol{x}', y') \in \mathcal{B}_i^r(t)} \boldsymbol{x}', \ y'_i(t) + \sum_{r=1}^{s} \sum_{(\boldsymbol{x}', y') \in \mathcal{B}_i^r(t)} y'\right).$$

13:    **end for**
14: **end for**

---

communication is based on distributed averaging [26, 8], i.e., computing the average model $\bar{\boldsymbol{x}} = \frac{1}{m} \sum_{i \in V} \boldsymbol{x}_i$ of all learners $V = \{1, \ldots, m\}$ over a network $\mathcal{G}$. In general gossip-based approaches, at step $l \in \{1, \ldots, L\}$, each learner $i \in V$ accesses the models of its neighbors and averages these models with its own model to obtain $\boldsymbol{x}_i(l+1) = \sum_{j \in V} p_{ij}(l)\boldsymbol{x}_j(l)$, where $p_{ij}(l) \geq 0$ is the averaging weight. In particular, $p_{ij}(l) > 0$ only if learner $i$ accesses learner $j$'s model. The model averaging of all learners has a compact matrix form $\boldsymbol{X}(l+1) = \boldsymbol{P}(l)\boldsymbol{X}(l)$, where the $i$-th row of $\boldsymbol{X}(l)$ denotes learner $i$'s model $\boldsymbol{x}_i(l)$, and the entry at the $i$-th row and $j$-th column of $\boldsymbol{P}(l)$ is $p_{ij}(l)$.

In symmetric gossiping [8, 17], it is a common practice to design communication schemes such that the corresponding $\boldsymbol{P}(l)$ is doubly stochastic, i.e., $\boldsymbol{P}(l)\mathbf{1} = \boldsymbol{P}(l)^{\top}\mathbf{1} = \mathbf{1}$, where $\mathbf{1}$ is the $m \times 1$ all-one vector. Then by the property of Markov chain, $\lim_{L \to \infty} \prod_{l=1}^{L} \boldsymbol{P}(l) = \frac{1}{m}\mathbf{1}\mathbf{1}^{\top}$, which implies that the local model of each learner $i$ will approach $\boldsymbol{x}_i(\infty) = \bar{\boldsymbol{x}}$. The doubly stochastic property is often achieved by letting $p_{ij}(t) = p_{ji}(t)$, which requires local synchronization between the gossiping pair $(i, j)$. In asymmetric gossiping, however, we typically have $p_{ij}(l) \neq p_{ji}(l)$. So in general, it is very hard or even impossible to design a scheme in which $\boldsymbol{P}(l)$ is doubly stochastic [2]. The usual way [20, 2] to bypass such harsh requirements is to design a communication scheme whose $\boldsymbol{P}(l)$ only satisfies the column stochasticity, i.e., $\boldsymbol{P}(l)^{\top}\mathbf{1} = \mathbf{1}$. Then it can be proved that $\lim_{L \to \infty} \prod_{l=1}^{L} \boldsymbol{P}(l) = \boldsymbol{\pi}\mathbf{1}^{\top}$ for some $\boldsymbol{\pi} \in \mathbb{R}_+^m$, which implies that each learner $i$ will converge to $\pi_i \bar{\boldsymbol{x}}$. Push-sum strategy [19] removes the bias $\pi_i$ via very light extra communication. Specifically, each learner $i$ maintains a positive scalar $y_i$ termed the push-sum weight, initialized as $y_i(1) = 1$. Then it applies the same averaging operation of $\boldsymbol{x}_i$ to $y_i$ at each step $l$, i.e., $\boldsymbol{y}(l+1) = \boldsymbol{P}(l)\boldsymbol{y}(l)$. By the property of Markov chain, $\lim_{l \to \infty} y_i(l) = \pi_i$ for each learner $i$. Thus the ratio $\boldsymbol{x}_i(l)/y_i(l)$ approximately approaches $\bar{\boldsymbol{x}}$.

Given the above background knowledge, we now provide the details to implement push-sum in the asynchronous online setting. The core is to assign the averaging weights $p_{ij}(t)$ at each step $t$ to make $\boldsymbol{P}(t)$ column stochastic. For simplicity, we first assume there is no message delay. After each local update event $t$, learner $i_t$ can only send its model copy to its neighbors $\mathcal{N}(i_t)$. Hence $p_{ki_t}(t) > 0$ only if $k \in \mathcal{N}(i_t) \cup \{i_t\}$. To ensure $\sum_{k \in V} p_{ki_t}(t) = 1$, a natural choice is to assign the *same* weight $p_{ki_t}(t) = \gamma_{i_t}$ for all $k \in \mathcal{N}(i_t) \cup \{i_t\}$, where $\gamma_{i_t} = 1/(|\mathcal{N}(i_t)| + 1)$. Our implementation satisfies this condition: After each local update $t$, learner $i_t$ first multiplies its model $\boldsymbol{x}_{i_t}$ and push-sum weight $y_{i_t}$ by $\gamma_{i_t}$, i.e., $\boldsymbol{x}'_{i_t} = \gamma_{i_t}\boldsymbol{x}_{i_t}$ and $y'_{i_t} = \gamma_{i_t}y_{i_t}$. Then it sends a copy of $\boldsymbol{x}'_{i_t}$ and $y'_{i_t}$ to each neighbor $j \in \mathcal{N}(i_t)$. In the corresponding model averaging, learner $j$ adds $\boldsymbol{x}'_{i_t}$ and $y'_{i_t}$ to its own parameters,

i.e., $\boldsymbol{x}_j \leftarrow \boldsymbol{x}_j + \boldsymbol{x}'_{i_t} = \boldsymbol{x}_j + \gamma_{i_t}\boldsymbol{x}_{i_t}$ and $y_j \leftarrow y_j + y'_{i_t} = y_j + \gamma_{i_t}y_{i_t}$. The model averaging operation of learner $i$ using buffer $\mathcal{B}_i^s$ can be summarized as

$$\mathcal{A}((\boldsymbol{x}_i, y_i); \mathcal{B}_i^s) = \Big(\boldsymbol{x}_i + \sum_{(\gamma_j\boldsymbol{x}_j, \gamma_j y_j) \in \mathcal{B}_i^s} \gamma_j\boldsymbol{x}_j, \ y_i + \sum_{(\gamma_j\boldsymbol{x}_j, \gamma_j y_j) \in \mathcal{B}_i^s} \gamma_j y_j\Big).$$

Note that when there are message delays, our scheme can still induce column stochastic $\boldsymbol{P}(t)$ on the augmented graph [10] at each step $t \in \{1, \ldots, 2T\}$ (see detailed proofs in supplementary materials).

**Weighted Projected Online Gradient Descent**  Recall that, in push-sum, each learner $i \in V$ has two types of parameters, namely the push-sum model $\boldsymbol{x}_i$ and the push-sum weight $y_i$; and it always predicts with $\boldsymbol{w}_i = \boldsymbol{x}_i/y_i$. Learner $i$ is initialized by $\boldsymbol{x}_i \leftarrow \boldsymbol{w}_0 \in \mathcal{K}$ and $y_i \leftarrow 1$. In OGD's update with gradient $\boldsymbol{g}$, learner $i$ first performs a gradient step on its model $\boldsymbol{x}_i$, i.e., $\boldsymbol{x}_i \leftarrow \boldsymbol{x}_i - \eta\boldsymbol{g}$, where $\eta$ is the learning rate. However, its next prediction $\boldsymbol{x}_i/y_i$ may lie outside of the decision set $\mathcal{K}$, which violates the feasibility. To ensure feasibility, OGD executes a projection operation $\Pi_\mathcal{K}$ on the model $\boldsymbol{x}_i$ after the gradient step, i.e., $\Pi_\mathcal{K}(\boldsymbol{x}_i) = \mathrm{argmin}_{\boldsymbol{x}\in\mathcal{K}} \|\boldsymbol{x} - \boldsymbol{x}_i\|$, where $\|\cdot\|$ denotes the L2 norm. However, if we directly apply the standard projection operation, we may still have $\boldsymbol{w}_i = \Pi_\mathcal{K}(\boldsymbol{x}_i)/y_i \notin \mathcal{K}$ after projection, because $y_i$ can be less than 1. Hence we need a special design of the projection operation. Notice that when $y_i > 0$, $\boldsymbol{x}_i/y_i \in \mathcal{K}$ is equivalent to $\boldsymbol{x}_i \in y_i\mathcal{K} = \{y_i\boldsymbol{w} \mid \boldsymbol{w} \in \mathcal{K}\}$. Since $y_i\mathcal{K}$ is also convex, we can project $\boldsymbol{x}_i$ onto $y_i\mathcal{K}$ instead of $\mathcal{K}$. Then we have $\Pi_{y_i\mathcal{K}}(\boldsymbol{x}_i) = \mathrm{argmin}_{\boldsymbol{x}\in y_i\mathcal{K}} \|\boldsymbol{x} - \boldsymbol{x}_i\| = y_i\,\mathrm{argmin}_{\boldsymbol{w}\in\mathcal{K}} \|\boldsymbol{w} - \boldsymbol{x}_i/y_i\| = y_i\Pi_\mathcal{K}(\boldsymbol{x}_i/y_i)$, which can be viewed as the standard projection $\Pi_\mathcal{K}(\boldsymbol{x}_i)$ distorted by weight $y_i$. The above local update procedure can be summarized as $\boldsymbol{x}_i \leftarrow y_i\Pi_\mathcal{K}((\boldsymbol{x}_i - \eta\boldsymbol{g})/y_i)$ and $y_i \leftarrow y_i$. The feasibility of the iterations of AD-OGP is strictly proved in supplementary materials.

Recall that, in our implementation of push-sum, we multiply learner $i$'s parameters $\boldsymbol{x}_i$ and $y_i$ by a factor of $\gamma_i$ *immediately* after its local update. This multiplication operation can be combined with the above local update form, which gives a compact expression

$$\mathcal{U}((\boldsymbol{x}_i, y_i); \boldsymbol{g}) = (\gamma_i y_i\Pi_\mathcal{K}((\boldsymbol{x}_i - \eta\boldsymbol{g})/y_i), \gamma_i y_i).$$

**Instantaneous Model Averaging**  In the asynchronous online setting, another design issue, namely *when* to perform model averaging for each learner, should be considered carefully. The learner is unoccupied only when it completes a local update and waits to be selected by the environment to make the next prediction. Hence the learner can only perform model averaging during this period. However, the learner may receive messages at any time. Each message will be stored in the receiving buffer until the learner performs the next model averaging. The waiting time of this message also contributes to message delays. To effectively avoid increasing message delays, we allow each learner to perform model averaging as soon as possible. (i) If the learner receives any message during gradient calculation or local update, it will use this message for model averaging as soon as it completes the local update. (ii) If the learner receives any message during its unoccupied period, it will perform model averaging immediately.

We remark that instantaneous model averaging is a novel design specific to the online setting since when to perform model averaging does not even constitute a problem in stochastic optimization. In fact, in asynchronous decentralized stochastic optimization [3, 23], when to calculate the gradient and perform the local update is actively decided the learner itself, as the data instances do not come in a streaming manner. Particularly, the learner can begin to process the next instance immediately after it completes any local update. Hence the learner is always occupied, and the time available for model averaging is only when it finishes a local update. In the online setting, however, the time points of making predictions are adversarially decided by the environment, and thus there may be some time lag between any local update and its subsequent prediction, during which the learner stays unoccupied. Therefore, when to perform model averaging during such unoccupied period needs to be decided. In AD-OGP, we design instantaneous model averaging, which allows each learner to perform model averaging immediately when it enters the unoccupied period.

## 3.2  Theoretical Analysis

In theoretical analysis, we only need to consider the case of $m \geq 2$. In fact, when $m = 1$, there is only one single learner in the network and hence there is no communication. In this case, the push-sum weight of the only learner is always $y_1(t) = 1$ and its local model simply evolves as

$\boldsymbol{x}_1(t+1) = \Pi_{\mathcal{K}}(\boldsymbol{x}_1(t) - \eta\nabla f_t(\boldsymbol{x}_1(t)))$, which reduces to the standard OGD [33]. Therefore, in the following analysis, we assume $m \geq 2$. Now we introduce two mild assumptions that are commonly used in the analysis of asynchronous distributed algorithms [3, 23].

**Assumption 1.** *(a) The message delays are bounded by some integer $D^{msg} \geq 0$.*
*(b) Each learner performs local updates at least once every $\Gamma_d$ steps for some integer $\Gamma_d > 0$.*

Before presenting the regret bound, we first give some necessary notations. We use $\mathcal{Q}_T = \{t \leq 2T \mid \delta_t = 1\}$ to denote the indices of all local update events, then equivalently $\mathcal{Q}_T = \{1, \ldots, 2T\} - \mathcal{P}_T$. We further use $\mathcal{Q}_{s,t} = \mathcal{Q}_T \cap (s, t)$ to denote the local updates that occurs between events $s$ and $t$. Now for any local update event $t \in \mathcal{Q}_T$, the processing delay $d^p(t)$ is exactly $|\mathcal{Q}_{l_t,t}|$. Moreover, denote the gradients associated with each local update event $t$ as $\boldsymbol{g}_t = \nabla f_{l_t}(\boldsymbol{x}_{i_t}(l_t)/y_{i_t}(l_t)), \hat{\boldsymbol{g}}_t = \nabla f_{l_t}(\boldsymbol{x}_j(l_t)/y_j(l_t))$, their $L2$-norms as $g_t = \|\boldsymbol{g}_t\|, \hat{g}_t = \|\hat{\boldsymbol{g}}_t\|$, the total processing delay of all feedbacks as $D^{proc} = \sum_{t \in \mathcal{Q}_T} d^p(t)$, and the diameter of the underlying network $\mathcal{G}$ as $\mathcal{D}$.

**Theorem 1.** *For any reference learner $j \in V$, AD-OGP attains the following regret bound*

$$\textbf{Regret}_j \leq \frac{m}{2\eta}\|\boldsymbol{w}_0 - \boldsymbol{w}^*\|^2 + \frac{2\eta}{m}\sum_{t \in \mathcal{Q}_T}\sum_{s \in \mathcal{Q}_{l_t,t+1}}g_t g_s + \eta\sum_{t \in \mathcal{Q}_T}\frac{g_t^2}{y_{i_t}(t)}$$

$$+ 2\eta\sum_{t \in \mathcal{Q}_T}\sum_{s \in \mathcal{Q}_{1,l_t}}\lambda^{\lfloor\frac{l_t-s}{2B}\rfloor}(g_t + 2\hat{g}_t)\frac{g_s}{y_{i_s}(t)} + 2\eta\sum_{t \in \mathcal{Q}_T}\sum_{s \in \mathcal{Q}_{1,t+1}}\lambda^{\lfloor\frac{t-s}{2B}\rfloor}g_t\frac{g_s}{y_{i_s}(t)},$$

*where $\lambda = 1 - m\alpha^4, \alpha = (1/m)^B, B = (\mathcal{D}+1)(D^{msg} + \Gamma_d)$.*

***Proof sketch*** Our proof consists of three major steps. *First*, we extend the graph augmentation technique [10] to the online setting to handle message delays. Specifically, we add $D^{msg}$ virtual nodes for each node $i \in V$, and derive a column stochastic $\boldsymbol{P}(t)$ on the augmented graph at each event $t \in \{1, \ldots, 2T\}$. *Second*, we sort out the complex effects of predictions, local updates, and communication of all learners via our proposed event indexing system. Then we derive a general regret bound in terms of the transition matrix $\boldsymbol{P}(t)$ at each event and the gradients $\boldsymbol{g}_t, \hat{\boldsymbol{g}}_t$ at each local update event. *Third*, we analyze the asymptotic properties of $\boldsymbol{P}(t)$ induced by our push-sum strategy. Although this part is enlightened by prior research [20, 23], we make an extra contribution to the convergence analysis of push-sum. Specifically, when analyzing the diffusion of any model copy over the network, we tackle all of its $n$ dimensions simultaneously by utilizing the fact that the rank of the updating matrix $\boldsymbol{\Delta}(t) = \boldsymbol{g}_t\boldsymbol{e}_{i_t}^\top$ is at most one. In this way, we save a factor of $\sqrt{n}$ in the convergence rate compared to the previous analysis [2]. As a consequence, our bound is now independent of $n$.

The above bound can be further simplified under the bounded gradient norm assumption [27].

**Corollary 1.** *Suppose $g_t \leq G, \hat{g}_t \leq G, \forall t \in \mathcal{Q}_T$ for some $G < \infty$. Then AD-OGP attains the regret*

$$\textbf{Regret}_j \leq \frac{m}{2\eta}\|\boldsymbol{w}_0 - \boldsymbol{w}^*\|^2 + \frac{2\eta}{m}G^2(T + D^{proc}) + \frac{2\eta}{m}(\frac{8+\alpha^4}{\alpha^5})BG^2T,$$

*for any reference learner $j \in V$, where $\alpha = (1/m)^B, B = (\mathcal{D}+1)(D^{msg} + \Gamma_d)$.*

***Remark*** (a) Suppose the diameter of the decision set $\mathcal{K}$ is bounded by $F$. Set $\eta = (mF/2G)(D^{proc} + T + CBT)^{-1/2}$, where $C = (8+\alpha^4)/\alpha^5$, then the bound will attain $O((CBT + T + D^{proc})^{1/2})$. It will be larger with more learners (larger $m$), heavier processing delays (larger $D^{proc}$), heavier message delays (larger $D^{msg}$), and "less connected" graphs (larger diameter $\mathcal{D}$).
(b) When the processing delays $d^p(t)$ are upper bounded, then $D^{proc} = O(T)$. Now the above derived bound reduces to $O((CBT)^{1/2})$, which is sublinear $O(\sqrt{T})$ w.r.t. $T$.
(c) The second term in the bound captures the effect of processing delays $\sum_{t \in \mathcal{Q}_T} d^p(t)$. This term is in line with the delay effect term in the asynchronous centralized online setting [18].
(d) The third term in the bound captures the effect of decentralization. Note that the factor $\alpha$ represents a lower bound of the positive entries in any $B$-hop transition matrix $\boldsymbol{P}(t+B-1)\cdots\boldsymbol{P}(t+1)\boldsymbol{P}(t), t \in \{1, \ldots, 2T - B\}$, which is commonly seen in the analysis of push-sum [1, 20, 2, 3].
(e) When there is no delays, namely $D^{proc} = D^{msg} = 0$, the bound reduces to that of its synchronous counterpart [1] (for a fair comparison, we remove the assumption of strong convexity in [1]).

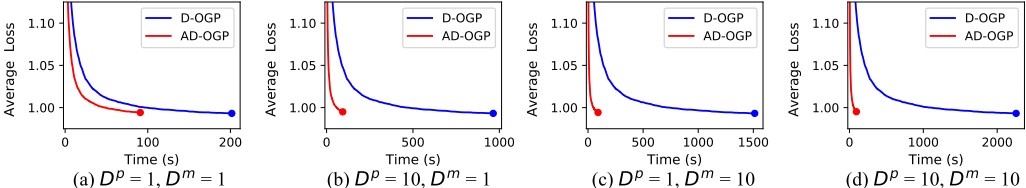

Figure 1: Illustration of the benefit of asynchronization. The four plots compare AD-OGP and D-OGP on 64-node Watts-Strogatz graphs on *pokerhand* under varying delay levels.

## 4    Experiments

In this section, we evaluate AD-OGP on two large-scale real-world online classification tasks. We first compare AD-OGP with its synchronous counterpart to show the advantage of asynchronization, then design elaborate experiments to verify the effectiveness of asymmetric gossiping and instantaneous model averaging in our algorithm design, and finally verify our theoretical regret bound.

### 4.1    Experimental Setup

We select two large-scale real-world datasets. (i) The *higgs* dataset is a benchmark dataset in high-energy physics [4] for binary classification, which consists of 11 million instances with 28 features. (ii) The *poker-hand* dataset is a commonly used dataset in automatic rule generation [6, 7] for 10-class classification, which has 1 million instances with 25 features. For binary classification, we use the logistic loss; for multi-class classification, we use the multivariate logistic loss [11] (see detailed definitions in supplementary materials).

In our experiments, the process of asynchronous decentralized online learning is simulated as follows. Given a dataset, the data instances are fed to the learners one after another at each prediction event in a streaming manner. We simulate three types of time intervals: (i) the computation time for each learner to calculate the gradients of data instances and perform local updates, (ii) the transmission time for each message to reach its gossiping receiver, and (iii) the waiting time for each learner to wait for the next prediction after it finishes a local update. Note that the waiting time is unique to the simulation of our online learning setting, which is not needed in that of stochastic optimization. Similar to the simulation method in [25], the computation time of learner $i$ follows the exponential distribution $\mathrm{Exp}(\mu_i)$ with mean value $\mu_i = |\bar{\mu}| + 2$, where $\bar{\mu}$ is sampled from the standard normal distribution $\mathrm{Normal}(0, 1)$. The transmission time of each message is independently sampled from $\mathrm{Exp}(0.6)$. The waiting time is independently sampled from $\mathrm{Exp}(\delta)$ (the mean value $\delta$ will be specified later). The units of all time are milliseconds. Note that we also examine other time distributions, such as uniform distributions, and observe similar results, which are omitted due to the page limit.

We use the *average loss* as the performance metric, which is a common practice in most online learning experiments [18, 31, 24]. Specifically, we randomly select a reference learner $j \in V$ and measure $\sum_{t \in \mathcal{P}_T} f_t(\boldsymbol{w}_j(t))/T$, where the model at each prediction event $t \in \mathcal{P}_T$ is given as $\boldsymbol{w}_j(t) = \boldsymbol{x}_j(t)/y_j(t)$. Moreover, we adopt the commonly used $L2$-norm balls as decision sets and set their diameters as $100$. The learning rate $\eta$ is set as what the corresponding theory suggests (see more details in supplementary materials).

### 4.2    The Benefit of Asynchronization

We compare AD-OGP with synchronous decentralized online gradient-push (D-OGP) [1], which synchronizes the local updates and communication of all learners at each round. We conduct the comparison on three representative types of graphs with varying connectivities [8, 31], i.e., the complete graph with high connectivity, the Watts-Strogatz graph as a kind of random graph with medium connectivity, and the ring graph with low connectivity (see detailed descriptions in supplementary materials).

We also investigate the effects of processing delays and message delays on the comparison of algorithms. We simulate different levels of delays via a technique commonly used in experiments of asynchronous decentralized stochastic optimization [17, 23]. Specifically, for the reference learner $j$,

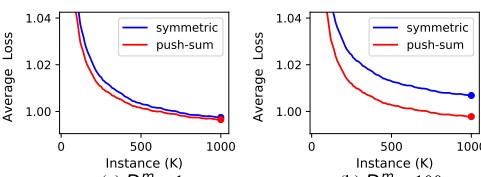
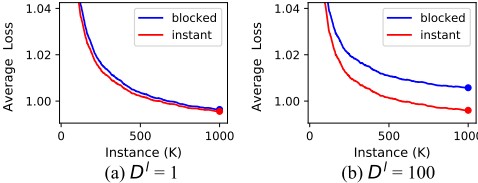

Figure 2: Illustration of the effectiveness of asymmetric gossiping. The two plots compare push-sum and symmetric gossiping [17] on a 64-node Watts-Strogatz graph on *pokerhand*. We enlarge the transmission time of any message sent from/to the reference learner $j$ by $D^m$ times.

Figure 3: Illustration of the effectiveness of instantaneous model averaging. The two plots compare AD-OGP and its counterpart with blocked averaging [3] on a 64-node Watts-Strogatz graph on *pokerhand*. We enlarge the waiting time of the reference learner $j$ by $D^l$ times.

we first enlarge its computation time by $D^p$ times throughout the learning process, where $D^p$ is a parameter that controls the level of processing delays. Then we randomly select a neighbor $k \in \mathcal{N}(j)$ of learner $j$ and enlarge the transmission time of any message from $j$ to $k$ or $k$ to $j$ by $D^m$ times, where $D^m$ is a parameter that controls the level of message delays. Similar to [17], we set $D^p = 1$ and $D^m = 1$ to represent a weak delay level, and set $D^p = 10$ and $D^m = 10$ to represent a strong delay level. As for the waiting time, we set the mean value $\delta = 0.01$ to make it relatively small compared to the processing time and the transmission time. In this way, the effect of waiting time on the total running time of both algorithms is almost negligible, and the running time difference is merely due to synchronization or asynchronization.

Figure 1 plots the performance of both algorithms on a 64-node Watts-Strogatz graph on *pokerhand* under varying levels of processing delays and message delays. The results on *higgs* as well as other types of graphs are qualitatively similar and deferred to supplementary materials due to the page limit. From the results, we draw two main observations. (i) AD-OGP runs substantially faster than D-OGP, which is in accord with our intuition that asynchronization eschews the straggler problem and hence saves considerable time. Moreover, the gap in the total running time dramatically increases as the delays become stronger. (ii) In all the examined settings, AD-OGP yields comparable performance to D-OGP, implying that asynchronization significantly improves the efficiency while hardly sacrificing the performance.

### 4.3 The Effectiveness of Asymmetric Gossiping and Instantaneous Model Averaging

We verify the effectiveness of the two innovations in AD-OGP by showing that, with either removed or weakened, the reduced algorithm will become less effective in some specific settings.

#### 4.3.1 The Effectiveness of Asymmetric Gossiping

We consider a scenario where the communication of a certain learner $i \in V$ is slow. In symmetric gossiping [17], each time learner $i$ sends a model copy to any of its neighbor $k \in \mathcal{N}(i)$, it must wait until receiving the response message sent back from $k$ before it can proceed to the next action. Thus its processing speed is slowed down. In contrast, in our asymmetric gossiping, the processing speed of such learner is not much affected since it is fully asynchronous. To verify this point empirically, we enlarge the transmission time of any message sent from/to the reference learner $j$ by $D^m$ times. We compare AD-OGP and the counterpart algorithm using symmetric gossiping (with [17]'s implementation) on a 64-node Watts-Strogatz graph under varying $D^m$. Results in Figure 2 show that symmetric gossiping will make the algorithm much less effective as message delays increase, which verifies the effectiveness of asymmetric gossiping.

#### 4.3.2 The Effectiveness of Instantaneous Model Averaging

We consider a scenario where a certain learner $i \in V$ is seldom selected by the environment to make predictions. In this scenario, each time learner $i$ completes a local update, it may wait for a long time until it is selected to make the next prediction. During this period, learner $i$ stays unoccupied and may consecutively receive model copies from its neighbors. In AD-OGP, these copies are used for model averaging immediately after they arrive. In contrast, if we directly apply other asynchronous

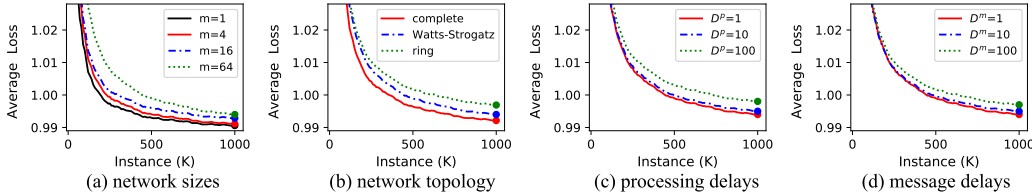

Figure 4: Verification of our regret bound. The four plots show AD-OGP's performance on *poker-hand* with varying characteristics. (a) Varying network sizes. (b) Varying network topology. (c) Varying levels of processing delays. (d) Varying levels of message delays.

stochastic algorithms [3, 23] to our online setting, these copies will be stored in learner $i$'s receiving buffer until it makes the next local update, which essentially magnifies the message delays and will harm the performance of the algorithms. To verify this point empirically, we set $\delta = 0.6$ in the waiting time simulation and enlarge the waiting time on the reference learner $j$ by $D^l$ times. We compare AD-OGP and the counterpart algorithm without using instantaneous model averaging (with [3]'s implementation which we term as blocked averaging) on a $64$-node Watts-Strogatz graph under varying $D^l$. The results in Figure 3 show that the counterpart algorithm will become much less effective when some learners need to wait for model averaging, which verifies the effectiveness of instantaneous model averaging.

### 4.4 Verification of the Theoretical Bound

We finally verify our derived regret bound by examining the effects of various factors on the performance of AD-OGP. Specifically, we run AD-OGP under varying network sizes, varying network topology, and varying levels of processing delays and message delays. Figure 4 presents the results on *poker-hand*. The results show that AD-OGP will incur higher losses with more learners, less connected networks, heavier processing delays, and heavier message delays, indicating an excellent agreement between empirical behaviors and theoretical results.

## 5 Conclusions

In this paper, we present the first systematic study of asynchronous decentralized online learning. We begin by formulating the framework of Asynchronous Decentralized Online Convex Optimization. Then we devise the Asynchronous Decentralized Online Gradient-Push algorithm, which is fully asynchronous and includes two novel innovations, i.e., weighted projection and instantaneous model averaging. Theoretically, we provide the first regret analysis paradigm for asynchronous decentralized online learning. Finally, we conduct extensive experiments to demonstrate the benefit of asynchronization, verify the effectiveness of the two innovations, and corroborate the theoretical bound. Our work paves the way for future research investigating asynchronous decentralized online learning.

## Limitations

As the first step of studying asynchronous learning in the decentralized online setting, our framework formulation and theoretical analysis are conducted in the convex setting. Although such limitation does not affect its usage in the non-convex setting, we would like to establish a formal non-convex analysis in the future.

## Acknowledgements

This work was supported by the National Key Research and Development Program of China No. 2020AAA0106300 and the National Natural Science Foundation of China (No. 62050110, No. 61936011, No. 62102222). This work was supported by Ant Group through Ant Research Intern Program. We thank Steven C. H. Hoi, Peilin Zhao, and Tong Zhang for introducing us to the fascinating research areas of online learning and distributed optimization. We also thank anonymous reviewers for their insightful and valuable comments to help improve the quality of this work.

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
