# Supplementary Materials:
# Asynchronous Decentralized Online Learning

Supplementary materials are organized as follows. Appendix A gives a more detailed review of related work. Appendix B analyzes the feasibility of the iterations of AD-OGP. Appendix C provides more detailed experimental setup and supplements additional empirical results. Appendix D and Appendix E provide detailed proofs of Theorem 1 and Corollary 1, which have been omitted in our main paper due to the space limit.

## A   More Related Work

In this section, we provide a more detailed review of related work in two fields, namely decentralized online learning and decentralized stochastic optimization.

### A.1   Decentralized Online Learning

In many distributed online learning scenarios such as learning on mobile networks or sensor networks, centralized architectures are usually unsuitable due to the very high computation and communication overhead on the central node [7]. As a general solution to reduce such overhead, decentralized online learning has received much research attention in recent years.

As mentioned in the main paper, most existing studies in this field are conducted in the synchronous setting, namely, they require some global coordination among learners during the learning process. Such coordination is used in either feedback processing, local update, or node communication (also see [3] for a more detailed explanation of synchronicity). Many previous works have investigated exemplar online algorithms in the synchronous decentralized setting [36, 13, 15, 24, 38, 16, 18]. Other recent researches have focused on developing synchronous algorithms in specific scenarios, such as learning on dynamic networks [1, 28, 37, 20], more efficient information exchange [32, 39, 33], or optimization under constraints [4].

Different from these previous works, we propose to conduct decentralized online learning in a **fully asynchronous** manner [3]. In particular, our work does not need any global synchronization mechanism to coordinate the learners throughout the learning process. In other words, in our framework, each learner makes predictions, processes feedback, performs local updates, sends out messages, receives messages, and performs model averaging independently; in particular, it does not need to know the states of other learners. As far as we are concerned, the recently proposed [14] is the only work that investigates delay and asynchronization in decentralized online learning. However, their work only allows learners to communicate specific gradients, hence it is restricted to dual averaging method where the decision only depends on historic gradients. By taking a completely different approach, our work allows each learner to communicate its model parameters with other learners, which is more efficient and suitable to more general online update methods.

### A.2   Decentralized Stochastic Optimization

Decentralized stochastic optimization aims to minimize the objective function $\sum_{i \in V} f_i / |V|$, where each learner $i \in V$ only has the access to a local loss $f_i(\boldsymbol{w}) = \mathbb{E}_{\boldsymbol{\xi} \sim \mathcal{D}} F_i(\boldsymbol{w}; \boldsymbol{\xi})$. In this setting, each instance point $\boldsymbol{\xi}$ are i.i.d. sampled from some fixed but unknown distribution $\mathcal{D}$; this is a core assumption in the stochastic optimization setting. Previous studies in this field can be classified into

35th Conference on Neural Information Processing Systems (NeurIPS 2021).

two categories: synchronous algorithms and asynchronous algorithms. In synchronous algorithms, at each iteration, all learners compute stochastic gradients locally and are synchronized to update their local models at the same pace [7, 26, 29, 30, 23, 17]. However, as stated before, synchronous algorithms suffer from the straggler problem. To resolve this problem, many recent works have developed asynchronous algorithms for stochastic optimization [21, 8, 35, 12, 2, 31].

Despite the existence of many previous works for asynchronous decentralized stochastic optimization, we note that, as a first systematic study for asynchronous decentralized online learning, our work is largely different from them in the following three aspects. *First*, we contribute the first framework of asynchronous decentralized online learning. In particular, we propose an event indexing system by taking both prediction and local update into account, which is distinct from previous stochastic frameworks using the update indexing system [21] in both formulation and analysis. *Second*, our algorithm has two specific and novel designs in the online setting, namely weighted projection and instantaneous model averaging. Both designs make our algorithm more effective than simply tailoring standard stochastic optimization algorithms [21, 2, 31] to the online setting. *Third*, the regret analysis for decentralized online learning is intrinsically different from the convergence analysis for decentralized stochastic optimization.

## B  The Feasibility of AD-OGP

In this section, we strictly prove that the predictions generated by our proposed AD-OGP algorithm are guaranteed to lie in the feasible region $\mathcal{K}$. The proof is not straightforward because the predictions $\boldsymbol{w}_i$ of learner $i$ are determined by its model $\boldsymbol{x}_i$ and push-sum weight $y_i$ together, namely $\boldsymbol{w}_i = \boldsymbol{x}_i/y_i$; during the learning process, both local update and model averaging can change the parameters $\boldsymbol{x}_i$ and $y_i$, which possibly induces predictions that lie outside of the feasible region.

We first introduce some notations to specify each learner $i$'s parameters $\boldsymbol{x}_i$ and $y_i$ at some certain event time points $t \in \{1, \ldots, 2T\}$. Specifically, we use $\boldsymbol{x}_i(t)$ and $y_i(t)$ to denote its most recent local model and push-sum weight *before* event $t$; and use $\boldsymbol{x}_i'(t)$ and $y_i'(t)$ to denote its local model and push-sum weight immediately *after* event $t$. Then its predictions immediately before and after event $t$ are $\boldsymbol{w}_i(t) = \boldsymbol{x}_i(t)/y_i(t)$ and $\boldsymbol{u}_i(t) = \boldsymbol{x}_i'(t)/y_i'(t)$, respectively. To help better understand our algorithm, we describe the full procedure of AD-OGP in Algorithm 1, which is aligned with our proposed AD-OCO framework (Protocol 1 in our main paper).

**Proposition 1.** *In AD-OGP, for any $i \in V, t \in \mathcal{P}_T$, it holds that*

$$y_i(t) > 0, \quad \boldsymbol{w}_i(t) = \frac{\boldsymbol{x}_i(t)}{y_i(t)} \in \mathcal{K}.$$

*Proof.* As a prerequisite of proving feasibility, we first show that, for each learner $i \in V$, its push-sum weight $y_i(t)$ is always positive; otherwise, the division $\boldsymbol{x}_i(t)/y_i(t)$ may be invalid.

**Lemma 1.** *In AD-OGP, for each learner $i \in V$, its push-sum weight is positive throughout the learning process, i.e., $y_i(t) > 0, y_i'(t) > 0, \forall t \in \{1, \ldots, 2T\}$.*

*Proof.* There are only two types of actions that may change the value of push-sum weights, namely local update $\mathcal{U}$ and model averaging $\mathcal{A}$. Consider an arbitrary learner $i \in V$. At each of its local update event, its push-sum weight $y_i$ is multiplied by a factor of $\gamma_i$ ($1/m \leq \gamma_i < 1$). Moreover, each time it executes a model averaging operation, its push-sum weight $y_i$ is added with some weight $y'$ of another learner. In fact, the positiveness of $y_i(t)$ can be proved by inducting on $t \in \{1, \ldots, 2T\}$.

At the beginning of the learning process ($t = 1$), for each learner $i \in V$, its push-sum weight is initialized as $y_i(1) = 1$. Since the first event must be a prediction event, we must have $y_i'(1) = y_i(1) = 1$ for any $i \in V$. Now we suppose that, for some $t \leq 2T$, we have $y_i(l) > 0, y_i'(l) > 0, \forall i \in V, \forall l \in \{1, \ldots, t\}$. We now check the positiveness of $y_i(t+1)$ and $y_i'(t+1)$.

(i) During the interval between events $t$ and $t + 1$, $y_i$ can only change during model averaging on learner $i$. Recall the notations in our framework, namely, learner $i$ performs model averaging using the copies in $\mathcal{M}_i(t)$ during this interval. If $\mathcal{M}_i(t) = \emptyset$, we directly have $y_i(t + 1) = y_i'(t) > 0$. If $\mathcal{M}_i(t) \neq \emptyset$, consider any single copy $(\boldsymbol{x}', y') = (\gamma_j \boldsymbol{x}_j', \gamma_j y_j) \in \mathcal{M}_i(t)$. We suppose that its corresponding message delay is $d'$. Then the weight $y_j'$ is exactly learner $j$'s push-sum weight

---

**Algorithm 1** Asynchronous Decentralized Online Gradient-Push (**AD-OGP**)

---

1: **Input:** Time horizon $T$, convex set $\mathcal{K}$, and learning rate $\eta$.
2: **Initialize:** Local model $\boldsymbol{x}_j(1) \leftarrow \boldsymbol{w}_0 \in \mathcal{K}$ and push-sum weight $y_j(1) \leftarrow 1, \forall j \in V$.
3: **for** $t = 1, \ldots, 2T$ **do**
4:     **if** $\delta_t = 0$ **then**                                         *// for a prediction event*
5:         Learner $i_t$ predicts with $\boldsymbol{w}_{i_t}(t) = \boldsymbol{x}_{i_t}(t)/y_{i_t}(t)$.
6:         Learner $i_t$ starts to compute the gradient $\nabla f_t(\boldsymbol{w}_{i_t}(t))$.
7:         (Parameters of *all* learners $j \in V$ stay unchanged: $\boldsymbol{x}'_j(t) = \boldsymbol{x}_j(t), y'_j(t) = y_j(t)$.)
8:     **else**                                                       *// for a local update event*
9:         Learner $i_t$ performs weighted projected gradient descent:

$$\boldsymbol{x}'_{i_t}(t) = \gamma_{i_t} y_{i_t}(t) \Pi_{\mathcal{K}}\left(\frac{\boldsymbol{x}_{i_t}(t) - \eta \nabla f_{l_t}(\boldsymbol{w}_{i_t}(l_t))}{y_{i_t}(t)}\right), \quad y'_{i_t}(t) = \gamma_{i_t} y_{i_t}(t).$$

10:         Learner $i_t$ sends $\boldsymbol{x}'_{i_t}(t)$ and $y'_{i_t}(t)$ to each neighbor $j \in \mathcal{N}(i_t)$.
11:         (Parameters of *other* learners $j \neq i_t$ stay unchanged: $\boldsymbol{x}'_j(t) = \boldsymbol{x}_j(t), y'_j(t) = y_j(t)$.)
12:     **end if**
13:     **for** $i \in V$ **do**
14:         Learner $i$ performs model averaging as long as it is *not* occupied:

$$\boldsymbol{x}_i(t+1) = \boldsymbol{x}'_i(t) + \sum_{\boldsymbol{x}' \in \mathcal{M}_i(t)} \boldsymbol{x}', \quad y_i(t+1) = y'_i(t) + \sum_{y' \in \mathcal{M}_i(t)} y'.$$

15:     **end for**
16: **end for**

---

immediately after the event $t - d'$, namely $y'_j(t - d')$. Since we have assumed that $y'_j(t - d') > 0$, we thus have $y_i(t+1) = y'_i(t) + \sum_{(\boldsymbol{x}', y') \in \mathcal{M}_i(t)} y' > 0$.

(ii) At the event $t + 1$, from Algorithm 1 we know that either $y'_i(t+1) = y_i(t+1)$ or $y'_i(t+1) = \gamma_i y_i(t+1)$. Since $\gamma_i > 0$, in both cases we have $y'_i(t+1) > 0$.

In summary, we have $y_i(t+1) > 0$ and $y'_i(t+1) > 0$ for any $i \in V$. Hence we prove this lemma. $\qquad \square$

Now we can formally analyze the feasibility of AD-OGP. In intuition, we expect that both operations of local update and model averaging will not violate the feasibility of the generated predictions. Recall that, in our algorithm, for each local update, the feasibility is always preserved by our proposed weighted projection mechanism. Hence we only need to prove that, during model averaging, the feasibility of AD-OGP is also preserved. Indeed, such a property is implied by the following lemma.

**Lemma 2.** *Suppose the parameters $(\boldsymbol{x}_i, y_i)$ of all learners satisfy $y_i > 0, \boldsymbol{x}_i/y_i \in \mathcal{K}, \forall i \in V$. Then given any averaging weights $\{\alpha_i\}_{i \in V}$ such that $\alpha_i \geq 0, \forall i \in V$ and $\sum_{i \in V} \alpha_i > 0$, it holds that*

$$\frac{\sum_{i \in V} \alpha_i \boldsymbol{x}_i}{\sum_{i \in V} \alpha_i y_i} \in \mathcal{K}.$$

*Proof.* We first consider the simple case of two learners with the same weights $\alpha_1 = \alpha_2 > 0$. In this case, we directly have

$$\frac{x_1 + x_2}{y_1 + y_2} = \frac{y_1}{y_1 + y_2} \cdot \frac{x_1}{y_1} + \frac{y_2}{y_1 + y_2} \cdot \frac{x_2}{y_2},$$

thus $(x_1 + x_2)/(y_1 + y_2)$ is a convex combination of $x_1/y_1$ and $x_2/y_2$. If $x_1/y_1, x_2/y_2 \in \mathcal{K}$, the parameters after model averaging will still generate the prediction $(x_1 + x_2)/(y_1 + y_2)$ that lies in $\mathcal{K}$.

Now we prove this lemma by inducting on the cardinality $m$ of $V$. It trivially holds for the single-learner setting where $m = 1$. Now we suppose that it is satisfied under $m \leq k$ for some $k \geq 1$. We now intend to prove that it still holds when $m = k + 1$. Denote $V = \{1, \ldots, k + 1\}$. Since $\sum_{i \in V} \alpha_i > 0$, $\alpha_i$ cannot be all zero for $i \in V$. We now consider the following two cases.

(i) There is only a single $\alpha_i$ being positive. Then, without loss of generality, we can assume $\alpha_1 > 0$ and $\alpha_i = 0$ for any $i > 1$. In this case, we trivially have $(\alpha_1 x_1)/(\alpha_1 y_1) \in \mathcal{K}$.

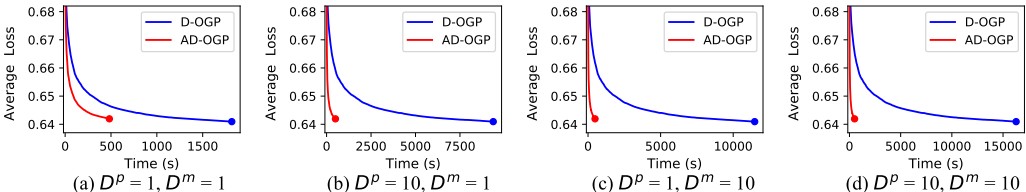

Figure 5: More results to verify the benefit of asynchronization. The plots compare AD-OGP and D-OGP on a 64-node *ring* graph under varying levels of processing delays $D^p$ and message delays $D^m$ on *higgs*.

(ii) There are at least two $\alpha_i$ that are positive. Without loss of generality, we can assume $\alpha_1, \alpha_2 > 0$. Since $\sum_{i=2}^{k+1} \alpha_i > 0$, from our assumption we have $(\sum_{i=2}^{k+1} \alpha_i x_i)/(\sum_{i=2}^{k+1} \alpha_i y_i) \in \mathcal{K}$. Set $x_1' = \alpha_1 x_1, y_1' = \alpha_1 y_1, x_2' = \sum_{i=2}^{n+1} \alpha_i x_i, y_2' = \sum_{i=2}^{k+1} \alpha_i y_i$, then from the initial case of two learners, we have $(x_1' + x_2')/(y_1' + y_2') \in \mathcal{K}$.

Combining the above two cases, we check the setting of $|V| = k + 1$. Hence, by inducting on $k$, we prove the lemma. □

Finally, we can prove the feasiblity of AD-OGP by inducting on $t \in \{1, \ldots, 2T\}$. Recall that, at the beginning of the learning process $t = 1$, each learner $i \in V$ is initialized as $\boldsymbol{x}_i(1) = \boldsymbol{w}_0 \in \mathcal{K}$ and $y_i(1) = 1$; and $\boldsymbol{x}_i'(1) = \boldsymbol{x}_i(1), y_i'(1) = y_i(1)$.

We now suppose that for some $t \leq 2T$, it holds that $\boldsymbol{x}_i(l)/y_i(l) \in \mathcal{K}, \boldsymbol{x}_i'(l)/y_i'(l) \in \mathcal{K}, \forall i \in V, l \in \{1, \ldots, t\}$. Then during the interval between events $t$ and $t + 1$, each learner $i \in V$ may perform model averaging using the copies in $\mathcal{M}_i(t)$. From Lemma 2 we know that $\boldsymbol{x}_i(t+1)/y_i(t+1) \in \mathcal{K}$. In addition, recall our designed mechanism of weighted projection, we also have $\boldsymbol{x}_i'(t+1)/y_i'(t+1) \in \mathcal{K}$. Consequently, we prove the proposition by induction. □

## C  More Details of Experiments

In this section, we first provide more details of our experimental setup and algorithm configurations, then supplement additional empirical results.

### C.1  A Summary of Baselines

In our experiments, we have compared AD-OGP with three baseline algorithms. We now give a summary of the compared baselines, as the readers might expect.

1. *Synchronous online gradient push* [2]. This is the synchronous counterpart of AD-OGP. It is compared in Figure 1 of Section 4.1, which verifies the efficiency of AD-OGP compared to its synchronous counterpart.

2. *Asynchronous stochastic gradient descent* [4]. This is a classic asynchronous decentralized algorithm from stochastic optimization. We tailor it to the pure online setting, and present the comparison results in Figure 2 of Section 4.3.1. It verifies the effectiveness of asymmetric gossiping in AD-OGP.

3. *Asynchronous stochastic gradient-push* [5]. This is the state-of-the-art asynchronous algorithm from stochastic optimization. We also tailor it to the pure online setting, and present the comparison results in Figure 3 of Section 4.3.2 (Figure 3 is on the right of Figure 2). It verifies the effectiveness of instantaneous model averaging in AD-OGP.

### C.2  Logistic Loss for Binary Classification

In binary classification, the class set is $\mathcal{C} = \{\pm 1\}$. Each learner is parametrized by the (vector) model $\boldsymbol{x} \in \mathbb{R}^n$ and the push-sum weight $y > 0$. Given the feature vector of an instance $\boldsymbol{\xi} \in \mathbb{R}^n$, the learner

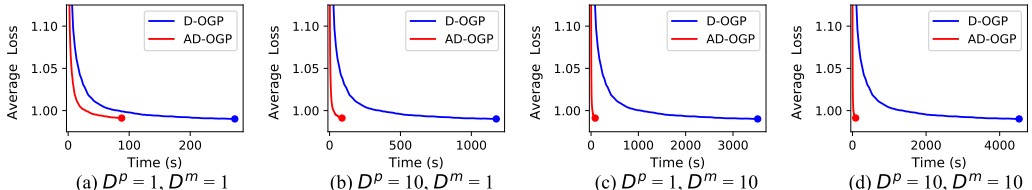

Figure 6: More results to verify the benefit of asynchronization. The four plots compare AD-OGP and D-OGP on a 64-node *complete* graph under varying levels of processing delays $D^p$ and message delays $D^m$ on *pokerhand*.

predicts its class label as $\text{sgn}(\boldsymbol{w}^\top \boldsymbol{\xi})$, where $\boldsymbol{w} = \boldsymbol{x}/y$ is restricted in $\mathcal{K}$. Suppose the true class label of such instance is $l \in \{\pm 1\}$. We adopt the logistic loss, which is defined as

$$f(\boldsymbol{w}) = \log(1 + \exp(-l\boldsymbol{w}^\top \boldsymbol{\xi})).$$

### C.3 Multivariate Logistic Loss for Multi-Class Classification

In $H$-class classification ($H \geq 3$), the class set is $\mathcal{C} = \{1, \ldots, H\}$. Each learner is parametrized by the model (in a matrix form) $\boldsymbol{X} = [\boldsymbol{x}_1; \cdots ; \boldsymbol{x}_H] \in \mathbb{R}^{n \times H}$ and the push-sum weight $y > 0$. Given the feature vector of an instance $\boldsymbol{\xi} \in \mathbb{R}^n$, the learner predicts its class label as $\arg\max_{h \in \mathcal{C}} \boldsymbol{w}_h^\top \boldsymbol{\xi}$, where $\boldsymbol{w}_h = \boldsymbol{x}_h/y, \forall h \in \mathcal{C}$, is restricted within $\mathcal{K}$. Suppose the true class label of the instance is $l \in \mathcal{C}$. We adopt the multivariate logistic loss [11], which is defined as

$$f(\boldsymbol{W}) = \log(\sum_{h \in \mathcal{C}} \exp(\boldsymbol{w}_h^\top \boldsymbol{\xi} - \boldsymbol{w}_l^\top \boldsymbol{\xi})),$$

where $\boldsymbol{W} = \boldsymbol{X}/y$ is the decision (in a matrix form). Note that, the matrix form is compatible with our framework and theoretical analysis. Specifically, for each learner, we can concatenate all column vectors $\boldsymbol{x}_1, \ldots, \boldsymbol{x}_h$ of the matrix model $\boldsymbol{X}$ into a $nh \times 1$ vector $\boldsymbol{x} \in \mathbb{R}^{nh}$, and all column vectors $\boldsymbol{w}_1, \ldots, \boldsymbol{w}_h$ of the matrix prediction $\boldsymbol{W}$ into a $nh \times 1$ vector $\boldsymbol{w} \in \mathbb{R}^{nh}$. Then we have $\boldsymbol{w} = \boldsymbol{x}/y$, and the loss function $f$ is convex w.r.t. $\boldsymbol{w}$.

### C.4 Network Topology

We here provide detailed descriptions of the various types of network topology examined in our experiments, namely the complete graph, the Watts-Strogatz graph, and the ring graph. All these types of graphs are commonly used in previous studies for decentralized algorithms [7, 38], which represent different levels of connectivity.

1. *Complete graph.* In this kind of graph, each node is connected to any other node. Any complete graph has a diameter of $\mathcal{D} = 1$, which represents a high level of connectivity.

2. *Ring graph.* In this kind of graph, all nodes are arranged in the shape of a ring, and each node is only connected to its two immediate neighbors in the ring. The ring graph with $m$ nodes has a diameter of $\mathcal{D} = \lfloor m/2 \rfloor$, which represents a low level of connectivity.

3. *Watts-Strogatz graph.* This is a kind of random graph, which has two parameters that control its topology, namely the average degree $k$ and the rewiring probability $p$. In general, a higher average degree or a higher rewiring probability will result in better connectivity of the generated random graph [6]. In our experiments, we follow [34] and set $k = 4, p = 0.3$, to generate random graphs that represent a medium level of connectivity.

### C.5 More Details of Algorithm Configurations

We adopt L2-norm balls as the decision set in our experiments. Specfically, for binary classification, it is defined as $\mathcal{K} = \{\boldsymbol{w} \in \mathbb{R}^n \mid \|\boldsymbol{w}\| \leq F/2\}$, where $\|\cdot\|$ denotes the L2-norm of any vector. For multi-class classification, it is defined as $\mathcal{K} = \{\boldsymbol{W} \in \mathbb{R}^{n \times H} \mid \|\boldsymbol{W}\|_F \leq F/2\}$, where $\|\boldsymbol{W}\|_F$ is the

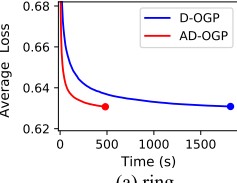 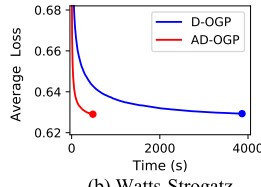 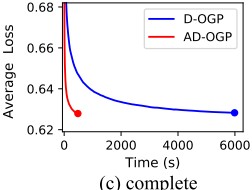

(a) ring  (b) Watts-Strogatz  (c) complete

Figure 7: Illustration of the efficiency of AD-OGP in the non-convex setting. The plots compare FMs using AD-OGP and D-OGP on 64-node graphs of varying topologies on *higgs*.

Frobenius norm of the matrix $\boldsymbol{W} = [\boldsymbol{w}_1; \ldots; \boldsymbol{w}_H] \in \mathbb{R}^{n \times H}$, i.e., $\|\boldsymbol{W}\|_F = (\sum_{h=1}^H \|\boldsymbol{w}_h\|^2)^{1/2}$. We set their diameters $F = 10$.

To configure the learning rate, we follow the common practice in asynchronous online learning [22] by setting $\eta = \alpha \eta^*$, where $\eta^*$ is the appropriate learning rate theoretically suggested by the given algorithm, and $\alpha$ is chosen from a grid search over $\{\alpha_0 (1.25)^i \mid i \in \mathbb{N}\}$ ($\alpha_0$ is an initial guess for each dataset). In particular, as suggested by Corollary 1 in our main paper, the theoretically optimal $\eta^*$ is determined by several quantities, namely $T, D^{proc}, D^{msg}, \Gamma_u$ and $G$. In our experiments, we use $G = 1$, which has been suggested as a generally good choice in practice [22]. To specify the quantities related to delays ($D^{proc}, D^{msg}$ and $\Gamma_u$), we first simulate a small number of rounds (say $T_0 = 10000$) and measure these quantities within the first $T_0$ rounds [19], and then use them to estimate the quantities over the entire time horizon $T$.

Note that, although our proposed framework characterizes the number of predictions $N_i$ for each learner $i \in V$, we do NOT need to know these values. Instead, we only need to know the time horizon $T = \sum_{i \in V} N_i$. In some specific scenarios where $T$ is unknown, we can adopt the commonly used doubling trick technique [5] or tune the learning rates by hand [35].

In our experimental setup, the time complexity of each local update or model averaging operation is in $O(n)$, where $n$ is the dimension of model parameters. Hence the overall time complexity is $O(nT)$. All runs are deployed on Xeon(R) E5-2699 @ 2.2GHz.

### C.6 More Results to Verify the Benefit of Asynchronization

In our main paper, we report the results of the experiment on Watts-Strogatz graphs on *pokerhand*. Here we also compare AD-OGP and D-OGP on *higgs* as well as other types of networks (i.e., ring graphs or complete graphs). Specifically, we measure the performance of both algorithms on a 64-node ring graph on *higgs*, as well as on a 64-node complete graph on *pokerhand*, under varying levels of processing delays $D^p$ and message delays $D^m$ (recall that, we use $D^p$ ($D^m$) $= 1$ to represents a low delay level and $D^p$ ($D^m$) $= 10$ to represent a high delay level). The results are plotted in Figure 5 and Figure 6 respectively. They are consistent with the empirical results presented in our main paper. In particular, AD-OGP runs significantly faster than its synchronous counterpart while incurring hardly any loss in the performance.

### C.7 Experiments in the Non-Convex Setting

Recall that, our framework and theoretical analysis originate from online convex optimization, which relies on the convexity assumption [10]. Nevertheless, in principle, our proposed AD-OGP algorithm can also be applied to the non-convex setting. Here we conduct additional experiments to verify this point, namely, the efficiency of AD-OGP against its synchronous counterpart [28] is consistent in the non-convex setting.

In this experiment, we adopt the Factorization Machine (FM) [27] as the non-convex model. Specifically, a FM model of degree $d \in \mathbb{N}_+$ is parametrized by a weight vector $\boldsymbol{w} \in \mathbb{R}^n$, a bias $w_0 \in \mathbb{R}$, and an interaction matrix $\boldsymbol{V} \in \mathbb{R}^{n \times d}$. Given the feature vector $\boldsymbol{\xi} = (\xi_1, \ldots, \xi_n) \in \mathbb{R}^n$ of any instance,

FM predicts its value as

$$h(\boldsymbol{\xi}; \boldsymbol{w}, w_0, \boldsymbol{V}) = w_0 + \boldsymbol{w}^\top \boldsymbol{\xi} + \sum_{i=1}^{n} \sum_{j=i+1}^{n} \langle \boldsymbol{v}_i, \boldsymbol{v}_j \rangle \xi_i \xi_j,$$

where $\boldsymbol{v}_i$ represents the $i$-th row of $\boldsymbol{V}$.

We conduct online binary classification experiments on *higgs* and adopt the logistic loss. Specifically, the class label of $\boldsymbol{\xi}$ is predicted as $\mathrm{sgn}(h(\boldsymbol{\xi}; \boldsymbol{w}, w_0, \boldsymbol{V}))$. Suppose its true class label is $l \in \{\pm 1\}$, then the loss function is given as $f(\boldsymbol{w}, w_0, \boldsymbol{V}) = -\log \sigma(l \cdot h(\boldsymbol{\xi}; \boldsymbol{w}, w_0, \boldsymbol{V}))$ where $\sigma(u) = 1/(1 + e^{-u})$.

We employ AD-OGP and its synchronous counterpart D-OGP in the aforementioned non-convex setting. In the experiment, we set the FM's degree $d$ to be 8, which is chosen via a grid search. Other configurations are set to be the same as those for the convex experiments in our main paper. Figure 7 presents the results on 64-node graphs with different topologies. The results show that, in the online non-convex setting, AD-OGP runs significantly faster than its synchronous counterpart, with negligible loss in performance, which accords well with our intuition that AD-OGP can also be applied to the non-convex setting.

# D  Omitted Proof of Theorem 1

In this section, we present the detailed proof of Theorem 1, which is omitted in our main paper.

Our proof consists of three steps. First, we extend the graph augmentation technique [9] to our online setting to handle message delays, then explicate the transition matrices $\boldsymbol{P}(t)$ at each round $t \in \{1, \ldots, 2T\}$. Second, we decouple the effect of prediction, local update and model averaging, then derive a general regret bound in terms of the transition matrices $\boldsymbol{P}(t)$ and the associated gradients $\boldsymbol{g}_t$ and $\hat{\boldsymbol{g}}_t$. Third, we look deep into the mechanism of push-sum and analyze its convergence rate; plugging it into the above general bound, we successfully derive the bound required in Theorem 1.

## D.1  Formulate Transition Matrices on the Augmented Graph

As the first step of regret analysis, we characterize the transition matrices $\boldsymbol{P}(t)$ at each iteration $t \in \{1, \ldots, 2T\}$. Recall the message sending after any local update event $t \in \mathcal{Q}_T$, namely, the updated learner $i_t$ will sent out a copy of its updated parameters $(\boldsymbol{x}'_{i_t}(t), y'_{i_t}(t))$ to each of its neighbor in $\mathcal{N}(i_t)$. Consider its arbitrary neighbor $j \in \mathcal{N}(i_t)$, and suppose that the message $(\boldsymbol{x}'_{i_t}(t), y'_{i_t}(t))$ will experience a message delay of $d_{i_t j}^m(t)$. For simplicity, abbreviate $(\boldsymbol{x}'_{i_t}(t), y'_{i_t}(t))$ into $(\boldsymbol{x}', y')$ and $d_{i_t j}^m(t)$ into $d$. Then such message will be used for learner $j$'s model averaging between events $t + d$ and $t + d + 1$. Notably, it is captured by NONE of the learners during the interval $(t, t + d]$.

Such a property will make the recursive relation of local models at different time points extremely complex. In the above example, due to the message delay, a copy of learner $i_t$'s model $(\boldsymbol{x}'_{i_t}(t), y'_i(t))$ after its local update event $t$ will be added to learner $j$'s model $(\boldsymbol{x}_j(t + d + 1), y_j(t + d + 1))$ at event $t + d + 1$. Since the message delay $d$ is at most $D^{msg}$ (see Assumption 1), the order of the recursion can be as large as $D^{msg} + 1$. Moreover, each message has its specific delay. Hence the recursion of parameters will be very complex, which makes the analysis almost intractable.

To alleviate this complexity, we introduce the graph augmentation technique [9]. Its main idea is to append some virtual nodes to the decentralized network $G$, so that any message that undergoes delays will be captured by exactly one of the virtual nodes during its transmission. With this approach, the algorithm can be viewed as running on the augmented graph with zero message delay. Then we can derive a first-order recursion of parameters, which is relatively easier in the subsequent analysis.

Formally, the augmented version $\tilde{G} = (\tilde{V}, \tilde{E})$ of $G = (V, E)$ is constructed in the following two steps. First, regarding each actual node $i \in V$ in the original graph, we add $D^{msg}$ virtual nodes $b_i^1, \ldots, b_i^{D^{msg}}$. Second, we add two types of virtual edges to connect these virtual nodes and some actual nodes: (i) Corresponding to each actual node $i \in V$, we add $D^{msg}$ virtual edges, namely $(b_i^1, i)$ and $(b_i^{l+1}, b_i^l), \forall l \in \{1, \ldots, D^{msg} - 1\}$; then any actual node $i \in V$ and its corresponding $D^{msg}$ virtual nodes $b_i^1, \ldots, b_i^{D^{msg}}$ form a chain $(i, b_i^1, \ldots, b_i^{D^{msg}})$ in the augmented graph $\tilde{G}$. (ii) Corresponding to each actual edge $(i, j) \in E$, we add $2D^{msg}$ virtual edges $(b_i^l, j), (b_j^l, i), \forall l \in$

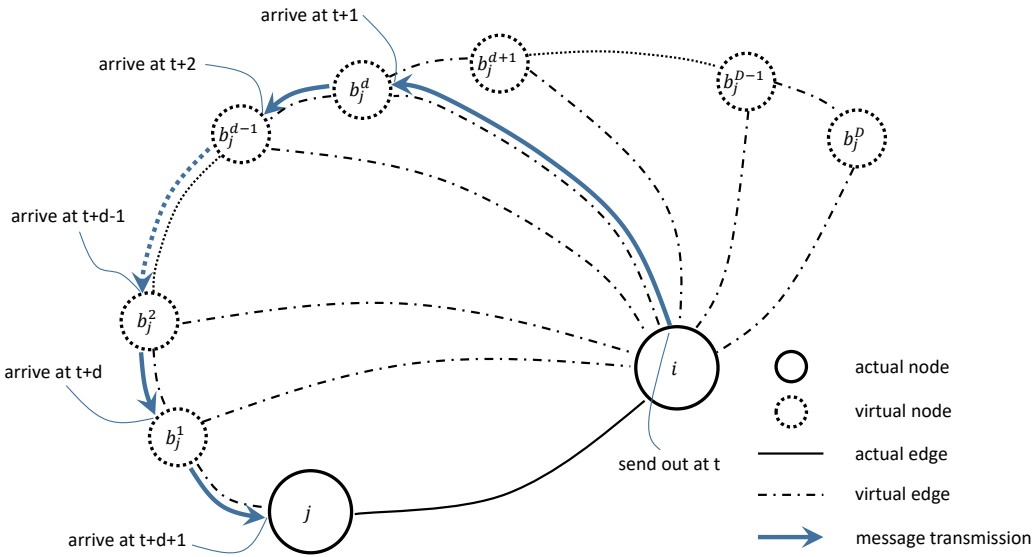

Figure 8: Illustration of message passing on the augmented graph. The two solid circles $i$ and $j$ represent two actual nodes. The solid curve represent the actual edge $(i, j)$. The $D$ dash circles denote all virtual nodes corresponding to the actual node $j$, namely $b_j^1, \ldots, b_j^D$. The dash curves represent two types of virtual edges, i.e., those in the chain $(b_j^D, \ldots, b_j^1, j)$ as well as those corresponding to the actual edge $(i, j)$. The blue arrows illustrate the transmission of the message sent from node $i$ to node $j$ with a message delay of $d$. In particular, it transmits along $i \to b_j^d \to \cdots \to b_j^1 \to j$.

$\{1, \ldots, D^{msg}\}$. Mathematically, the augmented graph $\tilde{G}$ is constituted by the augmented vertex set

$$\tilde{V} = V \cup \{b_i^l \mid i \in V, 1 \le l \le D^{msg}\},$$

and the augmented edge set

$$\tilde{E} = E \cup \{(b_i^{D^{msg}}, b_i^{D^{msg}-1}), \ldots, (b_i^2, b_i^1), (b_i^1, i) \mid i \in V\}$$
$$\cup \{(b_i^l, j), (b_j^l, i) \mid (i, j) \in E, 1 \le l \le D^{msg}\}.$$

To ensure that the message is captured by exactly one virtual node during its transmission, we can imagine an information flow along the chain $(b_j^{D^{msg}}, \ldots, b_j^1, j)$. Specifically, suppose that some message is sent from learner $i$ to learner $j$ after some event $t \in \{1, \ldots, 2T\}$, which has a message delay of $d \ge 0$. Then we can imagine that it first arrives at the virtual node $b_j^d$ at event $t + 1$, then is passed to $b_j^{d-1}$ at event $t + 2$, and so forth until it arrives at $b_j^1$ at event $t + d$, and finally reaches its destination $j$ at event $t + d + 1$. In this imaginary process, this message is captured by exactly one node (either actual or virtual) at each step during its transmission. Therefore, the message can be viewed as transmitting via the path $i \to b_j^d \to \cdots \to b_j^1 \to j$, with zero message delay at each hop. Note that, when $d = 0$, the transmission trivially reduces to $i \to j$, namely, the message reaches node $j$ immediately and is processed by node $j$ before the incoming event. To better explicate this point, in Figure 8 we give a demonstration of how a message is diffusing over the augmented graph. Note that, for conciseness, we abbreviate the maximal message delay $D^{msg}$ into $D$ throughout the proof of Theorem 1.

Now we can characterize the evolving dynamics of the whole process via formulating the recursive relation of parameters. For each virtual node $b_i^l \in \tilde{V} - V$, we assume that it also maintains model parameters $\boldsymbol{x}_i^l$ and push-sum weight $y_i^l$ during the process, and initialize these parameters as $\boldsymbol{x}_i^l(1) = \boldsymbol{0} \in \mathbb{R}^n$ and $y_i^l(1) = 0$. In addition, to give a more concise expression of the nodes in $\tilde{G}$, for each actual node $i \in V$, we also represent it by $b_i^0$ and rewrite its parameters as $(\boldsymbol{x}_i^0(t), y_i^0(t)) = (\boldsymbol{x}_i(t), y_i(t))$. In the following, we will can formulate the evolution of model parameters $(\boldsymbol{x}_i^l(t), y_i^l(t))$ of either actual ($l = 0$) or virtual ($l \ge 1$) node throughout the learning process.

**(A)** First we analyze the evolution of model parameters *at any specific event* $t \in \{1, \ldots, 2T\}$. If event $t$ is a local update event ($t \in \mathcal{P}_T$), then the local model of any learner will not change at this event. If event $t$ is a local update event ($t \in \mathcal{Q}_T$), then $i_t$ is the only learner whose model parameters changes at this event, with the update form

$$\boldsymbol{x}'_{i_t}(t) = \gamma_{i_t} y_{i_t}(t) \Pi_{\mathcal{K}}(\frac{\boldsymbol{x}_{i_t}(t) - \eta \boldsymbol{g}_t}{y_{i_t}(t)}), \quad y'_{i_t}(t) = \gamma_{i_t} y_{i_t}(t).$$

In the above update form, we first cast aside the multiplication operation with $\gamma_{i_t}$ (which is regarded as part of model averaging and will be tackled later). Denote the local model immediately after the projected gradient descent step as

$$\boldsymbol{z}_{i_t}(t) = y_{i_t}(t) \Pi_{\mathcal{K}}(\frac{\boldsymbol{x}_{i_t}(t) - \eta \boldsymbol{g}_t}{y_{i_t}(t)}).$$

Since the projection operation $\Pi_{\mathcal{K}}$ will introduce nonlinearity to the local update, we denote the residual of weighted projection as $\boldsymbol{r}_t = \boldsymbol{z}_{i_t}(t) - (\boldsymbol{x}_{i_t}(t) - \eta \boldsymbol{g}_t)$ and linearize the local update as

$$\boldsymbol{z}_{i_t}(t) = \boldsymbol{x}_{i_t}(t) - \eta \boldsymbol{g}_t + \boldsymbol{r}_t.$$

More compactly, we denote the incremental term associated with this local update as $\boldsymbol{\epsilon}_t = -\eta \boldsymbol{g}_t + \boldsymbol{r}_t \in \mathbb{R}^n$. In summary, for each learner $i \in V$, the change of its local model (without multiplication with $\gamma_i$) at any event $t \in \{1, \ldots, 2T\}$ takes

$$\boldsymbol{z}_i^l(t) = \begin{cases} \boldsymbol{x}_i^l(t) + \boldsymbol{\epsilon}_t, & t \in \mathcal{Q}_T, i = i_t, l = 0; \\ \boldsymbol{x}_i^l(t), & \text{else.} \end{cases}$$

In addition, its push-sum weight $y_i^l(t)$ stays unchanged.

**(B)** Then we give a matrix form of model averaging *between any two consecutive events* $t$ and $t + 1$. Specifically, for the transition matrix $\boldsymbol{P}(t)$ between events $t$ and $t + 1$ (which is now defined on the augmented graph $\tilde{G}$), we identify each of its entries $p_{ij}(t)$. Recall that, for each entry $p_{ij}(t)$ of the transition matrix $\boldsymbol{P}(t)$, it specifies how much portion of node $j$'s push-sum weight $y_j^l(t)$ immediately before event $t$ will be added to node $i$'s push-sum weight $y_i^l(t + 1)$ until the next event $t + 1$. Note that, as the averaging operation on model parameters $\boldsymbol{x}_i^l(t)$ and push-sum weight $y_i^l(t)$ are totally the same, the same transition matrix $\boldsymbol{P}(t)$ is also applied to model parameters.

As a prerequisite, we represent the parameters of all nodes in the augmented graph via a compact matrix form. Specifically, for each node $b_i^l \in \tilde{V}$, we view its local model $\boldsymbol{x}_i^l$ as a *row vector* in $\mathbb{R}^{1 \times n}$. Then we can formulate the matrix model $\boldsymbol{X}(t)$ by stacking the local models of all nodes (both actual and virtual) together into a $m(D + 1) \times n$ matrix, namely

$$\boldsymbol{X}(t) = [\boldsymbol{x}_1(t)^\top, \ldots, \boldsymbol{x}_m(t)^\top, \boldsymbol{x}_1^1(t)^\top, \ldots, \boldsymbol{x}_m^1(t)^\top, \ldots, \boldsymbol{x}_1^D(t)^\top, \ldots, \boldsymbol{x}_m^D(t)^\top]^\top \in \mathbb{R}^{m(D+1) \times n}.$$

Similarly, we can denote the compact form of push-sum weights $\boldsymbol{y}(t)$ by stacking the push-sum weights $y_i^l(t)$ of all nodes (both actual and virtual) into a $m(D + 1) \times 1$ *column vector*, i.e.,

$$\boldsymbol{y}(t) = [y_1(t), \ldots, y_m(t), y_1^1(t), \ldots, y_m^1(t), \ldots, y_1^D(t), \ldots, y_m^D(t)]^\top \in \mathbb{R}^{m(D+1)}.$$

Now the parameters of each node can be specified via $\boldsymbol{X}(t)$ and $\boldsymbol{y}(t)$. Specifically, (i) For each actual node $i \in V$, its local model constitutes the $i$-th row of $\boldsymbol{X}(t)$, i.e., $\boldsymbol{x}_i(t) = \boldsymbol{e}_i^\top \boldsymbol{X}(t) = \boldsymbol{X}_{i\cdot}(t)$[1], and its push-sum weight $y_i(t)$ is exactly the $i$-th entry of $\boldsymbol{y}(t)$. (ii) For any virtual node $b_i^l, i \in V, l \in \{1, \ldots, D\}$, its local model constitutes the $(lm + i)$-th row of $\boldsymbol{X}(t)$, i.e., $\boldsymbol{x}_i^l(t) = \boldsymbol{e}_{lm+i}^\top \boldsymbol{X}(t) = \boldsymbol{X}_{(lm+i)\cdot}(t)$, and its push-sum weight $y_i^l(t)$ is the $(lm + i)$-th entry of $\boldsymbol{y}(t)$. Since the index $b_i^l$ is a bit lengthy, we simplify the notations by using an integer $h = lm + i$ to identify this virtual node $b_i^l$. Then its parameters can be rewritten as $\boldsymbol{x}_h(t) = \boldsymbol{e}_h^\top \boldsymbol{X}(t) = \boldsymbol{X}_{h\cdot}(t)$ and $y_h(t) = \boldsymbol{e}_h^\top \boldsymbol{y}(t)$. Note that, now the vertex set of $\tilde{G}$ is equivalent to $\tilde{V} = \{1, \ldots, m(D + 1)\}$, and the virtual nodes are specified in $\tilde{V} - V = \{m + 1, \ldots, m(D + 1)\}$.

---

[1]We use $\boldsymbol{e}_i$ to represent the $i$-th unit vector in $\mathbb{R}^{m(D+1)}$, whose $i$-th entry takes 1 and other entries take 0. We also use $\boldsymbol{A}_{i\cdot}$ to represent the $i$-th row of any matrix $\boldsymbol{A}$.

The transition matrix $\boldsymbol{P}(t)$ now acts on the model parameters as
$$\boldsymbol{y}(t+1) = \boldsymbol{P}(t)\boldsymbol{y}(t).$$
The entries $p_{ij}(t)$ of $\boldsymbol{P}(t)$ formulate the multiplication and the averaging operation on push-sum weights, in the form of $y_i(t+1) = \sum_{j \in \tilde{V}} p_{ij}(t)y_j(t)$. To specify the matrix $\boldsymbol{P}(t)$, first recall that, after a local update event $t \in \mathcal{Q}_T$, learner $i$ will first multiple its model and push-sum weight by a factor of $\gamma_{i_t}$, then sent out copies of its updated parameters (e.g.,the push-sum weight $y'_{i_t}(t) = \gamma_{i_t} y_{i_t}(t)$). In the following, we investigate the two types of events separately.

(i) If event $t$ is a prediction event, i.e., $t \in \mathcal{P}_T$, then $y'_i(t) = y_i(t), \forall i \in \tilde{V}$.

(i-a) For each actual learner $i \in V$, it can only perform model averaging during the interval between events $t$ and $t+1$. In Algorithm 1, the model averaging takes
$$y_i(t+1) = y'_i(t) + \sum_{(\boldsymbol{x}',y') \in \mathcal{M}_i(t)} y' = y_i(t) + \sum_{(\boldsymbol{x}',y') \in \mathcal{M}_i(t)} y',$$
where the set $\mathcal{M}_i(t)$ contains all messages used for model averaging on learner $i$ during the interval between events $t$ and $t+1$. Recall that, in the graph augmentation technique, any message sent to learner $i$ with a delay of $d$ is viewed as transmitting along the chain $b_i^d \to b_i^{d-1} \to \cdots \to b_i^1 \to i$. Consequently, $\mathcal{M}_i(t)$ inherits all messages that are captured by node $b_i^1$ immediately before event $t$, and we simply have
$$y_i(t+1) = y_i(t) + y_i^1(t).$$
(i-b) For each virtual node $h = b_i^l, i \in V, l \in \{1, \ldots, D\}$, it will pass its parameters to the successor $b_i^{l-1}$ in the chain, and may inherit the parameters of the predecessor $b_i^{l+1}$ in the chain (if $b_i^{l+1}$ does exist, namely $l < D$). Therefore, we have
$$y_i^l(t+1) = \begin{cases} y_i^{l+1}(t), & l \in \{1, \ldots, D-1\}, \\ 0, & l = D. \end{cases}$$

Combining (i-a) and (i-b), the transition matrix $\boldsymbol{P}(t)$ after each prediction event $t \in \mathcal{P}_T$ and before the next event can be formulated as[2]
$$\boldsymbol{P}(t) = \begin{bmatrix} I_m & I_m & O_m & \cdots & O_m \\ O_m & O_m & I_m & \cdots & O_m \\ \vdots & \vdots & \vdots & \ddots & \vdots \\ O_m & O_m & O_m & \cdots & I_m \\ O_m & O_m & O_m & \cdots & O_m \end{bmatrix}. \tag{1}$$

(ii) If event $t$ is a local update event, i.e., $t \in \mathcal{Q}_T$, the averaging effect will be more complex, because the updated node $i_t$ will also send out messages.

(ii-a) We first consider actual nodes $i \in V$. For the updated node $i_t$, it will multiply its push-sum weight by $\gamma_{i_t}$ immediately after the projected gradient descent step, namely $y'_{i_t}(t) = \gamma_{i_t} y_{i_t}(t)$. For any other actual node $j \in V - \{i_t\}$, its push-sum weight does not need to multiply $\gamma_j$, it may receive the message sent from learner $i_t$ if it is a neighbor of $i_t$ and the message has zero message delay, i.e., $j \in \mathcal{N}(i_t)$ and $d_{i_t j}^m(t) = 0$. Then, just like the above case (i-a), each actual node $i \in V$ will inherit the parameters from the virtual node $b_i^1$. To give a more compact expression, similar to [31], we introduce an indicator variable $\delta_i^l(t)$ for $i \in V, 0 \le l \le D$, i.e.,
$$\delta_i^l(t) = \begin{cases} 1, & i \in \mathcal{N}(i_t) \cup \{i_t\}, l = d_{i_t i}^m(t), \\ 0, & \text{otherwise.} \end{cases}$$

$\delta_i^l(t)$ equals to 1 if and only if node $i_t$ sends a message to node $i$ and such message exactly has a delay of $l$. Note that, we assume $d_{i_t i_t}^m(t) = 0$, which makes the indicator variable $\delta_i^l(t)$ well-defined. In summary, we have
$$y_i(t+1) = \begin{cases} \gamma_{i_t} y_i(t) + y_i^1(t), & i = i_t, \\ y_i(t) + y_i^1(t) + \delta_i^0(t)\gamma_{i_t} y_{i_t}(t), & i \ne i_t. \end{cases}$$

---

[2]We use $I_m$ to represent the $m \times m$ identity matrix and $O_m$ to represent the $m \times m$ all-zero matrix.

(ii-b) We then consider virtual nodes $h = b_i^l, i \in V, l \in \{1, \ldots, D\}$. If its corresponding actual node $i$ is a neighbor of the updated node $i_t$ and the message sent from node $i_t$ has a delay of $l$, namely $i \in \mathcal{N}(i_t)$ and $d_{i_t i}^m(t) = l$, then this virtual node will receive the model copy of learner $i_t$. Moreover, just like the above case (i-b), it will pass its parameters to the successor $b_i^{l-1}$ in the chain and may inherate the parameters of the predecessor $b_i^{l+1}$ in the chain (if node $b_i^{l+1}$ exists). In summary, we have

$$y_i^l(t+1) = \begin{cases} \delta_i^l(t)\gamma_{i_t} y_{i_t}(t) + y_i^{l+1}(t), & l \in \{1, \ldots, D-1\}, \\ \delta_i^l(t)\gamma_{i_t} y_{i_t}(t), & l = D. \end{cases}$$

Combining (ii-a) and (ii-b), the transition matrix $\boldsymbol{P}(t)$ after each local update event $t \in \mathcal{Q}_T$ takes

$$\boldsymbol{P}(t) = \begin{bmatrix} P^0(t) & I_m & O_m & \cdots & O_m \\ P^1(t) & O_m & I_m & \cdots & O_m \\ \vdots & \vdots & \vdots & \ddots & \vdots \\ P^{D-1}(t) & O_m & O_m & \cdots & I_m \\ P^D(t) & O_m & O_m & \cdots & O_m \end{bmatrix}, \tag{2}$$

where each block $P^l(t) \in \mathbb{R}^{m \times m}$ for $l \in \{0, \ldots, D\}$ takes

$$P_{ij}^l(t) = \begin{cases} 1, & i = j \neq i_t, l = 0, \\ \delta_i^l(t)\gamma_{i_t}, & j = i_t, \\ 0, & \text{else}, \end{cases}$$

for any $i, j \in \{1, \ldots, m\}$.

The transition matrix $\boldsymbol{P}(t)$ derived above has the following properties:

**Lemma 3.** *In AD-OGP, for any $t \in \{1, \ldots, 2T\}$, the transition matrix $\boldsymbol{P}(t)$ defined on the augmented graph satisfies:*
*(i) $\boldsymbol{P}(t)$ is non-negative and column stochastic, i.e., $\mathbf{1}^\top \boldsymbol{P}(t) = \mathbf{1}^\top$.*
*(ii) Its first $m$ diagonal entries $\boldsymbol{P}_{ii}(t), 1 \leq i \leq m$ are positive. In addition, its positive entries are at least $\gamma_{i_t}$ if $t \in \mathcal{Q}_T$, and $1$ if $t \in \mathcal{P}_T$.*

*Proof.* Since this lemma obviously holds for $t \in \mathcal{P}_T$, we only need to check the case of $t \in \mathcal{Q}_T$. From the above formulation in (B), $\boldsymbol{P}(t)$ is non-negative, and its minimal positive entry is exactly $\gamma_{i_t}$. Moreover, From the formulation of $P^0(t)$ and the fact that $\delta_{i_t}^0(t) = 1$, we have $\boldsymbol{P}_{jj}(t) = 1$ for $j \in \{1, \ldots, m\} - \{i_t\}$ and $\boldsymbol{P}_{i_t i_t}(t) = \gamma_{i_t}$, which are all positive.

Now we prove the column stochasticity of $\boldsymbol{P}(t)$. As it trivially holds for the last $mD$ columns, we only need to check the first $m$ columns. For any $j \in \{1, \ldots, m\} - \{i_t\}$, the only non-zero entry in the $j$-th column is $\boldsymbol{P}_{jj}(t) = 1$. For the $i_t$-th column, the entries in this column sum up to $\sum_{l=0}^{D} \sum_{i \in V} \delta_i^l(t)\gamma_{i_t}$. From Assumption 1, the message delay $d_{i_t i}^m(t)$ must take a value within $\{0, \ldots, D_m\}$ for any $i \in \mathcal{N}(i_t) \cup \{i_t\}$, and particularly $d_{i_t i_t}^m(t) = 0$. Hence

$$\sum_{l=0}^{D} \delta_i^l(t) = \begin{cases} 1, & i \in \mathcal{N}(i_t) \cup \{i_t\}, \\ 0, & \text{otherwise}. \end{cases}$$

Recall that $\gamma_{i_t} = 1/(|\mathcal{N}(i_t)| + 1)$. Thus the sum of the entries in its $i_t$-th column is exactly

$$\sum_{l=0}^{D} \sum_{i \in V} \delta_i^l(t)\gamma_{i_t} = \sum_{i \in \mathcal{N}(i_t) \cup \{i_t\}} \gamma_{i_t} = (|\mathcal{N}(i_t)| + 1)\gamma_{i_t} = 1.$$

Hence, we prove that $\boldsymbol{P}(t)$ is column stochastic. $\qquad \square$

## D.2 Derive a General Bound in Terms of Transition Matrices and Gradients

Now that we have formulated the transition matrix at each round, we can derive the compact matrix form of the learning procedure.

We define a matrix $\mathbf{\Delta}(t) \in \mathbb{R}^{m(D+1) \times n}$ at each round $t \in \{1, \ldots, 2T\}$: if $t \in \mathcal{P}_T$, we let $\mathbf{\Delta}(t)$ be the $m(D+1) \times n$ all-zero matrix; if $t \in \mathcal{Q}_T$, we let $\mathbf{\Delta}(t) = \boldsymbol{e}_{i_t} \boldsymbol{\epsilon}_t$, whose $i_t$-th row takes $\boldsymbol{\epsilon}_t$ and the other rows are the all-zero vector. Then for $t \in \mathcal{Q}_T$, the weighted projected gradient step (without multiplication with $\gamma_{i_t}$) takes $\boldsymbol{X}(t) + \mathbf{\Delta}(t)$; and this also holds for $t \in \mathcal{P}_T$ where $\mathbf{\Delta}(t)$ is an all-zero matrix. Hence we can call $\mathbf{\Delta}(t)$ the *incremental matrix* at round $t$. Combining it with the transition matrix $\boldsymbol{P}(t)$ derived before, we can write out the full recursive relations of model parameters or push-sum weights of all learners, namely,

$$
\begin{aligned}
\boldsymbol{X}(t+1) &= \boldsymbol{P}(t)(\boldsymbol{X}(t) + \mathbf{\Delta}(t)), \\
\boldsymbol{y}(t+1) &= \boldsymbol{P}(t)\boldsymbol{y}(t).
\end{aligned} \tag{3}
$$

From the above recursions, we can now express $\boldsymbol{X}(t)$ and $\boldsymbol{y}(t)$ in terms of the initial parameters $\boldsymbol{X}(0), \boldsymbol{Y}(0)$ and the incremental terms $\mathbf{\Delta}(s)$ at previous rounds $s < t$. For conciseness, for any $s < t$, we define the product of the $t - s$ consecutive transition matrices (termed the *multi-hop transition matrix*) as

$$
\boldsymbol{Q}(s,t) = \boldsymbol{P}(t-1) \cdots \boldsymbol{P}(s).
$$

We also set $\boldsymbol{Q}(t,t) = I_{m(D+1)}$ to make the notation $\boldsymbol{Q}(s,t)$ well-defined. Then, it is easy to verify that $\boldsymbol{Q}(s,t)$ is column stochastic for any $s \le t$. Specifically, since all $\boldsymbol{P}(t)$ are column stochastic, we have

$$
\mathbf{1}^\top \boldsymbol{Q}(s,t) = \mathbf{1}^\top \boldsymbol{P}(t-1) \cdots \boldsymbol{P}(s) = \mathbf{1}^\top.
$$

Recall that $\mathbf{\Delta}(t)$ is an all-zero matrix if $t \in \mathcal{P}_T$ and also notice that $\{1, \ldots, t-1\} - \mathcal{P}_T = \{1, \ldots, t-1\} \cap \mathcal{Q}_T = \mathcal{Q}_{1,t}$, hence we have

$$
\begin{aligned}
\boldsymbol{X}(t) &= \boldsymbol{Q}(1,t)\boldsymbol{X}(1) + \sum_{s=1}^{t-1} \boldsymbol{Q}(s,t)\mathbf{\Delta}(s) \\
&= \boldsymbol{Q}(1,t)\boldsymbol{X}(1) + \sum_{s \in \mathcal{Q}_{1,t}} \boldsymbol{Q}(s,t)\mathbf{\Delta}(s), \\
\boldsymbol{y}(t) &= \boldsymbol{Q}(1,t)\boldsymbol{y}(1).
\end{aligned} \tag{4}
$$

Based on the above recursions in the matrix form, we now give a general analysis of the regret. Since the regret is evaluated by the actual predictions made at prediction events, we consider each prediction event $t \in \mathcal{P}_T$. For each learner $i \in V$, since $\boldsymbol{x}_i(t)$ and $y_i(t)$ denote its most recent parameters immediately before event $t$, the actual prediction made by learner $i$ is exactly $\boldsymbol{w}_i(t) = \boldsymbol{x}_i(t)/y_i(t)$. To analyze the regret, we need to compare the loss $f_t(\boldsymbol{w}_j(t))$ evaluated at the reference learner $j$ with the loss $f_t(\boldsymbol{w}^*)$ evaluated by the best fixed prediction $\boldsymbol{w}^*$ in hindsight.

Inlighted by the analysis of synchronous online algorithms [36], we introduce the notion of "average model $\bar{\boldsymbol{x}}(t)$" as a communication-invariant quantity. However, such a quantity in the asynchronous setting is much more complex than that in the synchronous setting due to the existence of message delay. Specifically, in the synchronous setting, $\bar{\boldsymbol{x}}(t)$ can be computed by simply averaging the local models of all actual learners, i.e., $\bar{\boldsymbol{x}}(t) = \sum_{i \in V} \boldsymbol{x}_i(t)/m$. In the asynchronous setting, however, due to the message delay, each message will not be captured by any actual learner during its transmission. Suppose some message is in transmission at some round $t$, then the average model $\sum_{i \in V} \boldsymbol{x}_i(t)/m$ of all actual learners will not cover this message, and consequently, the recursion of the average model will crash. To resolve this issue, recall that, each message is captured by exactly one virtual node in the augmented graph $\tilde{G}$ during its transmission. Therefore, when calculating the average model, the parameters captured by any virtual node should also be taken into account. Consequently, the average model can be defined as

$$
\bar{\boldsymbol{x}}(t) = \frac{1}{m} \sum_{i \in V} \sum_{l=0}^{D} \boldsymbol{x}_i^l(t) = \frac{1}{m} \sum_{h \in \tilde{V}} \boldsymbol{x}_h(t) = \frac{1}{m} \mathbf{1}^\top \boldsymbol{X}(t),
$$

where $\tilde{V}$ is the aforementioned augmented vertex set. Plug the formulation (4) of $\boldsymbol{X}(t)$ into the above equation, and recall that $\boldsymbol{Q}(s,t)$ is column stochastic for all $s \leq t$, then we have

$$\bar{\boldsymbol{x}}(t) = \frac{1}{m}\boldsymbol{1}^\top(\boldsymbol{X}(1) + \sum_{s \in \mathcal{Q}_{1,t}} \boldsymbol{\Delta}(s)).$$

Also recall that, the local model of each node is initialized as $\boldsymbol{x}_i(1) = \boldsymbol{w}_0$ and $\boldsymbol{x}_i^l(1) = \boldsymbol{0}$ for all $i \in V, l \in \{1, \ldots, D\}$. Hence the first $m$ rows of $\boldsymbol{X}(1)$ are $\boldsymbol{w}_0$, and the last $mD$ rows of $\boldsymbol{X}(1)$ are all-zero vectors. Moreover, for any $t \in \mathcal{Q}_T$, the $i_t$-th row of $\boldsymbol{\Delta}(t)$ takes $\boldsymbol{\epsilon}_t = -\eta\boldsymbol{g}_t + \boldsymbol{r}_t$, and the other rows of $\boldsymbol{\Delta}(t)$ are all-zero vectors. Therefore, the average model $\bar{\boldsymbol{x}}(t)$ has a simple form, namely,

$$\bar{\boldsymbol{x}}(t) = \boldsymbol{w}_0 + \frac{1}{m}\sum_{s \in \mathcal{Q}_{1,t}} (\boldsymbol{r}_s - \eta\boldsymbol{g}_s).$$

Similarly, we can define the average push-sum weight $\bar{y}(t)$ at each round $t \in \{1, \ldots, 2T\}$. Specifically, we sum up the push-sum weights $y_i^l(t)$ of all nodes $b_i^l$ for $i \in V, 0 \leq l \leq D$ and divide it by $m$, which turns out to be

$$\bar{y}(t) = \frac{1}{m}\boldsymbol{1}^\top\boldsymbol{y}(t) = \frac{1}{m}\boldsymbol{1}^\top\boldsymbol{Q}(1,t)\boldsymbol{y}(1) = \frac{1}{m}\boldsymbol{1}^\top\boldsymbol{y}(1),$$

where the last step utilizes the column stochasticity of $\boldsymbol{Q}(1,t)$. Moreover, since the push-sum weight is initialized as $y_i(1) = 1$ and $y_i^l(1) = 0$ for $i \in V, l \in \{1, \ldots, D\}$, we have $\bar{y}(t) = 1$ for any $t \in \{1, \ldots, 2T\}$, which means that the average push-sum weight stays unchanged during the learning process. For now, at each event $t \in \{1, \ldots, 2T\}$ (either a prediction event or a local update event), the average model $\bar{\boldsymbol{x}}(t)$ and the average push-sum weight $\bar{y}(t)$ can induce a prediction termed the *system prediction*[3]

$$\bar{\boldsymbol{w}}(t) = \frac{\bar{\boldsymbol{x}}(t)}{\bar{y}(t)},$$

which is equivalent to the average model $\bar{\boldsymbol{x}}(t)$ since $\bar{y}(t) = 1$. Using the formulation (4) of $\boldsymbol{X}(t)$, we can also write out the recursive relation of the system prediction $\bar{\boldsymbol{w}}(t)$ at different events

$$\bar{\boldsymbol{w}}(t+1) = \bar{\boldsymbol{x}}(t+1) = \boldsymbol{w}_0 + \frac{1}{m}\sum_{s \in \mathcal{Q}_{1,t+1}} (\boldsymbol{r}_s - \eta\boldsymbol{g}_s)$$

$$= \bar{\boldsymbol{x}}(t) + \delta_t(-\frac{\eta}{m}\boldsymbol{g}_t + \frac{1}{m}\boldsymbol{r}_t) = \bar{\boldsymbol{w}}(t) + \delta_t(-\frac{\eta}{m}\boldsymbol{g}_t + \frac{1}{m}\boldsymbol{r}_t).$$

Recall that, the indicator variable $\delta_t$ equals to 0 if $t \in \mathcal{P}_T$ and to 1 if $t \in \mathcal{Q}_T$.

We further investigate the distance between the system prediction $\bar{\boldsymbol{w}}(t+1)$ and the best fixed prediction $\boldsymbol{w}^*$ in hindsight. Specifically, for any $t \in \mathcal{P}_T$, we simply have

$$\|\bar{\boldsymbol{w}}(t+1) - \boldsymbol{w}^*\|^2 = \|\bar{\boldsymbol{w}}(t) - \boldsymbol{w}^*\|^2. \tag{5}$$

For any $t \in \mathcal{Q}_T$, the form is slightly more complex

$$\|\bar{\boldsymbol{w}}(t+1) - \boldsymbol{w}^*\|^2 = \|\bar{\boldsymbol{w}}(t) - \boldsymbol{w}^*\|^2 + \frac{1}{m^2}\|\boldsymbol{r}_t - \eta\boldsymbol{g}_t\|^2$$
$$+ \frac{2}{m}\boldsymbol{r}_t^\top(\bar{\boldsymbol{w}}(t) - \boldsymbol{w}^*) - \frac{2\eta}{m}\boldsymbol{g}_t^\top(\bar{\boldsymbol{w}}(t) - \boldsymbol{w}^*). \tag{6}$$

We further investigate the latter case $t \in \mathcal{Q}_T$. Notice that, the recursion (6) is related with the residual $\boldsymbol{r}_t$ of the weighted projection operation. We show that, the norm of the residual is upper bounded by the norm of the associated gradient $\boldsymbol{g}_t$.

**Lemma 4.** *In AD-OGP, the residual $\boldsymbol{r}_t$ of the weighted projection operation at each local update event $t \in \mathcal{Q}_T$ is defined as above. Then its norm is bounded as*

$$\|\boldsymbol{r}_t\| \leq \eta g_t.$$

---

[3]Remind that, in our definition, the system prediction $\bar{\boldsymbol{w}}(t)$ is induced by the average parameters $\bar{\boldsymbol{x}}(t)$ and $\bar{y}(t)$ of all learners at event $t$, which does NOT necessarily equal to the average of the predictions generated by all actual learners at event $t$, i.e., $\bar{\boldsymbol{w}}(t) \neq \sum_{i \in V} \boldsymbol{w}_i(t)/m$.

*Proof.* Recall our definition of weighted projection operation (note that, Lemma 1 ensures that $y > 0$)

$$\Pi_{y\mathcal{K}}(\boldsymbol{x}) = \underset{\boldsymbol{z} \in y\mathcal{K}}{\operatorname{argmin}} \|\boldsymbol{z} - \boldsymbol{x}\| = y \underset{\boldsymbol{w} \in \mathcal{K}}{\operatorname{argmin}} \|\boldsymbol{w} - \frac{\boldsymbol{x}}{y}\|,$$

or equivalently

$$\frac{\Pi_{y\mathcal{K}}(\boldsymbol{x})}{y} = \underset{\boldsymbol{w} \in \mathcal{K}}{\operatorname{argmin}} \|\boldsymbol{w} - \frac{\boldsymbol{x}}{y}\|.$$

Denote learner $i_t$'s local model after gradient descent and *before weighted projection* as $\boldsymbol{z}_{i_t}(t) = \boldsymbol{x}_{i_t}(t) - \eta \boldsymbol{g}_t$, and its induced prediction as $\boldsymbol{v}_{i_t}(t) = \boldsymbol{z}_{i_t}(t)/y_{i_t}(t)$. From Proposition 1 we have $\boldsymbol{x}_{i_t}(t) \in y_{i_t}(t)\mathcal{K}$, and from Lemma 1 we know $y_{i_t}(t) > 0$. Hence

$$\|\boldsymbol{r}_t\| = \|\Pi_{y_{i_t}(t)\mathcal{K}}(\boldsymbol{z}_{i_t}(t)) - \boldsymbol{z}_{i_t}(t)\| = y_{i_t}(t)\|\frac{\Pi_{y_{i_t}(t)\mathcal{K}}(\boldsymbol{z}_{i_t}(t))}{y_{i_t}(t)} - \frac{\boldsymbol{z}_{i_t}(t)}{y_{i_t}(t)}\|$$

$$\leq y_{i_t}(t)\|\frac{\boldsymbol{x}_{i_t}(t)}{y_{i_t}(t)} - \frac{\boldsymbol{z}_{i_t}(t)}{y_{i_t}(t)}\| = \|\boldsymbol{x}_{i_t}(t) - \boldsymbol{z}_{i_t}(t)\| = \|\eta \boldsymbol{g}_t\| = \eta g_t,$$

which proves the lemma. $\qquad\square$

Now we turn back to analyze the recursion (6). Specifically, in the right-hand side of the recursion:
(i) The first term $\|\bar{\boldsymbol{w}}(t) - \boldsymbol{w}^*\|$ is preserved for successive elimination.
(ii) The second term is bounded as

$$\|\boldsymbol{r}_t - \eta \boldsymbol{g}_t\| \leq \|\boldsymbol{r}_t\| + \eta\|\boldsymbol{g}_t\| \leq 2\eta g_t.$$

(iii) For the third term $\boldsymbol{r}_t^\top (\bar{\boldsymbol{w}}(t) - \boldsymbol{w}^*)$, recall the definition of $\boldsymbol{r}_t$ and the property of standard projection operation onto a convex set

$$(\Pi_{\mathcal{K}}(\boldsymbol{a}) - \boldsymbol{a})^\top (\boldsymbol{a} - \boldsymbol{b}) \leq 0, \quad \forall \boldsymbol{a} \in \mathbb{R}^n, \boldsymbol{b} \in \mathcal{K}.$$

Set $\boldsymbol{a} = \boldsymbol{v}_{i_t}(t), \boldsymbol{b} = \boldsymbol{w}^*$ in the above inequality, and utilize $y_{i_t}(t) > 0$, then we have

$$\boldsymbol{r}_t^\top (\boldsymbol{v}_{i_t}(t) - \boldsymbol{w}^*) = y_{i_t}(t)(\Pi_{\mathcal{K}}(\boldsymbol{v}_{i_t}(t)) - \boldsymbol{v}_{i_t}(t))^\top (\boldsymbol{v}_{i_t}(t) - \boldsymbol{w}^*) \leq y_{i_t}(t) \cdot 0 = 0.$$

Therefore, we can bound the third term as

$$\boldsymbol{r}_t^\top (\bar{\boldsymbol{w}}(t) - \boldsymbol{w}^*) = \boldsymbol{r}_t^\top (\bar{\boldsymbol{w}}(t) - \boldsymbol{v}_{i_t}(t)) + \boldsymbol{r}_t^\top (\boldsymbol{v}_{i_t}(t) - \boldsymbol{w}^*)$$

$$\leq \|\boldsymbol{r}_t\|\|\bar{\boldsymbol{w}}(t) - \boldsymbol{v}_{i_t}(t)\| \leq \eta g_t\|\bar{\boldsymbol{w}}(t) - \boldsymbol{v}_{i_t}(t)\|,$$

where the last inequality utilizes Lemma 4.
(iv) For the last term $\boldsymbol{g}_t^\top (\boldsymbol{w}^* - \bar{\boldsymbol{w}}(t))$, recall that $\boldsymbol{g}_t = \nabla f_{l_t}(\boldsymbol{w}_{i_t}(l_t))$ and $f_{l_t}$ is convex, then we have

$$\boldsymbol{g}_t^\top (\boldsymbol{w}^* - \bar{\boldsymbol{w}}(t)) = \boldsymbol{g}_t^\top (\boldsymbol{w}_{i_t}(l_t) - \bar{\boldsymbol{w}}(t)) + \boldsymbol{g}_t^\top (\boldsymbol{w}^* - \boldsymbol{w}_{i_t}(l_t))$$

$$\leq g_t\|\boldsymbol{w}_{i_t}(l_t) - \bar{\boldsymbol{w}}(t)\| + f_{l_t}(\boldsymbol{w}^*) - f_{l_t}(\boldsymbol{w}_{i_t}(l_t))$$

Plug the above terms (i)-(iv) into the recursion (6) and rearrange the inequality, then we obtain

$$f_{l_t}(\boldsymbol{w}_{i_t}(l_t)) - f_{l_t}(\boldsymbol{w}^*) \leq \frac{m}{2\eta}(\|\bar{\boldsymbol{w}}(t) - \boldsymbol{w}^*\|^2 - \|\bar{\boldsymbol{w}}(t+1) - \boldsymbol{w}^*\|^2)$$

$$+ g_t(\|\boldsymbol{w}_{i_t}'(t) - \bar{\boldsymbol{w}}(t)\| + \|\boldsymbol{w}_{i_t}(l_t) - \bar{\boldsymbol{w}}(t)\| + \frac{2\eta}{m}g_t).$$

Moreover, to relate the above inequality to loss evaluated at the reference learner $j$, namely $f_{l_t}(\boldsymbol{w}_j(l_t))$, we again utilize the convexity of $f_{l_t}$ and recall the definition $\hat{\boldsymbol{g}}_t = \nabla f_{l_t}(\boldsymbol{w}_j(l_t))$. Then we have

$$f_{l_t}(\boldsymbol{w}_j(l_t)) - f_{l_t}(\boldsymbol{w}_{i_t}(l_t)) \leq (\hat{\boldsymbol{g}}_t)^\top (\boldsymbol{w}_j(l_t) - \boldsymbol{w}_{i_t}(l_t)) \leq \hat{g}_t\|\boldsymbol{w}_j(l_t) - \boldsymbol{w}_{i_t}(l_t)\|.$$

Summing the above two inequality together, we derive an upper bound for $f_{l_t}(\boldsymbol{w}_j(l_t)) - f_{l_t}(\boldsymbol{w}^*)$. Notably, the upper bound is in terms of $\|\boldsymbol{w}_{i_t}'(t) - \bar{\boldsymbol{w}}(t)\|$, $\|\boldsymbol{w}_{i_t}(l_t) - \bar{\boldsymbol{w}}(t)\|$ and $\|\boldsymbol{w}_j(l_t) - \boldsymbol{w}_{i_t}(l_t)\|$, which capture the distance between some specific pair of predictions and need to be further analyzed. In fact, we can decompose these distance terms as

$$\|\boldsymbol{w}_{i_t}'(t) - \bar{\boldsymbol{w}}(t)\| \leq \|\boldsymbol{w}_{i_t}(t) - \bar{\boldsymbol{w}}(t)\| + \frac{\eta g_t}{y_{i_t}(t)},$$

$$\|\boldsymbol{w}_{i_t}(l_t) - \bar{\boldsymbol{w}}(t)\| \leq \|\boldsymbol{w}_{i_t}(l_t) - \bar{\boldsymbol{w}}(l_t)\| + \|\bar{\boldsymbol{w}}(t) - \bar{\boldsymbol{w}}(l_t)\|,$$

$$\|\boldsymbol{w}_j(l_t) - \boldsymbol{w}_{i_t}(l_t)\| \leq \|\boldsymbol{w}_{i_t}(l_t) - \bar{\boldsymbol{w}}(l_t)\| + \|\boldsymbol{w}_j(l_t) - \bar{\boldsymbol{w}}(l_t)\|.$$

In summary, for each $t \in \mathcal{Q}_t$, the quantity $f_{l_t}(\boldsymbol{w}_j(l_t)) - f_{l_t}(\boldsymbol{w}^*)$ can be bounded as

$$
\begin{aligned}
f_{l_t}(\boldsymbol{w}_j(l_t)) - f_{l_t}(\boldsymbol{w}^*) \leq \; & \frac{m}{2\eta}(\|\bar{\boldsymbol{w}}(t) - \boldsymbol{w}^*\|^2 - \|\bar{\boldsymbol{w}}(t+1) - \boldsymbol{w}^*\|^2) \\
& + g_t(\|\boldsymbol{w}_{i_t}(t) - \bar{\boldsymbol{w}}(t)\| + \|\boldsymbol{w}_{i_t}(l_t) - \bar{\boldsymbol{w}}(l_t)\|) \\
& + g_t(\|\bar{\boldsymbol{w}}(t) - \bar{\boldsymbol{w}}(l_t)\| + (\frac{2}{m} + \frac{1}{y_{i_t}(t)})\eta g_t) \\
& + \hat{g}_t(\|\boldsymbol{w}_{i_t}(l_t) - \bar{\boldsymbol{w}}(l_t)\| + \|\boldsymbol{w}_j(l_t) - \bar{\boldsymbol{w}}(l_t)\|).
\end{aligned}
\tag{7}
$$

To further analyze the above quantity, we are particularly interested in $\|\bar{\boldsymbol{w}}(s) - \bar{\boldsymbol{w}}(t)\|$ and $\|\boldsymbol{w}_i(t) - \bar{\boldsymbol{w}}(t)\|$ for any $i \in V, 0 \leq s \leq t$. In the following, we analyze these terms, respectively.

We first derive a bound for $\|\bar{\boldsymbol{w}}(s) - \bar{\boldsymbol{w}}(t)\|$ as follows. Recall that we use $\mathcal{Q}_{s,t} = \mathcal{Q}_T \cap \{s+1, \ldots, t-1\}$ to denote all update events that happen within $(s, t)$.

**Lemma 5.** *In AD-OGP, the average prediction $\bar{\boldsymbol{w}}(t)$ is defined above. Then for any $1 \leq s \leq t \leq 2T$,*

$$
\|\bar{\boldsymbol{w}}(s) - \bar{\boldsymbol{w}}(t)\| \leq \frac{2\eta}{m} \sum_{k \in \mathcal{Q}_{s-1,t}} g_k.
$$

*Proof.* Recall the recursion (6) of the system predictions, i.e.,

$$
\bar{\boldsymbol{w}}(t+1) = \bar{\boldsymbol{w}}(t) + \delta_t(-\frac{\eta}{m}\boldsymbol{g}_t + \frac{1}{m}\boldsymbol{r}_t).
$$

We then have

$$
\|\bar{\boldsymbol{w}}(s) - \bar{\boldsymbol{w}}(t)\| \leq \sum_{k=s}^{t-1} \delta_k(\frac{\eta}{m}\|\boldsymbol{g}_t\| + \frac{1}{m}\|\boldsymbol{r}_t\|) \leq \sum_{k \in \mathcal{Q}_{s-1,t}} (\frac{\eta}{m}\|\boldsymbol{g}_k\| + \frac{1}{m}\|\boldsymbol{r}_k\|).
$$

From Lemma 4 we also know that $\|\boldsymbol{r}_t\| \leq \eta g_t$. Hence we prove the lemma. $\square$

We then analyze the term $\|\boldsymbol{w}_i(t) - \bar{\boldsymbol{w}}(t)\|$, which is a bit more complex. In fact, this term measures the distance between the prediction of the specific learner $i$ and the system prediction, which is thus affected by both local update and communication. The following lemma shows that such term has a bound in terms of the incremental term $\boldsymbol{\epsilon}_s$ at local weight updates as well as the multiple-hop transition matrices $\boldsymbol{Q}(s,t)$.

**Lemma 6.** *In AD-OGP, the average prediction $\bar{\boldsymbol{w}}(t)$, the residual of projection $\boldsymbol{r}_t$ and the multi-hop transition matrix $\boldsymbol{Q}(s,t)$ are defined above. Then for any $t \in \{1, \ldots, 2T\}$, it holds that*

$$
\|\boldsymbol{w}_i(t) - \bar{\boldsymbol{w}}(t)\| \leq \sum_{s \in \mathcal{Q}_{1,t}} R_i(s,t),
$$

*where the quantity $R_i(s,t)$ is defined as[4]*

$$
R_i(s,t) = \|\frac{\boldsymbol{e}_i^\top \boldsymbol{Q}(s,t)\boldsymbol{\Delta}(s)}{\boldsymbol{e}_i^\top \boldsymbol{Q}(s,t)\boldsymbol{y}(s)} - \frac{\boldsymbol{\epsilon}_s}{m}\|.
$$

*Proof.* Recall that, for any actual learner $i \in V$, its prediction $\boldsymbol{w}_i(t)$ at round $t$ is determined by $\boldsymbol{x}_i(t)/y_i(t) = \boldsymbol{e}_i^\top \boldsymbol{X}(t)/\boldsymbol{e}_i^\top \boldsymbol{y}(t)$. From the above formulations (4) of $\boldsymbol{X}(t)$ and $\boldsymbol{y}(t)$, the prediction can be expressed in terms of the initial local model $\boldsymbol{X}(1)$ and the update matrix $\boldsymbol{\Delta}(s)$, namely,

$$
\boldsymbol{w}_i(t) = \frac{\boldsymbol{e}_i^\top \boldsymbol{X}(t)}{\boldsymbol{e}_i^\top \boldsymbol{y}(t)} = \frac{\boldsymbol{e}_i^\top \boldsymbol{Q}(1,t)\boldsymbol{X}(1)}{\boldsymbol{e}_i^\top \boldsymbol{Q}(1,t)\boldsymbol{y}(1)} + \sum_{s \in \mathcal{Q}_{1,t}} \frac{\boldsymbol{e}_i^\top \boldsymbol{Q}(s,t)\boldsymbol{\Delta}(s)}{\boldsymbol{e}_i^\top \boldsymbol{Q}(s,t)\boldsymbol{y}(s)}.
$$

Since all actual learners are initialized as $\boldsymbol{x}_i(1) = \boldsymbol{w}_0, y_i(1) = 1, i \in V$, and all virtual learners are initialized as $\boldsymbol{x}_i^l(1) = \boldsymbol{0}, y_i^l(1) = 0, i \in V, l \in \{1, \ldots, D\}$, we can verify that (here we view $\boldsymbol{w}_0 \in \mathbb{R}^{1 \times n}$ as a row vector)

$$
\boldsymbol{X}(1) = \begin{bmatrix} \boldsymbol{1}_m \boldsymbol{w}_0 \\ O_{mD \times n} \end{bmatrix} = \begin{bmatrix} \boldsymbol{1}_m \\ \boldsymbol{0}_{mD} \end{bmatrix} \boldsymbol{w}_0 = \boldsymbol{y}(1)\boldsymbol{w}_0.
$$

---

[4]Recall that, $\boldsymbol{e}_i$ denotes the $i$-th unit vector in $\mathbb{R}^{m(D+1)}$; for any $t \in \mathcal{Q}_T$, we have also denoted the update matrix at round $t$ as $\boldsymbol{\Delta}(s) = \boldsymbol{e}_{i_s}\boldsymbol{\epsilon}_t$ and the update term at round $t$ as $\boldsymbol{\epsilon}_t = \boldsymbol{r}_t - \eta\boldsymbol{g}_t$.

Since $e_i^\top Q(1,t)y(1)$ is a scalar, we directly have

$$\frac{e_i^\top Q(1,t)X(1)}{e_i^\top Q(1,t)y(1)} = \frac{(e_i^\top Q(1,t)y(1))w_0}{e_i^\top Q(1,t)y(1)} = w_0,$$

and thus

$$w_i(t) = w_0 + \sum_{s \in \mathcal{Q}_{1,t}} \frac{e_i^\top Q(s,t)\Delta(s)}{e_i^\top Q(s,t)y(s)}.$$

A similar deduction on $\bar{w}(t)$ gives

$$\bar{w}(t) = \frac{1}{m}\mathbf{1}^\top X(t) = w_0 + \sum_{s \in \mathcal{Q}_{1,t}} \frac{\epsilon_s}{m}.$$

Plugging the above expressions of $w_i(t)$ and $\bar{w}(t)$ into the term $\|w_i(t) - \bar{w}(t)\|$, we derive

$$\|w_i(t) - \bar{w}(t)\| \le \sum_{s \in \mathcal{Q}_{1,t}} R_i(s,t),$$

which proves the lemma. $\qquad\square$

Utilizing the above Lemma 5 and Lemma 6, we can derive a general bound in terms of $R_i(s,t)$ (which is a function of transition matrix $Q(s,t)$ and update term $\epsilon_t$) and the associated gradients $g_t, \hat{g}_t$. Specifically, for any $t \in \mathcal{Q}_T$, we can apply both lemmas to inequality (7), which derives

$$
\begin{aligned}
f_{l_t}(w_j(l_t)) - f_{l_t}(w^*) \le{}& \frac{m}{2\eta}(\|\bar{w}(t) - w^*\|^2 - \|\bar{w}(t+1) - w^*\|^2) \\
&+ \sum_{s \in \mathcal{Q}_{1,t}} g_t R_{i_t}(s,t) + \sum_{s \in \mathcal{Q}_{1,l_t}} g_t R_{i_t}(s,l_t) \\
&+ \frac{2\eta}{m}g_t\Big(g_t + \sum_{s \in \mathcal{Q}_{l_t-1,t}} g_s\Big) + \frac{1}{y_{i_t}(t)}\eta g_t^2 \\
&+ \sum_{s \in \mathcal{Q}_{1,l_t}} \hat{g}_t(R_{i_t}(s,l_t) + R_j(s,l_t)).
\end{aligned}
\tag{8}
$$

Recall the definition $\mathcal{Q}_{s,t} = \{k \mid s < k < t, \delta_k = 1\}$. Since event $l_t$ is a prediction event ($\delta_{l_t} = 0$) and event $t$ is a local update event ($\delta_t = 1$), we have $\mathcal{Q}_{l_t-1,t} \cup \{t\} = \mathcal{Q}_{l_t,t+1}$. Hence $g_t + \sum_{s \in \mathcal{Q}_{l_t-1,t}} g_s = \sum_{s \in \mathcal{Q}_{l_t,t+1}} g_s$.

Moreover, for any $t \in \mathcal{P}_T$, from the equation (5), we trivially have

$$0 = \frac{m}{2\eta}(\|\bar{w}(t) - w^*\|^2 - \|\bar{w}(t+1) - w^*\|^2). \tag{9}$$

We sum the above inequality (8) and equation (9) over $t \in \{1, \ldots, 2T\} = \mathcal{P}_T \cup \mathcal{Q}_T$. The left-hand side of the inequalities sum up to $\sum_{t \in \mathcal{Q}_T} f_{l_t}(w_j(l_t) - f_{l_t}(w^*))$. Consider the index set $\{l_t \mid t \in \mathcal{Q}_T\}$. Recall that, in our event indexing system, for any local update event $t \in \mathcal{Q}_T$, $l_t$ is defined as the index of its corresponding prediction event. Therefore, the index set $\{l_t \mid t \in \mathcal{Q}_T\}$ is exactly a *permutation* of the indices of the prediction events $\mathcal{P}_T$. Therefore, the left-hand side has an equivalent expression, namely,

$$\sum_{t \in \mathcal{Q}_T} f_{l_t}(w_j(l_t) - f_{l_t}(w^*)) = \sum_{t \in \mathcal{P}_T} f_t(w_j(t) - f_t(w^*)),$$

which is exactly the definition of our regret.

Therefore, to analyze the regret, it suffices to bound the right-hand side of the summation of the inequalities. From $\bar{w}(1) = w_0$, the first terms in the right-hand side of the inequalities ($t \in \mathcal{Q}_T$) and equations ($t \in \mathcal{P}_T$) sum up to

$$\frac{m}{2\eta}(\|\bar{w}(1) - w^*\|^2 - \|\bar{w}(2T+1) - w^*)\|^2 \le \frac{m}{2\eta}\|w_0 - w^*\|^2.$$

Now we can derive a general bound, i.e.,

$$
\begin{aligned}
\textbf{Regret}_j \leq{} & \frac{m}{2\eta}\|\boldsymbol{w}_0 - \boldsymbol{w}^*\|^2 + \frac{2\eta}{m}\sum_{t\in\mathcal{Q}_T}\sum_{s\in\mathcal{Q}_{l_t,t+1}} g_s g_t + \frac{1}{y_{i_t}(t)}\eta g_t^2 \\
& + \sum_{t\in\mathcal{Q}_T}\sum_{s\in\mathcal{Q}_{1,t}} g_t R_{i_t}(s,t) + \sum_{t\in\mathcal{Q}_T}\sum_{s\in\mathcal{Q}_{1,l_t}} (g_t R_{i_t}(s,l_t) + \hat{g}_t R_{i_t}(s,l_t) + \hat{g}_t R_j(s,l_t)),
\end{aligned}
\tag{10}
$$

where the quantity $R_i(s,t)$ for any $i\in V, s\in\mathcal{Q}_T, t\in\{s, s+1, \ldots, 2T\}$ is defined as

$$
R_i(s,t) = \|\frac{\boldsymbol{e}_i^\top \boldsymbol{Q}(s,t)\boldsymbol{\Delta}(s)}{\boldsymbol{e}_i^\top \boldsymbol{Q}(s,t)\boldsymbol{y}(s)} - \frac{\boldsymbol{\epsilon}_s}{m}\|.
$$

### D.3  Analyze the Effect of Push-Sum in Asynchronous Decentralized Online Learning

The final step of our analysis is to derive an upper bound of the quantity $R_i(s,t)$, which specialize the regret to our proposed algorithm. Recall that, the multi-hop transition matrix $\boldsymbol{Q}(s,t)$ characterizes the overall effect of push-sum among all nodes (actual or virtual) during the interval between events $s$ and $t$. Also notice that $\boldsymbol{\epsilon}_s/m = \mathbf{1}^\top\boldsymbol{\Delta}(s)/\mathbf{1}^\top\boldsymbol{y}(s)$ for all $s\leq 2T$, since $\mathbf{1}^\top\boldsymbol{y}(s) = \mathbf{1}^\top\boldsymbol{y}(1) = m$.

Now the physical meaning of the quantity $R_i(s,t)$ can be interpreted as below. We imagine a $(t-s)$-step *distributed averaging* process operated on the augmented graph $\tilde{G}$ (notably, the nodes communicate with each other at each step $k\in\{1,\ldots,t-s\}$, but they never perform local update). At the beginning of the averaging process, the local model $\boldsymbol{x}_h$ and push-sum weight $y_h$ of each node $h\in\tilde{V} = \{1,\ldots,m(D+1)\}$ are initialized to be the $h$-th row of $\boldsymbol{\Delta}(s)$ and the $h$-th entry of $\boldsymbol{y}(s)$, respectively. At each step $k = 1,\ldots,t-s$, the nodes mix their model parameters according to the transition matrix $\boldsymbol{P}(s+k-1)$. Consequently, $\boldsymbol{e}_i^\top\boldsymbol{Q}(s,t)\boldsymbol{\Delta}(s)$ and $\boldsymbol{e}_i^\top\boldsymbol{Q}(s,t)\boldsymbol{y}(s)$ are exactly node $i$'s local model $\boldsymbol{x}_i$ and push-sum weight $y_i$ after the whole imaginary averaging process, respectively. Moreover, $\mathbf{1}^\top\boldsymbol{\Delta}(s)$ and $\mathbf{1}^\top\boldsymbol{y}(s)$ are exactly the average parameters of all nodes in $\tilde{V}$, which actually stay unchanged during the averaging process as $\boldsymbol{Q}(s,t)$ is column stochastic. Therefore, the quantity $R_i(s,t)$ measures the distance between the prediction of node $i$ and the system prediction induced by the average parameters after the $(t-s)$-step averaging process.

From the above analysis, we know that the quantity $R_i(s,t)$ is determined by the initial states of nodes $h\in\tilde{V}$ and the multi-hop communication matrix $\boldsymbol{Q}(s,t)$. Therefore, to give a bound for $R_i(s,t)$, we need to investigate the property of multi-hop transition matrix $\boldsymbol{Q}(s,t)$. Our analysis is enlightened by prior work for asynchronous stochastic optimization [31].

By its definition, each entry $\boldsymbol{Q}_{ij}(s,t)$ of the transition matrix measures the portion of $j$'s initial model $\boldsymbol{u}_j$ attributing to $i$'s model after $t-s$ push-sum steps. If this entry is positive, namely $\boldsymbol{Q}_{ij}(s,t) > 0$, it then implies that node $i$ is able to receive information from node $j$ during the push-sum steps. In intuition, after sufficient number of communication steps, each actual node $i\in V$ can access the information on any other node (either actual or virtual) $h\in\tilde{V}$ via communication; alternatively speaking, when $t-s$ is sufficiently large, $\boldsymbol{Q}_{ih}(s,t)$ is expected to be positive for all $i\in V, l\in\tilde{V}$.

The following lemma gives a lower bound of $t-s$, which guarantees that the entries in first $m$ row of $\boldsymbol{Q}(s,t)$ are all positive. Note that, [31] also derives a similar result in their Lemma 7, namely $t-s \geq m(D^{msg} + \Gamma_d)$. In our work, via a finer-grained analysis, we only requires $t-s \geq (\mathcal{D}+1)(D^{msg} + \Gamma_d)$ where $\mathcal{D}$ denote the diameter of the original graph $G$. In other words, our bound is much smaller than the bound in [31], since the diameter $\mathcal{D}$ of a decentralized architecture is usually much smaller than the number of learners $m$. For example, the diameter of any complete graph is only 1.

**Lemma 7.** *Denote $\Gamma = D^{msg} + \Gamma_d$. In AD-OGP, for any $s\geq 1$, the entries in the first $m$ rows of $\boldsymbol{Q}(s, s+(\mathcal{D}+1)\Gamma)$ are positive. In addition, any positive entry of $\boldsymbol{Q}(s, s+(\mathcal{D}+1)\Gamma)$ is at least*

$$
\alpha = (\frac{1}{m})^{(\mathcal{D}+1)\Gamma}.
$$

*Proof.* Recall that, in our derived forms (8) and (9) of the transition matrix $\boldsymbol{P}(t)$ at each push-sum step, the first $m\times m$ block of $\boldsymbol{P}(t)$ for any event $t\in\{1,\ldots,2T\}$ has all-positive diagonals.

Therefore, for any multi-hop transition matrix $\boldsymbol{Q}(p, q)$ $(p \leq q)$, if its $(i, h)$-th entry located at the first $m$ rows (i.e., $i \leq m$) is positive, then the corresponding $(i, h)$-th entry of $\boldsymbol{Q}(p, q + 1)$ is exactly

$$\boldsymbol{Q}_{ih}(p, q + 1) = \sum_{j=1}^{m(D+1)} \boldsymbol{P}_{ij}(q) \boldsymbol{Q}_{jh}(p, q) \geq P_{ii}^0(q + 1) \boldsymbol{Q}_{ih}(p, q),$$

which must be positive as well. This suggests that the positiveness of the entries in the first $m$ row of the multi-hop transition matrix will be preserved as the push-sum process proceeds on. Therefore, to prove the positiveness of the first $m$ rows of $\boldsymbol{Q}(s, s + (\mathcal{D} + 1)\Gamma)$, it suffices to prove that, for any $i \leq m, h \leq m(D + 1)$, there exists some intermediate step $t \in [s, s + (\mathcal{D} + 1)\Gamma]$ such that the $(i, h)$-th entry of $\boldsymbol{Q}(s, t)$ is positive. Equivalently, we only need to prove that, the information on any node $h \in \tilde{V}$ is able to reach all actual nodes $i \in V$ within at most $(\mathcal{D} + 1)\Gamma$ steps.

To this end, we consider an arbitrary node $h = b_j^l \in \tilde{V}$ for any $j \in V, 0 \leq l \leq D$. The information on node $h$ will reach its corresponding actual node $j$ within $l$ steps, i.e., along the path $b_j^l \to b_j^{l-1} \to \cdots \to j$. In addition, since the underlying decentralized network $G$ is connected, there must exist some path linking $j$ and $i$ in $G$, and the path length $r$ is no more than the diameter $\mathcal{D}$ of the graph. We assume the path to be $j \to u_1 \to \cdots \to u_{r-1} \to i$ where $u_k \in V, k \in \{1, \ldots, r - 1\}$ are the intermediate nodes; also denote the endpoints of the path $u_0 = j, u_r = i$.

Consider each segment $(u_k, u_{k+1})$ of the path for any $k \in \{0, \ldots, r - 1\}$. From Assumption 1 and the graph augmentation technique, we know that: (i) the node $u_k$ will perform local update at least once every $\Gamma_d$ steps, after which it will send out a message to its neighbor $u_{k+1}$; (ii) such a message will take at most $D$ steps to reach its destination $u_{k+1}$. Therefore, starting from any step, it takes at most $\Gamma = D + \Gamma_d$ event steps for the information on $u_k$ to diffuse to $u_{k+1}$. Since such threshold $\Gamma$ holds for all segments $(u_k, u_{k+1})$, the information on $j$ will reach $i$ within at most $\mathcal{D}\Gamma$ steps. In summary, for an arbitrary node $h \in \tilde{V}$ in the augmented graph, it takes at most $D + \mathcal{D}\Gamma \leq (\mathcal{D} + 1)\Gamma$ communication steps in AD-OGP for its information to diffuse to any actual node $i \in V$. Therefore, all entries in the first $m$ row of $\boldsymbol{Q}(s, s + (\mathcal{D} + 1)\Gamma)$ are positive.

In addition, from the formulation (8) or (9) of $\boldsymbol{P}(t)$ for any $t \in \{1, \ldots, 2T\}$, its positive entries are at least $\lambda_{i_t} \geq 1/m$. Therefore, since $\boldsymbol{Q}(s, s + (\mathcal{D} + 1)\Gamma) = \boldsymbol{P}(s + (\mathcal{D} + 1)\Gamma - 1) \cdots \boldsymbol{P}(s)$, the positive entries of $\boldsymbol{Q}(s, s + (\mathcal{D} + 1)\Gamma)$ are at least $\prod_{k=0}^{(\mathcal{D}+1)\Gamma - 1} \lambda_{i_{s+k}} \geq (1/m)^{(\mathcal{D}+1)\Gamma}$. $\qquad \square$

The above property of multi-hop transition matrices also implies the range of the push-sum weights $\boldsymbol{y}(t)$. In fact, the following lemma gives an upper and a lower bound on the push-sum weights $y_h(t)$ of all nodes $h \in \tilde{V}$ throughout the learning process. Note that, [31] also investigates the range of the push-sum weight $y_h(t)$ in their Lemma 11, where the lower bound of the weight on the virtual node is given as $m\alpha^2$. Compared to their bound, we give a lower bound $\alpha$ by looking into the physical communication process; it is much tighter than their bound $m\alpha^2$, as $\alpha = (1/m)^{(\mathcal{D}+1)\Gamma}$ is usually much smaller than $1/m$.

**Lemma 8.** *In AD-OGP, for any $t \in \{1, \ldots, 2T\}$ and $i \in V$, it holds that*

$$m\alpha \leq y_i(t) \leq m.$$

*Moreover, for any $l \in \{1, \ldots, D\}$, we have either $y_i^l(t) = 0$ or*

$$\alpha \leq y_i^l(t) \leq m.$$

*Proof.* Recall our initialization of the push-sum weight on all nodes, namely $y_i(1) = 1$ and $y_i^l(1) = 0$ for all $i \in V, l \in \{1, \ldots, D\}$. Therefore, the stacked vector $\boldsymbol{y}(t)$ (which is constructed by stacking the parameter $y$ of all nodes in the augmented graph at step $t$ together, as we have defined before) can be expressed as

$$\boldsymbol{y}(t) = \boldsymbol{Q}(1, t)\boldsymbol{y}(1) = \boldsymbol{Q}(1, t) \begin{bmatrix} \mathbf{1}_m \\ \mathbf{0}_{mD} \end{bmatrix}.$$

From Lemma 3, for any $t > 0, i \in \{1, \ldots, m\}$, $\boldsymbol{P}_{ii}(t) > 0$, and $\boldsymbol{P}(t)$ is non-negative. Since $\boldsymbol{y}(1)$ is non-negative, $\boldsymbol{y}(t) = \boldsymbol{Q}(1, t)\boldsymbol{y}(1)$ is also non-negative. Moreover, since $\boldsymbol{P}(t)$ is column stochastic, i.e., $\mathbf{1}^\top \boldsymbol{P}(t) = \mathbf{1}^\top$, the sum of the parameters $y_h(t)$ of all nodes $h \in \tilde{V}$ at any step $t$ is exactly

$$\sum_{h \in \tilde{V}} y_h(t) = \mathbf{1}^\top \boldsymbol{y}(t) = \mathbf{1}^\top \boldsymbol{Q}(1, t)\boldsymbol{y}(1) = \mathbf{1}^\top \boldsymbol{y}(0) = m.$$

Therefore, the parameter $y_i^l(t)$ for any node $b_i^l \in \tilde{V}$ (recall that, we use $b_i^0$ to represent the actual node $i \in V$) is at most $m$, which gives the common upper bound $m$ for the push-sum weight $y_h(t)$ throughout the learning process.

To analyze the lower bound, notice that, for any actual node $i \in V$, we particularly have

$$y_i(t) = \boldsymbol{e}_i^\top \prod_{s=0}^{t-1} \boldsymbol{P}(t-s)\boldsymbol{y}(1) \geq \prod_{s=0}^{t-1} \boldsymbol{P}_{ii}(s) \cdot y_i(1) > 0.$$

To further investigate the range of the push-sum weight $y_h(t)$, we consider the following two cases.

(i) When $t < (\mathcal{D}+1)\Gamma$, the positive entries of $\boldsymbol{Q}(1,t)$ are at least $(1/m)^t \geq m\alpha$. Therefore, the positive entries of $\boldsymbol{y}(t)$ are at least $m\alpha$.

(ii) When $t \geq (\mathcal{D}+1)\Gamma$, we can split the multi-hop transition matrix into $\boldsymbol{Q}(1,t) = \boldsymbol{Q}(t-s,t)\boldsymbol{Q}(1,t-s)$ where $s = (\mathcal{D}+1)\Gamma$. From Lemma 7, the first $m$ rows of $\boldsymbol{Q}(t-s,t)$ are positive, and its positive entries are at least $\alpha$. Recall that $\boldsymbol{Q}(1,t-s)$ is column stochastic, i.e., $\sum_{j=1}^{m(D+1)} \boldsymbol{Q}_{jh}(1,t-s) = 1$ for any $h$. Therefore, for any $i \in V, h \in \tilde{V}$, $\boldsymbol{Q}_{ih}(1,t) = \sum_{j=1}^{m(D+1)} \boldsymbol{Q}_{jh}(1,t-s)\boldsymbol{Q}_{ij}(t-s,t)$ is a convex combination of entries in the $i$-th row of $\boldsymbol{Q}(t-s,t)$. Since all these entries are no less than $\alpha$ from Lemma 7, $\boldsymbol{Q}_{ih}(1,t)$ is also at least $\alpha$. From the arbitrariness of $i \in V$ and $h \in \tilde{V}$, the entries in the first $m$ rows of $\boldsymbol{Q}(1,t)$ are at least $\alpha$. Therefore, for any actual node $i \in V$, we further have

$$y_i(t) \geq \sum_{j=1}^m \boldsymbol{Q}_{ij}(1,t)y_j(1) \geq m\alpha.$$

As for any virtual node $b_i^l, i \in V, l \in \{1,\ldots,D\}$, we assume its parameter $y_i^l(t)$ is positive at some step $t$. Recall the transmission of message on the augment graph $\tilde{G}$ (you can also see the demonstration in Figure 8). Specifically, for any virtual node $h = b_i^l \in \tilde{V} - V$, we have $y_i^l(t) > 0$ only when this node is carrying at least one delayed message that is sent to node $i$. We suppose some specific message currently carried by node $b_i^l$ was sent by node $j \in V$ at some previous step $s$. From our communication strategy, the weight parameter captured in such message is exactly a $\gamma_j$-fraction of node $j$'s parameter $y_j(s)$. From the former case (i), the weight parameters carried by other messages are all positive. As we have also proved that $y_j(s) \geq m\alpha$, we directly have

$$y_i^l(t) \geq \gamma_j y_j(s) \geq \gamma_j m\alpha \geq \alpha.$$

Combining the above derived bounds, we successfully prove the lemma. $\qquad \square$

From Lemma 8, we know that the parameter $y_i^l$ of any virtual node $b_i^l$ is either zero, or restricted to the interval $[\alpha, m]$. The following lemma explores the first case: when $y_i^l(t) = 0$, its corresponding local model $\boldsymbol{x}_i^l(t)$ must also be zero.

**Lemma 9.** *For any virtual node $b_i^l, i \in V, l \in \{1,\ldots,D\}$, if its push-sum weight $y_i^l(t) = 0$ at some step $t \in \{1,\ldots,2T\}$, then its local model parameters $\boldsymbol{x}_i^l(t) = \boldsymbol{0}$.*

*Proof.* In intuition, $y_i^l(t) = 0$ implies that the virtual node $b_i^l$ is carrying no delayed message at step $t$. Since parameters $\boldsymbol{x}$ and $y$ are coupled in each message, node $b_i^l$ must not be carrying any local model as well, which means that its local model parameters $\boldsymbol{x}_i^l(t)$ is also zero.

In the following, we provide a strict proof of this lemma by inducting on $t$. The lemma obviously holds for $t = 1$, since the parameters of virtual nodes are initialized as $\boldsymbol{x}_i^l(1) = \boldsymbol{0}$ and $y_i^l(1) = 0$.

We suppose the lemma holds for some $t \geq 0$. From Lemma 8, the first $m$ entries of $\boldsymbol{y}(t)$ are all positive. Therefore, for any $h \in \tilde{V}$, $y_h(t) = 0$ implies that the $h$-th row in $\boldsymbol{X}(t)$ equals to $\boldsymbol{0}$.[5] Denote $\boldsymbol{X}'(t) = \boldsymbol{X}(t) + \boldsymbol{\Delta}(t)$. From the definition of $\boldsymbol{\Delta}(t)$, the entries in its last $mD$ rows all equal to zero. Therefore, $y_h(t) = 0$ also implies that the $h$-th row of $\boldsymbol{X}'(t)$ equals to $\boldsymbol{0}$. Recall the recursion

$$\boldsymbol{X}(t+1) = \boldsymbol{P}(t)\boldsymbol{X}'(t), \quad \boldsymbol{y}(t+1) = \boldsymbol{P}(t)\boldsymbol{y}(t).$$

---

[5]Recall that, we slightly abuse the notations of any virtual node $h = b_i^l \in \tilde{V} - V$. Notably, its push-sum weight at step $t$ can be represented by either $y_i^l(t)$ or $y_h(t)$ (here we mean $h = lm + i$), which refers to the $h$-th entry of $\boldsymbol{y}(t)$.

Suppose $y_h(t+1) = 0$ for an arbitrary $h \in \tilde{V} = \{1, \ldots, m(D+1)\}$, namely,

$$y_h(t+1) = \sum_{k=1}^{m(D+1)} \boldsymbol{P}_{hk}(t) y_k(t) = 0.$$

Since the entries of $\boldsymbol{P}(t)$ and $\boldsymbol{y}(t)$ are all non-negative, for any $k \in \tilde{V}$ such that $y_k(t) > 0$, we must have $\boldsymbol{P}_{hk}(t) = 0$. For simplicity, we use $\boldsymbol{A}_{h\cdot}$ to denote the $h$-th row of any given matrix $\boldsymbol{A}$. Therefore, the $h$-th row of $\boldsymbol{X}(t+1)$ can be expressed as

$$\boldsymbol{X}_{h\cdot}(t+1) = \sum_{k=1}^{m(D+1)} \boldsymbol{P}_{hk}(t) \boldsymbol{X}_{k\cdot}(t) = \sum_{k:y_k(t)=0} \boldsymbol{P}_{hk}(t) \boldsymbol{X}_{k\cdot}(t).$$

Since $y_k(t) = 0$ leads to $\boldsymbol{X}_{k\cdot}(t) = \boldsymbol{0}$, we directly have $\boldsymbol{X}_{h\cdot}(t+1) = \boldsymbol{0}$. Hence the lemma also holds at the next step $t+1$. $\square$

With the above preparations, we can now begin to analyze the quantity $R_i(s,t)$ in Lemma 6, which is a crucial term for regret analysis. Recall that $R_i(s,t)$ is defined as

$$R_i(s,t) = \left\| \frac{\boldsymbol{e}_i^\top \boldsymbol{Q}(s,t) \boldsymbol{\Delta}(s)}{\boldsymbol{e}_i^\top \boldsymbol{Q}(s,t) \boldsymbol{y}(s)} - \frac{\boldsymbol{\epsilon}_s}{m} \right\|.$$

As discussed before, we can imagine a $(t-s)$-step distributed averaging process operated on the augmented graph $\tilde{G}$. In the averaging process, the parameters of all nodes are initialized as $\tilde{\boldsymbol{X}}(1) = \boldsymbol{\Delta}(s)$ and $\tilde{\boldsymbol{y}}(1) = \boldsymbol{y}(s)$. The initial predictions are $\tilde{\boldsymbol{w}}(1)$ such that $\tilde{\boldsymbol{w}}_i(1) = \tilde{\boldsymbol{x}}_i(1)/\tilde{y}_i(1)$. At each step $k = 1, \ldots, \tau$ (here $\tau = t - s$), each node mixes its parameters with other nodes according to the transition matrix $\boldsymbol{P}(s+k-1)$, i.e.,

$$\tilde{\boldsymbol{X}}(k+1) = \boldsymbol{P}(s+k-1)\tilde{\boldsymbol{X}}(k),$$
$$\tilde{\boldsymbol{y}}(k+1) = \boldsymbol{P}(s+k-1)\tilde{\boldsymbol{y}}(k).$$

At the end of the process, the parameters of node $i$ are exactly $\tilde{\boldsymbol{x}}_i(\tau+1) = \boldsymbol{e}_i^\top \boldsymbol{Q}(s,t) \boldsymbol{\Delta}(s)$ and $\tilde{y}_i(\tau+1) = \boldsymbol{e}_i^\top \boldsymbol{Q}(s,t) \boldsymbol{y}(s)$, and its prediction $\tilde{\boldsymbol{w}}_i(\tau+1) = \tilde{\boldsymbol{x}}_i(\tau+1)/\tilde{y}_i(\tau+1)$.

With the above notations, we now analyze the effect of the imaginary $\tau$-step averaging process. As a prerequisite, we can verify that, the previously derived lemmas on model parameters, namely Lemma 8 and Lemma 9, also hold for $\tilde{\boldsymbol{x}}$ and $\tilde{y}$. Specifically, for Lemma 8, we directly utilize $\tilde{\boldsymbol{y}}(k) = \boldsymbol{y}(k-s)$; for Lemma 9, we know it holds for $k = 1$ and then induct on $k = 2, \ldots, \tau$.

We first try to give a matrix form of the evolution of predictions, just like our previously derived matrix form (4) of the recursion of model parameters $\tilde{\boldsymbol{X}}(k)$ and $\tilde{\boldsymbol{y}}(k)$. To this end, we first give a definition of the predictions $\tilde{w}_i^l(k)$ on virtual nodes $b_i^l$. However, a problem will arise in the formulation, namely, some $\tilde{y}_i^l(k)$ may be zero during the averaging process, so directly computing $\tilde{\boldsymbol{x}}_i^l(k)/\tilde{y}_i^l(k)$ does not necessarily make sense. To tackle this issue, from Lemma 9 we know that $\tilde{y}_i^l(k) = 0$ gives $\tilde{\boldsymbol{x}}_i^l(k) = \boldsymbol{0}$, we can thus set its prediction $\tilde{w}_i^l(k) = \boldsymbol{0}$ if $\tilde{y}_i^l(k) = 0$. Now the predictions of all nodes in $\tilde{V}$ are well-defined. For a more compact expression, we define the vector $\tilde{\boldsymbol{y}}^{-1}(k) = (\tilde{y}_1^{-1}(k), \ldots, \tilde{y}_{m(D+1)}^{-1}(k)) \in \mathbb{R}^{m(D+1)}$ as

$$\tilde{y}_h^{-1}(k) = \begin{cases} 1/\tilde{y}_h(k), & \tilde{y}_h(k) > 0, \\ 0, & \tilde{y}_h(k) = 0. \end{cases}$$

For any $1 \le l \le k \le \tau$, the matrix form of predictions $\tilde{\boldsymbol{W}}(k+1)$ can be expressed via $\tilde{\boldsymbol{X}}(l)$ as

$$\begin{aligned} \tilde{\boldsymbol{W}}(k+1) &= \mathrm{diag}(\tilde{\boldsymbol{y}}^{-1}(k+1))\tilde{\boldsymbol{X}}(k+1) \\ &= \mathrm{diag}(\tilde{\boldsymbol{y}}^{-1}(k+1))\boldsymbol{P}(s+k-1)\tilde{\boldsymbol{X}}(k) \\ &= \cdots \\ &= \mathrm{diag}(\tilde{\boldsymbol{y}}^{-1}(k+1))\boldsymbol{Q}(s+l-1, s+k)\tilde{\boldsymbol{X}}(l), \end{aligned}$$

where $\mathrm{diag}(\boldsymbol{u})$ denotes the diagonal matrix, whose $(i,i)$-th entry takes $u_i$ for all $i$, and other entries are zero.

From the above expression, we can directly derive the recursive function of the stacked predictions. The following lemma gives an arbitrary-order recursive relation on $\tilde{\boldsymbol{W}}(k)$.

**Lemma 10.** *In the above push-sum averaging process, for any $1 \leq l \leq k \leq \tau + 1$, the stacked predictions satisfy the recursive relation*

$$\tilde{\boldsymbol{W}}(k) = \boldsymbol{A}(l, k)\tilde{\boldsymbol{W}}(l),$$

*where the (multi-hop) transition matrix $\boldsymbol{A}(l, k)$ for decisions is defined as*

$$\boldsymbol{A}(l, k) = diag(\tilde{\boldsymbol{y}}^{-1}(k))\boldsymbol{Q}(s + l - 1, s + k - 1)diag(\tilde{\boldsymbol{y}}(l))$$

*when $l < k$, and $\boldsymbol{A}(l, k) = I_{m(D+1)}$ when $l = k$.*

*Proof.* Since the lemma trivially holds for $l = k$, we only need to investigate the case of $l < k$. We first prove that $\tilde{\boldsymbol{X}}(k) = \text{diag}(\tilde{\boldsymbol{y}}(k))\tilde{\boldsymbol{W}}(k)$ for any $k \in \{1, \ldots, \tau + 1\}$. Specifically, we check each row of the matrices on both sides. Recall our definition of predictions $\tilde{\boldsymbol{w}}_i^l(k)$ on virtual nodes $b_i^l$. For any $h \leq m(D + 1)$, the $h$-th row of $\tilde{\boldsymbol{W}}(k)$ is exactly (remind that, we use $\boldsymbol{A}_{i\cdot} = \boldsymbol{e}_i^\top \boldsymbol{A}$ to represent the $i$-th row vector of any matrix $\boldsymbol{A}$)

$$\tilde{\boldsymbol{W}}_{h\cdot}(k) = \begin{cases} \tilde{\boldsymbol{X}}_{h\cdot}(k)/\tilde{y}_h(k), & \tilde{y}_h(k) > 0, \\ \boldsymbol{0}, & \tilde{y}_h(k) = 0. \end{cases}$$

From Lemma 9, when $\tilde{y}_h(k) = 0$ for some virtual node, its corresponding local model parameters $\tilde{\boldsymbol{x}}_h(k)$ always satisfies $\tilde{\boldsymbol{X}}_{h\cdot}(k) = \boldsymbol{0}$. Besides, from Lemma 8, the push-sum weight parameter $y_i(k)$ of any actual node $i \in V$ is positive. Therefore, it is easy to check that, for any $h \in \tilde{V} = \{1, \ldots, m(D + 1)\}$, the $h$-th row of $\tilde{\boldsymbol{X}}(k)$ satisfies

$$\tilde{\boldsymbol{X}}_{h\cdot}(k) = \tilde{\boldsymbol{W}}_{h\cdot}(k) \cdot \tilde{y}_h(k).$$

From the arbitrariness of $h \in \tilde{V}$, we derive $\tilde{\boldsymbol{X}}(k) = \text{diag}(\tilde{\boldsymbol{y}}(k))\tilde{\boldsymbol{W}}(k)$. Plugging $\tilde{\boldsymbol{X}}(l) = \text{diag}(\tilde{\boldsymbol{y}}(l))\tilde{\boldsymbol{W}}(l)$ into the above relation $\tilde{\boldsymbol{X}}(k) = \text{diag}(\tilde{\boldsymbol{y}}(k))\tilde{\boldsymbol{W}}(k)$, we prove the lemma. $\square$

We now focus on the transition matrix $\boldsymbol{A}(l, k)$ of predictions, which is determined by the multi-hop communication matrix $\boldsymbol{Q}(s + l - 1, s + k - 1)$ induced by push-sum and the stacked weights $\tilde{\boldsymbol{y}}(l), \tilde{\boldsymbol{y}}(k)$, as suggested in the above lemma. We already know that $\boldsymbol{Q}(s, t)$ is non-negative and column stochastic, and expect $\boldsymbol{A}(l, k)$ to inherit some properties from $\boldsymbol{Q}(s, t)$. In addition, the transition matrix $\boldsymbol{A}(l, k)$ of predictions is analogous to the mixing matrices $\boldsymbol{P}$ in other decentralized online algorithms [36, 38]. However, their mixing matrices $\boldsymbol{P}$ are conventionally doubly stochastic (i.e., $\boldsymbol{P}\boldsymbol{1} = \boldsymbol{P}^\top\boldsymbol{1} = \boldsymbol{1}$), whereas our transition matrices are not. Therefore, their analysis which relies on the doubly stochastic property and spectral gaps can not be directly applied to our setting.

To tackle this issue, we introduce the technique in [25], but make a refinement on the analysis which *saves a factor of $\sqrt{n}$* in the convergence rate, where $n$ is the dimension of the decision set. Specifically, we utilize the internal structure of incremental term $\boldsymbol{\Delta}(s)$, which is a rank-one matrix, namely $\boldsymbol{\Delta}(s) = \boldsymbol{e}_{i_s}\boldsymbol{\epsilon}_s$, since there is only a single learner performing local update at each step. This enables us to tackle the $n$ dimensions of $\boldsymbol{\epsilon}_s$ simultaneously by directly analyzing its $L2$-norm $\|\boldsymbol{\epsilon}_s\|$ (in our analysis, $\|\cdot\|$ represents the $L2$-norm). In contrast, [25] tackles each of the $n$ dimensions separately. In their approach, the $k$-th dimension contributes a coefficient $|\epsilon_{s,k}|$ ($\epsilon_{s,k}$ is the $k$-th entry of $\boldsymbol{\epsilon}_s$) to the convergence rate; summing them together gives a bound $\|\boldsymbol{\epsilon}_s\|_1$ in terms of $L1$-norm (see Corollary 1 in [25]). To transform it into $L2$-norm, one need to utilize $\|\boldsymbol{\epsilon}_s\|_1 \leq \sqrt{n}\|\boldsymbol{\epsilon}_s\|$, which will incur an extra coefficient $\sqrt{n}$ as the expense of expressing the final bound in terms of $L2$-norm $\|\boldsymbol{\epsilon}_s\|$.

More specifically, since $\boldsymbol{\Delta}(s) = \boldsymbol{e}_{i_s}\boldsymbol{\epsilon}_s$, the quantity $R_i(s, t)$ can be written as

$$R_i(s, t) = \|\frac{\boldsymbol{e}_i^\top \boldsymbol{Q}(s, t)\boldsymbol{e}_{i_s}\boldsymbol{\epsilon}_s}{\boldsymbol{e}_i^\top \boldsymbol{Q}(s, t)\boldsymbol{y}(s)} - \frac{\boldsymbol{\epsilon}_s}{m}\|.$$

Since both $\boldsymbol{e}_i^\top \boldsymbol{Q}(s, t)\boldsymbol{e}_{i_s}$ and $\boldsymbol{e}_i^\top \boldsymbol{Q}(s, t)\boldsymbol{y}(s)$ are scalars, we further have

$$R_i(s, t) \leq |\frac{\boldsymbol{e}_i^\top \boldsymbol{Q}(s, t)\boldsymbol{e}_{i_s}}{\boldsymbol{e}_i^\top \boldsymbol{Q}(s, t)\boldsymbol{y}(s)} - \frac{1}{m}|\|\boldsymbol{\epsilon}_s\|. \tag{11}$$

Concretely, we investigate the effect of transition matrix $\boldsymbol{A}(l, k)$ on an arbitrary vector $\boldsymbol{u} \in \mathbb{R}^{m(D+1)}$, which satisfies $\boldsymbol{u}_h = 0$ for any $h \in \tilde{V}$ such that $\tilde{y}_h(k) = 0$ (then we can set $\boldsymbol{u} = \text{diag}(\tilde{\boldsymbol{y}}^{-1}(1))\boldsymbol{e}_{i_s} =$

$e_{i_s}/y_{i_s}(s)$ to restore the averaging process). Enlightened by the analysis in [31], we use the index set $I^k = \{h \leq m(D+1) \mid \tilde{y}_h(k) > 0\}$ to denote the positive entries of $\tilde{\boldsymbol{y}}(k)$. Lemma 8 implies that $V \subset I^k$. From Lemma 9, only those nodes inside $I^k$ will give valid predictions (in fact, for any other node $h \notin I^k$, its parameters $\tilde{\boldsymbol{x}}_h(k), \tilde{y}_h(k)$ are all zero, which cannot give any meaningful predictions). Therefore, we are mainly interested in the nodes inside $I^k$ at each step $k$. In this spirit, we call $\{u_j \mid j \in I^l\}$ the *valid entries* of $\boldsymbol{u} = (u_1, \ldots, u_{m(D+1)})$ at step $l$. Denote $\boldsymbol{v} = \boldsymbol{A}(l, k)\boldsymbol{u}$, we similarly call $\{v_j \mid j \in I^k\}$ the valid entries of $\boldsymbol{v}$ at step $k$. We can now give some general properties of $\boldsymbol{A}(l, k)$ as below.

**Lemma 11.** *For all $1 \leq l \leq k \leq \tau + 1$, the transition matrix $\boldsymbol{A}(l, k)$ of decisions satisfies:*

*(a) If $i \notin I^k$ or $j \notin I^l$, then $\boldsymbol{A}_{ij}(l, k) = 0$.*

*(b) The entries in the $h$-th row of $\boldsymbol{A}(l, k)$ sum up to*

$$\sum_{j=1}^{m(D+1)} \boldsymbol{A}_{hj}(l, k) = \sum_{j \in I^l} \boldsymbol{A}_{hj}(l, k) = \begin{cases} 1, & h \in I^k, \\ 0, & h \notin I^k. \end{cases}$$

*Proof.* (a) From the definition of $\boldsymbol{A}(l, k)$ in Lemma 10, its $(i, j)$-th entry

$$\boldsymbol{A}_{ij}(l, k) = \tilde{y}_i^{-1}(k)\boldsymbol{Q}_{ij}(s + l - 1, s + k - 1)\tilde{y}_j(l),$$

which equals to zero if $i \notin I^k$ or $j \notin I^l$.

(b) Since $\tilde{\boldsymbol{y}}(k)$ denotes the parameter after $k - 1$ push-sum steps starting from step $s$, we actually have $\tilde{\boldsymbol{y}}(k) = \boldsymbol{y}(s + k - 1)$. Therefore, the entries in the $h$-th row of $\boldsymbol{A}(l, k)$ sum up to

$$\begin{aligned}
\boldsymbol{e}_h^\top \boldsymbol{A}(l, k)\mathbf{1} &= \boldsymbol{e}_h^\top \mathrm{diag}(\tilde{\boldsymbol{y}}^{-1}(k))\boldsymbol{Q}(s + l - 1, s + k - 1)\tilde{\boldsymbol{y}}(l) \\
&= \tilde{y}_h^{-1}(k) \cdot \boldsymbol{e}_h^\top \boldsymbol{Q}(s + l - 1, s + k - 1)\tilde{\boldsymbol{y}}(l) \\
&= \tilde{y}_h^{-1}(k) \cdot \boldsymbol{e}_h^\top \boldsymbol{Q}(s + l - 1, s + k - 1)\boldsymbol{y}(s + l - 1) \\
&= \tilde{y}_h^{-1}(k) \cdot \boldsymbol{e}_h^\top \boldsymbol{y}(s + k - 1) = \tilde{y}_h^{-1}(k) \cdot \tilde{y}_h(k) \\
&= \begin{cases} 1, & h \in I^k, \\ 0, & h \notin I^k. \end{cases}
\end{aligned}$$

Moreover, we have $\boldsymbol{A}_{hj}(l, k) = 0$ if $j \notin I^l$. Therefore, for any $h \in I^k$, it holds that

$$1 = \boldsymbol{A}_{h\cdot}(l, k)\mathbf{1} = \sum_{j=1}^{m(D+1)} \boldsymbol{A}_{hj}(l, k) = \sum_{j \in I^l} \boldsymbol{A}_{hj}(l, k),$$

which proves the lemma. $\qquad\square$

Let's return to $\boldsymbol{v} = \boldsymbol{A}(l, k)\boldsymbol{u}$. The following lemma shows that, the transition matrix of predictions induced by push-sum steps will shrink the range of the valid entries of a vector, which is actually the core step to the analysis of push-sum.

**Lemma 12.** *Let $\boldsymbol{u} = (u_1, \ldots, u_{m(D+1)})$ be an arbitrary vector in $\mathbb{R}^{m(D+1)}$. Set $\boldsymbol{v} = \boldsymbol{A}(l, k)\boldsymbol{u}$ and denote $\boldsymbol{v} = (v_1, \ldots, v_{m(D+1)})$. The index set of valid entries $I^k = \{h : \tilde{y}_h(k) > 0\}$ is defined above. Then it holds that*

$$\max_{h \in I^k} v_h - \min_{h \in I^k} v_h \leq \max_{h \in I^l} u_h - \min_{h \in I^l} u_h.$$

*Proof.* For any $h \in I^k$, the $h$-th entry of $\boldsymbol{v}$ takes

$$v_h = \sum_{j=1}^{m(D+1)} \boldsymbol{A}_{hj}(l, k)u_j = \sum_{j \in I^l} \boldsymbol{A}_{hj}(l, k)u_j.$$

From Lemma 10 and Lemma 11 we know that $\boldsymbol{A}_{hj}(l, k) \geq 0, \forall j \in I^l$ and $\sum_{j \in I^l} \boldsymbol{A}_{hj}(l, k) = 1$, respectively. Hence $v_h$ is a convex combination of the elements in $\{u_j \mid j \in I^l\}$. Consequently, we have

$$\min_{j \in I^l} u_j \leq v_h \leq \max_{j \in I^l} u_j.$$

Since $h$ is arbitrarily chosen from $I^k$, we have

$$\min_{j \in I^l} u_j \leq \min_{h \in I^k} u_h \leq \max_{h \in I^k} u_h \leq \max_{j \in I^l} u_j,$$

which proves the lemma. $\qquad\square$

The above lemma implies that, any push-sum step will not expand the range of the entries in any given vector. To show the convergence to consensus, we expect that, after operating sufficient number of push-sum steps, the *range* of a vector will be scaled down. Such convergence effect is proved in the following lemma, in which the minimal number of steps $(\mathcal{D} + 1)\Gamma$ stems from Lemma 7, namely the minimal number of hops $t - s$ for $\boldsymbol{Q}(s, t)$ to guarantee the positiveness of its first $m$ rows. We note that, a similar approach to derive the convergence effect can be seen in the analysis of symmetric communication [36, 21], which relies on the double stochasticity of the transition matrices. However, their methods are unsuitable to our setting, where $\boldsymbol{A}(l, k)$ is actually not doubly stochastic.

As a prerequisite, we derive some properties of any $r = (\mathcal{D} + 1)\Gamma$-hop transition matrix $\boldsymbol{A}(l, l + r)$ in the lemma below. Remind that, the vertex set and the augmented vertex set are specified as $V = \{1, \ldots, m\}$ and $\tilde{V} = \{1, \ldots, m(D + 1)\}$, respectively.

**Lemma 13.** *Set $r = (\mathcal{D} + 1)\Gamma$. For any $l \geq 1$, the transition matrix $\boldsymbol{A}(l, l + r)$ satisfies:*

*(a) The first $m \times m$ block of $\boldsymbol{A}(l, l + r)$ is positive, namely $\boldsymbol{A}_{ij}(l, l + r) > 0, \forall i, j \in V$.*

*(b) The positive entries in the first $m$ columns of $\boldsymbol{A}(l, l + r)$ are at least $\alpha^2$. Moreover, the positive entries in the last $mD$ columns of $\boldsymbol{A}(l, l + r)$ are at least $\alpha^2/m$.*

*(c) For any $m + 1 \leq h \leq m(D + 1)$, if $h \in I^{l+r}$, then there must exist some $i \in V$ such that $\boldsymbol{A}_{hi}(l, l + r) \geq \alpha^2$.*

*Proof.* From Lemma 10, for any $i, j \in \tilde{V}$, $\boldsymbol{A}_{ij}(l, l+r) = \tilde{y}_i^{-1}(l+r)\boldsymbol{Q}_{ij}(s+l-1, s+l+r-1)\tilde{y}_j(l)$.

(a) When $i, j \in V$, from Lemma 8, $\tilde{y}_i(l + r), \tilde{y}_j(l) > 0$; also from Lemma 7, $\boldsymbol{Q}_{ij}(s + l - 1, s + l + r - 1) > 0$. Therefore, $\boldsymbol{A}_{ij}(l, l + r) > 0$.

(b) Suppose $\boldsymbol{A}_{hi}(l, l + r) > 0$ for some $h \in \tilde{V}, i \in V$. Then we have $\tilde{y}_h(t + s) > 0$ and $\boldsymbol{Q}_{hi}(s + l - 1, s + l + r - 1) > 0$. From Lemma 8, we have $1/\tilde{y}_h(l + r) \geq 1/m, \tilde{y}_i(t) \geq m\alpha$; in addition, from Lemma 7, we have $\boldsymbol{Q}_{hi}(s + l - 1, s + l + r - 1) \geq \alpha$, therefore $\boldsymbol{A}_{hi}(l, l + r) \geq (1/m)\alpha(m\alpha) = \alpha^2$. Similarly, we can prove that, if $\boldsymbol{A}_{hj}(l, l + r) > 0$ for some $h \in \tilde{V}, j > m$, then $\boldsymbol{A}_{hj}(l, l + r) \geq (1/m)\alpha\alpha = \alpha^2/m$.

(c) From (b), we only need to prove that $\boldsymbol{A}_{hi}(l, l + r) > 0$ for some $i \in V$. Since $\boldsymbol{A}(l, l + r)$ is non-negative, and from Lemma 8 $\tilde{y}_i(l) > 0$ for all $i \in V$, it suffices to prove that $\boldsymbol{Q}_{hi}(s + l - 1, s + l + r - 1) > 0$ for some $i \in V$. For simplicity we denote $p = s + l - 1$. We now assume $\boldsymbol{Q}_{hi}(p, p + r) = 0$ for all $i \in V$. From the definition of multi-hop communication matrix, we have $\boldsymbol{Q}_{ii}(p + k, p + r) > 0$ for all $i \in V, k \leq r$. Since $\boldsymbol{Q}_{hi}(p, p + r) \geq \boldsymbol{Q}_{hi}(p, p + k)\boldsymbol{Q}_{ii}(p + k, p + r)$ for all $k \leq r$, we must have $\boldsymbol{Q}_{hi}(p, p + k) = 0$ for all $i \in V, k \leq r$, which means that the information on any actual node $i \in V$ will not reach the virtual node $h$ during step $l$ and $l + r$. Since the message delay is at most $D < r - 1$, then the node $h$ will not capture any message at step $l + r$, which means $\tilde{y}_h(l + r) = 0$ and will lead to contradiction. $\qquad\square$

Given the above lemma, we now return to the relation $\boldsymbol{v} = \boldsymbol{A}(l, l + r)\boldsymbol{u}$. The following lemma shows that $\boldsymbol{A}(l, l + r)$ will shrink the range of the valid entries of $\boldsymbol{u}$. Then, after sufficient number of push-sum steps, all valid entries of the vector tend to reach a consensus (i.e., attain a fixed value).

**Lemma 14.** *Given $1 \leq l \leq k \leq \tau + 1$. Assume $\boldsymbol{u} = (u_1, \ldots, u_{m(D+1)}) \in \mathbb{R}^{m(D+1)}$ satisfies $\tilde{y}_h(l) = 0 \Rightarrow u_h = 0, \forall h \in \tilde{V} = \{1, \ldots, m(D + 1)\}$. Define*

$$\beta(\boldsymbol{w}; p) = \max_{h \in I^p} w_h(p) - \min_{h \in I^p} w_h(p)$$

*as the range of the valid entries of $\boldsymbol{w} = (w_1, \ldots, w_{m(D+1)})$ at step $p$. Set $r = (\mathcal{D} + 1)\Gamma$, then we have*

$$\beta(\boldsymbol{A}(l, k)\boldsymbol{u}; k) \leq (1 - m\alpha^4)^{\lfloor \frac{k-l}{2r} \rfloor} \beta(\boldsymbol{u}; l).$$

*Proof.* Denote $\boldsymbol{v} = \boldsymbol{A}(l,k)\boldsymbol{u}$ and $q = \lfloor (k-l)/r \rfloor$. We construct a sequence of vectors $\{\boldsymbol{u}(p) \mid 0 \le p \le q+1\}$, namely, $\boldsymbol{u}(0) = \boldsymbol{u}$, $\boldsymbol{u}(p) = \boldsymbol{A}(l+(p-1)r, l+pr)\boldsymbol{u}(p-1)$ for $p \in \{1,\ldots,q\}$, and $\boldsymbol{u}(q+1) = \boldsymbol{A}(l+qr,k)\boldsymbol{u}(q)$. We now specify $\boldsymbol{u}(p) = (u_1(p),\ldots,u_{m(D+1)}(p))$ for any $p \in \{1,\ldots,q+1\}$.

As assumed in this lemma, $\tilde{y}_h(l) = 0$ leads to $u_h = 0$. From the definition of $\boldsymbol{A}(l,r)$ in Lemma 10, we can inductively prove that $\tilde{y}_h(l+pr) = 0$ leads to $u_h(p) = 0$, and further $\boldsymbol{u}(p) = \boldsymbol{A}(l, l+pr)\boldsymbol{u}(0)$ for all $p \in \{1,\ldots,q\}$. Similarly, $\tilde{y}_h(k) = 0$ leads to $u_h(p+1) = 0$; and we also have $\boldsymbol{u}(p+1) = \boldsymbol{A}(l,k)\boldsymbol{u}(0) = \boldsymbol{v}$.

For any $0 \le p \le q$, denote $\beta_p = \beta(\boldsymbol{u}(p); l+pr)$. We now derive the recursive relation of $\beta_p$. Specifically, we introduce an auxiliary sequence $\{\gamma_p\}_{p \in \{0,\ldots,q\}}$, i.e.,

$$\gamma_p = \max_{i \in V} u_i(p) - \min_{i \in V} u_i(p),$$

which measures the range of the first $m$-entries of $\boldsymbol{u}(p)$. From Lemma 8, the first $m$ entries of $\tilde{\boldsymbol{y}}(k)$ must be positive for any $k$. Therefore, for any $p \in \{1,\ldots,q\}$, we have $V \subset I^{l+pr}$ and further

$$\max_{i \in V} u_i(p) \le \max_{h \in I^{l+pr}} u_i(p),$$
$$\min_{i \in V} u_i(p) \ge \min_{h \in I^{l+pr}} u_i(p),$$

which implies $\gamma_p \le \beta_p$. Now we investigate the relation between $\gamma_p, \beta_p$ and $\gamma_{p-1}, \beta_{p-1}$.

(i) First we consider an arbitrary actual node $i \in V$. From Lemma 8, $\tilde{y}_i(l+pr) > 0$. We abbreviate $\boldsymbol{A}_{ih}(l+(p-1)r, l+pr)$ into $c_h$ for $h \in \tilde{V}$. Since $\boldsymbol{A}(l+(p-1)r, l+pr)$ is non-negative, we have $c_h \ge 0$. From Lemma 11 we further have $\sum_{h \in \tilde{V}} c_h = 1$. Thus $u_i(p) = \sum_{h \in \tilde{V}} c_h u_h(p-1)$, which means that $u_i(p)$ is a convex combination of the entries in $\boldsymbol{u}(p-1)$. From Lemma 13 we also know that $c_j \ge \alpha^2$ for any $j \in V$. Denote the maximal valid entry of $\boldsymbol{u}(p-1)$ as $u' = \max_{h \in I^{l+(p-1)r}} u_h(p-1)$, then we have

$$u_i(p) = \sum_{h=1}^{m(D+1)} c_h u_h(p-1) = \sum_{j=1}^{m} c_j u_j(p-1) + \sum_{h=m+1}^{m(D+1)} c_h u_h(p-1)$$
$$\le \sum_{j=1}^{m} c_j u_j(p-1) + u'(1 - \sum_{j=1}^{m} c_h) \le \sum_{j=1}^{m} \alpha^2 u_j(p-1) + u'(1 - m\alpha^2),$$

where the last step utilizes $c_j \ge \alpha^2$ and $u_j(p-1) \le u'$ for $j \in V$. Similarly, denote the minimal valid entry of $\boldsymbol{u}(p-1)$ as $u'' = \min_{h \in I^{l+(p-1)r}} u_h(p-1)$, then we have

$$u_i(p) = \sum_{h=1}^{m(D+1)} c_h u_h(p-1) = \sum_{j=1}^{m} c_j u_j(p-1) + \sum_{h=m+1}^{m(D+1)} c_h u_h(p-1)$$
$$\ge \sum_{j=1}^{m} c_j u_j(p-1) + u''(1 - \sum_{j=1}^{m} c_h) \ge \sum_{j=1}^{m} \alpha^2 u_j(p-1) + u''(1 - m\alpha^2).$$

Since the above two inequalities hold for all $i \in V$, we set $i = \arg\max_{j \in V} u_j(p)$ in the first inequality, and set $i = \arg\min_{j \in V} u_j(p)$ in the second inequality, to derive

$$\gamma_p = \max_{j \in V} u_j(p) - \min_{j \in V} u_j(p) \le (1 - m\alpha^2)(u' - u'') = (1 - m\alpha^2)\beta_{p-1}.$$

(ii) Then we consider an arbitrary "valid" node $i' \in I^{l+pr}$, and denote $c_h = \boldsymbol{A}_{i'h}(l+(p-1)r, l+pr)$ for $h \in \tilde{V}$. Via a very similar deduction in (i), we can prove that $u_{i'}(p) = \sum_{h \in \tilde{V}} c_h u_h(p-1)$ is a convex combination of the entries in $\boldsymbol{u}(p-1)$. However, since $c_h$ may locate at the last $mD$ rows of $\boldsymbol{A}(l+(p-1)r, l+pr)$, from Lemma 13 we only have $c_j \ge \alpha^2$ for some $j \in V$, which leads to $\sum_{j \in V} c_j \ge \alpha^2$. Therefore, using the same notations $u'$ and $u''$ as the first case (i) to denote the

maximal and minimal valid entry of $\boldsymbol{u}(p-1)$, respectively, we have

$$u_{i'}(p) = \sum_{h=1}^{m(D+1)} c_h u_h(p-1) = \sum_{j=1}^{m} c_j u_j(p-1) + \sum_{h=m+1}^{m(D+1)} c_h u_h(p-1)$$

$$\leq \sum_{j=1}^{m} c_j u_j(p-1) + u'(1 - \sum_{j=1}^{m} c_h) \leq \alpha^2 \max_{j \in V} u_j(p-1) + u'(1 - \alpha^2),$$

and similarly

$$u_{i'}(p) = \sum_{h=1}^{m(D+1)} c_h u_h(p-1) = \sum_{j=1}^{m} c_j u_j(p-1) + \sum_{h=m+1}^{m(D+1)} c_h u_h(p-1)$$

$$\geq \sum_{j=1}^{m} c_j u_j(p-1) + u''(1 - \sum_{j=1}^{m} c_h) \geq \alpha^2 \min_{j \in V} u_j(p-1) + u''(1 - \alpha^2).$$

Set $i' = \arg\max_{h \in I^{l+pr}} u_h(p)$ in the first inequality, and set $i' = \operatorname{argmin}_{h \in I^{l+pr}} u_h(p)$ in the second inequality, then we have

$$\beta_p \leq \alpha^2 \gamma_{p-1} + (1 - \alpha^2)\beta_{p-1}.$$

From Lemma 12, we have $\beta_{p-1} \leq \beta_{p-2}$. Hence we can combine both cases (i) and (ii) to derive

$$\beta_p \leq \alpha^2(1 - m\alpha^2)\beta_{p-2} + (1 - \alpha^2)\beta_{p-1} \leq (1 - m\alpha^4)\beta_{p-2},$$

which gives the recursive relation of $\beta_p$.

Now we can prove the desired lemma. Specifically, from the above recursion and the fact that $\beta_p \leq \beta_{p-1}, \forall p \in \{1, \ldots, q\}$, we have

$$\beta(\boldsymbol{u}(q); l + qr) = \beta_q \leq \beta_{2\lfloor \frac{q}{2} \rfloor} \leq (1 - m\alpha^4)\beta_{2(\lfloor \frac{q}{2} \rfloor - 1)} \leq \cdots$$

$$\leq (1 - m\alpha^4)^{\lfloor \frac{q}{2} \rfloor}\beta_0 = (1 - m\alpha^4)^{\lfloor \frac{q}{2} \rfloor}\beta(\boldsymbol{u}, l),$$

and consequently,

$$\beta(\boldsymbol{v}; k) \leq \beta(\boldsymbol{u}(q); l + qr) \leq (1 - m\alpha^4)^{\lfloor \frac{q}{2} \rfloor}\beta(\boldsymbol{u}, l).$$

Since $q = \lfloor (k - l)/r \rfloor$, we have $\lfloor q/2 \rfloor \leq \lfloor (k - l)/2r \rfloor$, which proves the lemma. $\qquad\square$

To relate the above lemma with the imaginary push-sum averaging process on $\tilde{\boldsymbol{X}}(1)$ and $\tilde{\boldsymbol{y}}(1)$, we set $l = 1, k = \tau + 1$, and $\boldsymbol{u} = \boldsymbol{e}_{i_s}/\tilde{y}_{i_s}(1)$ in the above lemma. Also denote $\boldsymbol{z} = \boldsymbol{A}(1, \tau + 1)\boldsymbol{e}_{i_s}/\tilde{y}_{i_s}(1) \in \mathbb{R}^{m(D+1)}$ and specify its entries as $\boldsymbol{z} = (z_1, \ldots, z_{m(D+1)})$. Recall the inequality (11) where the bound of $R_i(s, t)$ is related with the term $|z_i - 1/m|$. In other words, we need to compare the quantity $z_i$ with $1/m$. Notice that $\min_{h \in I^{\tau+1}} z_h \leq z_i \leq \max_{h \in I^{\tau+1}} z_h$. We now verify that $\min_{h \in I^{\tau+1}} z_h \leq 1/m \leq \max_{h \in I^{\tau+1}} z_h$.

**Lemma 15.** *The transition matrix of predictions $\boldsymbol{A}(1, \tau + 1)$ is defined as above. Set $\boldsymbol{z} = (z_1, \ldots, z_{m(D+1)}) = \boldsymbol{A}(1, \tau + 1)\boldsymbol{e}_{i_s}/\tilde{y}_{i_s}(1)$, then we have*

$$\min_{h \in I^{\tau+1}} z_h \leq \frac{1}{m} \leq \max_{h \in I^{\tau+1}} z_h.$$

*Proof.* It suffices to prove that, $1/m$ is a convex combination of the elements in $\{z_h \mid h \in I^{\tau+1}\}$. To this end, we observe that, $z_h$ can be viewed as the prediction of learner $h \in I^{\tau+1}$ after $\tau$ push-sum steps, where the scalar models and push-sum weights of each learner $k \in \tilde{V}$ are initialized as the $k$-th entry of $\boldsymbol{e}_{i_s}$ and $\tilde{\boldsymbol{y}}(1)$, respectively. Concretely, from the definition of $\boldsymbol{A}(1, \tau + 1)$ in Lemma 10, we have

$$z_h = \frac{\boldsymbol{e}_h^\top \boldsymbol{Q}(s, t)\boldsymbol{e}_{i_s}}{\tilde{y}_h(\tau + 1)} = \frac{\boldsymbol{Q}_{h\cdot}(s, t)\boldsymbol{e}_{i_s}}{\tilde{y}_h(\tau + 1)}.$$

Recall that $\boldsymbol{1}^\top \boldsymbol{e}_{i_s} = 1$ and $\boldsymbol{1}^\top \tilde{\boldsymbol{y}}(\tau + 1) = \boldsymbol{1}^\top \tilde{\boldsymbol{y}}(t) = m$; in addition, $\boldsymbol{Q}(s, t)$ is column stochastic. Hence we have

$$\frac{1}{m} = \frac{\boldsymbol{1}^\top \boldsymbol{Q}(s, t)\boldsymbol{e}_{i_s}}{m} = \sum_{h \in \tilde{V}} \frac{\boldsymbol{Q}_{h\cdot}(s, t)\boldsymbol{e}_{i_s}}{m}.$$

From Lemma 9, for any node $h \in \tilde{V} - I^{\tau+1}$, its push-sum weight $\tilde{y}_h(\tau+1) = 0$ and local scalar model $\boldsymbol{Q}_{h\cdot}(s,t)\boldsymbol{e}_{i_s} = 0$. Therefore, we further have

$$\frac{1}{m} = \sum_{h \in I^{\tau+1}} \frac{\boldsymbol{Q}_{h\cdot}(s,t)\boldsymbol{e}_{i_s}}{m} = \sum_{h \in I^{\tau+1}} \frac{\tilde{y}_h(\tau+1)}{m} \cdot \frac{\boldsymbol{Q}_{h\cdot}(s,t)\boldsymbol{e}_{i_s}}{\tilde{y}_h(\tau+1)} = \sum_{h \in I^{\tau+1}} \frac{\tilde{y}_h(\tau+1)}{m} z_h,$$

which is a convex combination of $\{z_h \mid h \in I^{\tau+1}\}$, because $\tilde{y}_h(\tau+1) \geq 0$ and $\sum_{h \in I^{\tau+1}} \tilde{y}_h(\tau+1)/m = \sum_{h \in \tilde{V}} \tilde{y}_h(\tau+1)/m = 1$. Therefore, $1/m$ must lie between the maximal and the minimal values of the elements in $\{z_h \mid h \in I^{\tau+1}\}$, which proves the lemma. $\qquad\square$

Set $\lambda = 1 - m\alpha^4$, $B = (\mathcal{D}+1)\Gamma = (\mathcal{D}+1)(D^{msg} + \Gamma_d)$, and $\boldsymbol{u} = \boldsymbol{e}_{i_s}/\tilde{y}_{i_s}(1)$. From the above lemma, the term $|z_i - 1/m|$ can be bounded as

$$|z_i - \frac{1}{m}| \leq \max_{h \in I^{\tau+1}} z_h - \min_{h \in I^{\tau+1}} z_h$$

$$\leq \lambda^{\lfloor \frac{\tau}{2B} \rfloor}(\max_{h \in I^1} u_h - \min_{h \in I^1} u_h) = \frac{1}{\tilde{y}_{i_s}(1)}\lambda^{\lfloor \frac{t-s}{2B} \rfloor} = \frac{1}{y_{i_s}(s)}\lambda^{\lfloor \frac{t-s}{2B} \rfloor}.$$

Now, we successfully derive a bound for the term $R_i(s,t)$ for any $i \in V, s \in \mathcal{Q}_T, t \in \{s, s+1, \ldots, 2T\}$, i.e.,

$$R_i(s,t) = |z_i - \frac{1}{m}| \|\boldsymbol{\epsilon}_s\| \leq \lambda^{\lfloor \frac{t-s}{2B} \rfloor} \frac{\|\boldsymbol{\epsilon}_s\|}{y_{i_s}(s)}.$$

Moreover, recall our definition of the incremental term $\boldsymbol{\epsilon}_s = -\eta \boldsymbol{g}_s + \boldsymbol{r}_s$. From Lemma 4, the residual term $\|\boldsymbol{r}_s\| \leq \eta g_s$. Therefore, for any $i \in V, s \in \mathcal{Q}_T, t \in \{s, s+1, \ldots, 2T\}$, we have

$$R_i(s,t) \leq 2\eta\lambda^{\lfloor \frac{t-s}{2B} \rfloor} \frac{g_s}{y_{i_s}(s)}.$$

Finally, we plug the above inequality of $R_i(s,t)$ into the general regret bound (10), then derive the final bound in the main theorem.

# E    Omitted Proof of Corollary 1

In this section, we provide the detailed proof of Corollary 1 in our main paper.

Starting from the regret bound in Theorem 1. We analyze each regret term under the bounded gradient norm assumption separately. The first term is preserved. For the second term, we have

$$\sum_{t \in \mathcal{Q}_T} \sum_{s \in \mathcal{Q}_{l_t,t+1}} g_t g_s \leq G^2 \sum_{t \in \mathcal{Q}_T} |\mathcal{Q}_{l_t,t+1}|.$$

We first focus on the elements in $Q_{l_t,t+1}$, which is defined as $\{l_t < s < t+1 \mid s \in \mathcal{Q}_T\}$ in our main paper. Specifically, for any $t \in \mathcal{Q}_T$, recall our definition of processing delay $d^p(t) = |\mathcal{Q}_{l_t,t}|$, we directly have

$$|\mathcal{Q}_{l_t,t+1}| = |\mathcal{Q}_{l_t,t} \cup \{t\}| = d^p(t) + 1.$$

Summing the above equation over $t \in \mathcal{Q}_T$, we thus have

$$\sum_{t \in \mathcal{Q}_T} |\mathcal{Q}_{l_t,t+1}| = (\sum_{t \in \mathcal{Q}_T} d^p(t)) + |\mathcal{Q}_T| = D^{proc} + T.$$

To analyze the last three terms in the regret bound, we first utilize Lemma 8 which gives $1/y_{i_t}(t) \leq 1/m\alpha$ for any $t \in \mathcal{Q}_T$. Then we investigate the geometric series of $\lambda$. Specifically, for any $t \in \mathcal{Q}_T$, we have

$$\sum_{s \in \mathcal{Q}_{1,t+1}} \lambda^{\lfloor \frac{t-s}{2B} \rfloor} \leq \sum_{s=1}^{t} \lambda^{\lfloor \frac{t-s}{2B} \rfloor} = \sum_{s=0}^{t-1} \lambda^{\lfloor \frac{s}{2B} \rfloor} \leq \sum_{s=0}^{T-1} \lambda^{\lfloor \frac{s}{2B} \rfloor} \leq \sum_{s=0}^{2B\lceil \frac{T}{2B} \rceil - 1} \lambda^{\lfloor \frac{s}{2B} \rfloor}$$

$$= \sum_{t=0}^{\lceil \frac{T}{2B} \rceil - 1} \sum_{s=0}^{2B-1} \lambda^{\lfloor \frac{2tB+s}{2B} \rfloor} = \sum_{t=0}^{\lceil \frac{T+1}{B} \rceil - 1} 2B\lambda^t \leq \frac{2B}{1-\lambda}.$$

Therefore, the last three terms in the regret bound in Theorem 1 can be bounded by

$$8\eta \sum_{t \in \mathcal{Q}_T} \frac{2B}{1-\lambda} \frac{G^2}{m\alpha} + \eta \sum_{t \in \mathcal{Q}_T} \frac{G^2}{m\alpha} = \frac{\eta}{m}(\frac{16B}{m\alpha^5} + \frac{1}{\alpha})G^2|\mathcal{Q}_T| \le \frac{2\eta}{m}(\frac{8}{\alpha^5} + \frac{1}{\alpha})BG^2T.$$

Combining the above analysis for each regret term, we prove this corollary.