# OpenReview forum: "Asynchronous Decentralized Online Learning"
_NeurIPS.cc/2021/Conference — NeurIPS 2021 Poster_

### Official Review · Reviewer_8TCf · 2021-07-15

**Rating:** 6
**Confidence:** 4

**Summary:**

This paper proposes an asynchronous decentralized learning protocol. The motivation is to remove the need for computation nodes to synchronize broadcasting their local models (e.g., to a server), which improves efficiency and brings the setting closer to practical scenarios. The main contributions include (1) an event indexing system; (2) an projection scheme to ensure weight aggregation does not exit the feasibility region; (3) an instantaneous averaging scheme that aggregates neighboring models as soon as each node goes idle and (4) a regret bound analysis.

**Limitations And Societal Impact:**

The author did not discuss any broader impact of this work, but I believe there is no potential negative impact to society.

**Main Review:**

--Clarity
The paper is clearly written although the descriptions can be a little dense. In my opinion, some diagrams would help to explain the details more efficiently. Minor questions:

1. Is line 5 in Protocol 1 supposed to be outside the if/else block? Else, how does learner i_t gets picked to perform local update?
2. For the averaging scheme after line 139, does the order of broadcasted copies matter? For instance, when we do pairwise naive averaging, the result would be different from one-shot naive averaging, e.g., (a/2 + b/2)/2 + c/2 != (a + b + c)/3
3. Seems like a typo in line 203 with the update for y_i, which I think should be y_i \leftarrow \gamma_i y_i
4. What exactly does it mean by "adversarially chosen by the environment"?

--Significance & Originality
I think the problem is important and the solution is well designed. However, I feel like the related work (Appendix) could be more extensive in reviewing other lines of works. Explicitly:
1. It seems that the related work section only focuses on decentralized literature. I believe a fair review has to include asynchronous centralized learning too.
2. When it was claimed that the event indexing system is superior because it "takes both prediction and local update into account", what are previous indexing systems designed to achieve and what exactly is the implication of this innovation? Also, apart from the computation delay, why do we have to index prediction events if they do not make any weight update? Can't they be modeled under the message delay component?
3. It was claimed that the weighted projection & instantaneous model averaging component makes the analysis more complex. To me, this feels like a consequence of incorporating these designs, rather than a contribution.

--Quality
The strength of this paper is the thorough theoretical treatment. I find both the main results and the remark (line 256-268) very insightful. One thing I find somewhat unintuitive is that to achieve sublinear regret bound, the learning rate is set to \eta = (mF/2G)\mathcal{O}(T^{-1/2}), which seems excessively large on scenarios with massive number of nodes. How would this affect the quality of local update in practice?

Regarding experiments: I find the overall quality of the empirical study somewhat disappointing for many reasons:
1. Why is there no classification error plot? I only see average loss plots, for which the loss value does not seem to improve much (i.e. on the order of 0.00x).
2. No error bar on the average plot. It would be interesting to see if different nodes have similar learning pattern, which is implied by the regret bound holding universally on every reference learner.
3. Not enough comparison to existing methods. The proposed method is only compared with its own counterpart, which doesn't say much. I think it needs to be compared with several methods from synchronous decentralized online learning as well as asynchronous federated learning to verify all the claims.

---- Post-rebuttal review ----

Thanks for the detailed answers. Most of my concerns have been clarified and I will upgrade my score. I would still recommend the authors to acquire more benchmarks to improve the empirical study section.

**Time Spent Reviewing:**

4

---

> ### Author Response · Authors · 2021-08-12
> **Author Response (Part 1)**
>
> Thank you for your insightful review! Due to the space limit of each comment, we divide our response into two parts. In this part, we would like to address your concerns on experiments. Your other questions will be answered in the next part.
>
> **1. Questions on experiments**
>
> **Q1-1a.“why is there no classification error plot … only see average loss plots”**
>
> Although traditional batch learning often reports classification error plots, it is indeed a common practice for online learning to report average loss plots (see reference [5, 10, 11]). The main reason is that, the performance metric of online learning is regret, and when evaluating algorithms in experiments we care most about the average regret. Note that, for the $j$-th reference learner, the average loss is defined as $\sum_tf_t(x_{t,j})/T$, which is exactly the first term of the average regret $\sum_t f_t(x_{t,j})/T-\sum_t f_t(x^*)/T$; and hence, the difference between the average loss and the average regret is $-\sum_t f_t(x^*)/T$, which is irrelevant to the compared algorithms. Thus, reporting the average loss is equivalent to reporting the average regret.
>
> Actually, we have measured the classification error, and find that it is consistent with the average loss. Due to the space limit, we only report average loss plots. We will add classification error plots in the appendix in the revision.
>
> **Q1-1b.“the loss value does not seem to improve much”**
>
> The loss improvement issue should be discussed case by case. We begin with the case of Figure 1. In general, the major benefit of asynchronous algorithms over their synchronous counterparts is the speedup of running time, rather than the loss improvement (see reference [1, 2] for more detailed discussions). Thus, in Figure 1, the main objective is to verify that our algorithm runs significantly faster than its synchronous counterpart; and we do not expect a large improvement on the final loss. Note that, this point has already been approved by Reviewer S1nK.
>
> As for the case of Figure 2 and Figure 3, our algorithm already has a 1% improvement in the average loss compared to the two asynchronous baselines. Actually, we have examined the settings with larger delays, and find that the improvements become much more significant. Due to the space limit, we only report the setting with a delay level of 100, which already shows a clear advantage of our algorithm over other asynchronous baselines.  We will add more experimental results with larger improvements in the appendix in the revision.
>
> **Q1-2.“no error bar on the average plot. It would be interesting to see if different nodes have similar learning patterns”**
>
> Indeed, in distributed online learning, it is very common not to report the error bar, as the average loss plot is usually very stable in different runs (see reference [5, 10, 11] for more examples). We do find such stableness in our experiments. We can add figures that include error bars in the appendix in the revision.
>
> In addition, what you suggested on the learning patterns among different learners is strictly guaranteed in theory of decentralized optimization (see reference [9]). In fact, it is a common practice for distributed online learning to report the average loss only on a given reference learner (see reference [5, 11] for more examples). Actually, we have examined the average losses of different learners, and find that they are nearly identical.
>
> **Q1-3.“not enough comparison to existing methods … only compared with its own counterpart”**
>
> In fact, besides its own counterpart, we have also compared our algorithm with two other asynchronous baselines in Section 4.3.1 (Figure 2) and Section 4.3.2 (Figure 3). It seems that you have missed them, presumably because these comparisons are organized by verifying the design choices of our algorithm, rather than a summary of baselines as you might expect. Now we give such a summary in the following.
>
> **Baseline 1: Synchronous online gradient-push** [6]. It is compared in Figure 1 of Section 4.2. This comparison verifies the efficiency of asynchronization, which was well recognized in your review.
>
> **Baseline 2: Asynchronous stochastic gradient descent** [2]. This is a classic asynchronous decentralized algorithm from stochastic optimization. We tailor it to the pure online setting, and present the comparison results in Figure 2 of Section 4.3.1. This comparison justifies the use of push-sum.
>
> **Baseline 3: Asynchronous stochastic gradient-push** [1]. This is the state-of-the-art asynchronous algorithm from stochastic optimization. We also tailor it to the pure online setting, and present the comparison results in Figure 3 of Section 4.3.2. This comparison verifies the effectiveness of instantaneous model averaging, which is a special design of our algorithm.
>
> In summary, we have indeed conducted extensive experiments and compared various baselines, including both synchronous and asynchronous algorithms, which verifies the superiority of our algorithm. Note that, we do not compare algorithms from asynchronous federated learning. The above synchronous and asynchronous baselines have already verified the claims on the superiority of our algorithm. This has been approved by Reviewer S1nK. Last, in many asynchronous stochastic learning papers, the proposed algorithms were not compared with asynchronous federated learning algorithms either (see reference [1, 2]); of course, we believe our novel designs can be easily applied to the setting of federated learning.
>
> **Reference**
>
> [1] Asynchronous gradient-push.
>
> [2] Asynchronous decentralized parallel stochastic gradient descent.
>
> [3] Decentralized consensus optimization with asynchrony and delays.
>
> [4] Introduction to online convex optimization.
>
> [5] Projection-free distributed online learning in networks.
>
> [6] Distributed online convex optimization on time-varying directed graphs.
>
> [7] Online learning with adversarial delays.
>
> [8] Stochastic gradient push for distributed deep learning.
>
> [9] Dual averaging for distributed optimization: convergence analysis and network scaling.
>
> [10] Delay-tolerant algorithms for asynchronous distributed online learning.
>
> [11] Projection-free distributed online convex optimization with sublinear communication complexity.

---

> > ### Author Response · Authors · 2021-08-16
> > **Author Response (Part 2)**
> >
> > Thank you for your insightful review again! In this part, we will answer your other questions besides experiments.
> >
> > **2. Questions on clarity**:
> >
> > **Q2-1.“some diagrams would help to explain the details more efficiently”**
> >
> > Good suggestion! We will add some diagrams in the revision.
> >
> > **Q2-2. Minor questions on clarity**
> >
> > *(1)“Is line 5 in Protocol 1 supposed to be outside the *if/else* block? Else, how does learner $i_t$ gets picked to perform local update?”*
> >
> > Line 5 is indeed inside the *if/else* block. We understand your confusion, and now we give a detailed explanation in the following. As stated in the footnote on page 3, at each local update event $t$ ($\delta_t=1$), the learner $i_t$ that performs the local update is exactly the learner $i_{l_t}$ that made the prediction at some previous prediction event $l_t$. Note that, the prediction event $l_t$ and the local update event $t$ are associated with the same loss function, and they have a one-to-one correspondence $l_t=t-d_t^{proc}$. Thus, we only need to specify the selected learner at each prediction event $t$ ($\delta_t=0$); then this learner is naturally selected at the corresponding local update event after it finishes the processing time. We are sorry for the confusion, and we will give a more detailed explanation in the revision.
> >
> > *(2) “Does the order of broadcasted copies matter?”*
> >
> > Indeed, different orders of broadcasted copies imply different patterns of processing and message diffusion. Hence, to give a general formulation, we do characterize the order of broadcasted copies in our framework in line 139.
> >
> > However, whether different orders will eventually affect the algorithm depends on the specific implementation of model averaging. In your mentioned naïve averaging, the order does matter. However, in push-sum, each copy $x_i$ from a given learner $i$ has an accompanying coefficient $\gamma_i$ which only depends on the network structure, i.e., the number of direct neighbors of learner $i$. When we perform push-sum averaging, we directly sum all the $\gamma_i x_i$, and hence the result $\sum\gamma_i x_i$ is irrelevant to the order of copies. Specifically, in your mentioned case, the push-sum average is always $\gamma_a\cdot a+\gamma_b\cdot b+\gamma_c\cdot c$, and it is clearly irrelevant to the order of copies $a$, $b$, and $c$. Actually, such an order invariance property is a design consideration of push-sum, which makes the averaging more robust.
> >
> > *(3) “seems like a typo in line 203 with the update for $y$”*
> >
> > Here it is indeed $y_i\leftarrow y_i$ rather than $y_i\leftarrow\gamma_iy_i$. Line 203 explains the operation in weighted projected gradient descent. The push-sum weight $y_i$ does not change during this operation. When it comes to the push-sum operation immediately after weighted projected gradient descent, we will multiply $y_i$ with $\gamma_i$ via the multiplication operation, which results in the $\gamma_iy_i$ as you expected. This has been explained in line 204-206 on page 5.
> >
> > *(4) “What exactly does it mean by ‘adversarially chosen by the environment’?”*
> >
> > It means that, at each prediction event, the environment selects a learner, presents a data sample to the learner, and the selected learner is required to predict the label of the sample. Here the "adversarially" means the learner can be selected arbitrarily from all learners, which can also be selected randomly.
> >
> > **3. Questions on originality**
> >
> > **Q3-1. “related work section … has to include asynchronous centralized learning”**
> >
> > Good suggestion! Admittedly, asynchronous centralized learning can provide a simpler perspective to understand the basic concepts of asynchronization. However, decentralized learning and centralized learning are often regarded as two individual research directions. In fact, many asynchronous decentralized papers did not review asynchronous centralized learning either (see reference [1, 3] for more examples). We will add a review of asynchronous centralized learning in the appendix in the revision.
> >
> > **Q3-2a. “what are previous indexing systems designed to achieve and what is the implication of this innovation?”**
> >
> > In general, indexing systems are designed to achieve two goals: (i) to characterize the whole learning process, and (ii) to provide notations and definitions that are necessary for theoretical analysis. In stochastic optimization, the performance metric is the convergence rate. Hence, indexing systems in this field are based on update events and they do not care about when to predict (see reference [1, 2, 3]). In online learning, the performance metric is the regret, which is measured by the actual losses. To analyze the regret, we need to specify the exact models when the predictions are made. Hence, prediction events are indispensable for indexing systems in online learning (see reference [4, 7]).
> >
> > However, as for asynchronous decentralized online learning, there does not even exist a formal framework that can accurately describe the whole learning process. The lack of a formal framework makes it very hard to conduct a solid theoretical analysis. Our proposed event indexing system can accurately describe the whole learning process and enable a meaningful regret analysis. This is the implication of our innovation, which has been already well recognized by Reviewer S1nK.
> >
> > **Q3-2b. “why do we have to index prediction events if they do not make any weight update?”**
> >
> > We index prediction events for two reasons. First, it is a common practice for online learning to index prediction events in order to exactly measure the regret (see reference [4, 7]). For the $j$-th reference learner, the regret is defined as $\sum_t(f_t(x_{t,j})-f_t(x^*))$, and the loss $f_t$ is evaluated by learner $j$'s prediction (i.e., model parameter) $x_{t,j}$ at time point $t$. To measure the regret, we need to exactly specify the time point $t$, as the model parameter $x_{t',j}$ at another time point $t'$ will be different from $x_{t,j}$, which incurs a different loss value.
> >
> > Second, as explained in line 123-126 on page 3, specific to the asynchronous decentralized online learning, if we do not index prediction events, we cannot analyze the effect of model averaging on the regret accurately. If we only index the local update events, then for the $j$-th reference learner, whether its model averaging occurs before or after some prediction event may be indistinguishable. Since model averaging will change learner $j$'s model parameter, we cannot specify learner $j$'s model parameter at the prediction event, and hence we can not exactly measure the actual losses.
> >
> > **Q3-2c. "can't they be modeled under the message delay component?"**
> >
> > As explained in line 123-126 on page 3, in our framework, message delays must be formulated via both prediction and update events, hence predictions cannot be modeled under the message delay component.
> >
> > Note that, what you suggest can be achieved via another new indexing system that indexes update events and message averaging operations. In such "update & message indexing system", we increase the round counter $t$ by $1$ each time some learner completes a local update or executes a model averaging operation. We have actually tried this before, but we found that, such an indexing system is much more complex and tedious than our event indexing system. This is because, if we use this new indexing system, we need to use the computation delay and message delay components to infer the predictions, which will make the regret definition very complex and hard to analyze. Hence, we use the indices of prediction and local update events, which is much more natural.
> >
> > **Q3-3.“instantaneous model averaging component makes the analysis more complex … this feels like a consequence of incorporating these designs, rather than a contribution”**
> >
> > Yes, you are right. This is a consequence of incorporating the instantaneous model averaging component. In line 218-225 of Section 3.1, we argue that this component is very important in online learning, but it may not be that important in stochastic optimization. We have verified its effectiveness empirically in Figure 3. Its analysis is not a contribution, but the algorithm design is. We will polish the presentation in the revision.
> >
> > **4. Question on theory**
> >
> > **Q4-1. “the learning rate … seems excessively large with massive number of nodes”**
> >
> > The learning rate will not be excessively large with massive number of nodes. Note that, it seems that you have missed the coefficient $C$ in the optimal learning rate $\eta^*=(mF/2G)(T+BCT)^{-1/2}$, which is commonly seen in the push-sum analysis (see reference [1, 8]). Obviously, $\eta^*$ is negatively correlated with $C$. As stated in Corollary 1 and the following remark, $C$ is positively correlated with $m$. When $m$ becomes large, the effect of $C$ counteracts the effect of $m$, hence $\eta^*$ will not be too large.
> >
> > **Reference**
> >
> > [1] Asynchronous gradient-push.
> >
> > [2] Asynchronous decentralized parallel stochastic gradient descent.
> >
> > [3] Decentralized consensus optimization with asynchrony and delays.
> >
> > [4] Introduction to online convex optimization.
> >
> > [5] Projection-free distributed online learning in networks.
> >
> > [6] Distributed online convex optimization on time-varying directed graphs.
> >
> > [7] Online learning with adversarial delays.
> >
> > [8] Stochastic gradient push for distributed deep learning.
> >
> > [9] Dual averaging for distributed optimization: convergence analysis and network scaling.
> >
> > [10] Delay-tolerant algorithms for asynchronous distributed online learning.
> >
> > [11] Projection-free distributed online convex optimization with sublinear communication complexity.

---

### Official Review · Reviewer_ec5A · 2021-07-16

**Rating:** 7
**Confidence:** 3

**Summary:**

In this paper, authors propose asynchronous algorithm for convex optimization in decentralized distributed model. In contrast with other algorithms, the averaging procedure is asymmetric that leads to faster convergence comparing to the synchronous updates.

**Limitations And Societal Impact:**

yes

**Main Review:**

In this paper, authors consider distributed decentralized model with asynchronous communication between nodes. However, in this setup there is no one to one correspondence between nodes and loss functions.
First, authors explain their framework that is following: 1) some learner(node) is selected to predict with its own predictions. 2) some loss function is sent to the learner to calculate the gradient. 3) learner shares its update with some of the neighbors. 4) then all learners, who are not busy (see calculating the gradient) are performing the model averaging.

This setup allows to solve the common problem of synchronous algorithms, when some machine can be slow and all the others should wait for it to finish to perform their updates.

Then authors present the weighted projected online gradient descent that is based on the push-sum communications: instead of using doubly stochastic gossip matrix for averaging only the column stochasticity is required.

Then authors describe the main contribution of the work; however it is hard to follow since there is no algorithm in the paper. Furthermore, I was lost in the theoretical result.

In the experimental part, authors provide the comparison of the symmetric and push-sum algorithm that proves the interest of push-sum algorithm for big delays that makes the asymmetric gossiping more robust. However, I think that this comparison is not good enough and some other algorithms should be added. Furthermore, some additional figures were provided to show the dependence of the efficiency of the push-sum algorithm of network size, network topology, and delays. Actually, I missed the message of these plots since I find these comparisons quite useless. Moreover, these plots are not colorblind-friendly so it was impossible for me to read them.

**Time Spent Reviewing:**

1-2 hour

---

> ### Author Response · Authors · 2021-08-11
> **Author Response**
>
> Thank you for your detailed review! Respectfully, we find that you may have some possible misunderstandings on our paper. We will make some clarifications and address other concerns in the following.
>
> **1. Questions on the algorithm and theoretical results**
>
> **Q1-1.“the main contribution … is hard to follow since there is no algorithm in the paper”**
>
> We *do* have an algorithm in the paper. It is termed as Asynchronous Decentralized Online Gradient-Push (AD-OGP), which is mainly described in Section 3.1. The algorithm consists of two core modules, and the key operations of the two modules are illustrated in line 188 and line 206 respectively. We do not explicitly give a whole diagram of AD-OGP, since it is highly overlapped with the protocol of our framework given in Protocol 1, and the space is really limited. We suppose this might be the reason for which you have missed it.
>
> In fact, only presenting the key operations of an algorithm is a typical treatment in many online learning literature when there is already a protocol of the algorithmic framework (see reference [1, 2]). Actually, we have provided a complete diagram of AD-OGP on page 2 of the appendix. Note that, our algorithm is well justified by the other two reviewers.
>
> Additionally, besides the algorithm and theoretical analysis, we also make a contribution in formulating the first formal framework of asynchronous decentralized online learning, which fully characterizes the whole learning process. The contribution of this framework is well approved of by the other two reviewers.
>
> **Q1-2.“Furthermore, I was lost in theoretical results”**
>
> In a nutshell, our theory gives the regret bound for our proposed AD-OGP algorithm. Such theoretical results are very typical in online learning literature (see reference [1, 2, 6, 7]). The regret bound is a standard performance metric for online algorithms (see reference [3]); it can be translated into the convergence rate through an “online-to-batch” conversion. In Section 3.2, we derive a regret bound in the order of $O(T^{1/2})$, which implies that, as $T$ goes very large, on average, every learner will perform as well as the best fixed model in hindsight [3]. Our bound also illustrates how various factors, such as network size, network topology, and delays, will affect the performance of the algorithm. Indeed, our theoretical results are well recognized by all the other two reviewers.
>
> **2. Questions on experiments**
>
> **Q2-1.“the experimental part … the comparison is not good enough and some other algorithms should be added”**
>
> It seems that you have ignored many of our experiments. In fact, we have compared our algorithm with *three baseline algorithms*. You seem to only notice the comparison in Section 4.3.1 (Figure 2) and miss the other two comparisons in Section 4.2 (Figure 1) and Section 4.3.2 (Figure 3). This is presumably because in our paper each comparison is presented in one separate subsection. We now give a summary of the compared algorithms, as you might expect.
>
>    **Baseline 1: Synchronous online gradient push** [2]. This is the synchronous counterpart of AD-OGP. It is compared in Figure 1 of Section 4.1, which verifies the efficiency of AD-OGP compared to its synchronous counterpart.
>
>    **Baseline 2: Asynchronous stochastic gradient descent** [4]. This is a classic asynchronous decentralized algorithm from stochastic optimization. We tailor it to the pure online setting, and present the comparison results in Figure 2 of Section 4.3.1. This comparison justifies the use of push-sum, and was well recognized in your review.
>
>    **Baseline 3: Asynchronous stochastic gradient-push** [5]. This is the state-of-the-art asynchronous algorithm from stochastic optimization. We also tailor it to the pure online setting, and present the comparison results in Figure 3 of Section 4.3.2 (Figure 3 is on the right of Figure 2). It verifies the effectiveness of instantaneous model averaging, which is a novel design of our algorithm.
>
> In summary, we have conducted extensive experiments and compared many baselines, including both synchronous algorithms and asynchronous algorithms, which well verifies the superiority of our algorithm.
>
> **Q2-2.“additional figures show the dependence of the efficiency of the push-sum algorithm of network size, network topology, and delays. I missed the message of these plots since I find these comparisons quite useless”**
>
> In Figure 4, we examine the effect of network size, topology, and delays, on the algorithm’s performance. These results illustrate the perfect alignment between the empirical behaviors and our theoretical results, which proves the correctness of our regret bound. This kind of experiment is very typical in online learning literature, which is designed to verify the theoretical results (see reference [6, 7]).
>
> **Q2-3. “these plots are not colorblind-friendly"**
>
> We are sorry for the confusion. In Figure 4, the plots show that, AD-OGP suffers higher loss with larger network size, less connected network, heavier processing delays, and heavier message delays. These observations perfectly align with our theoretical regret bound. We will change the colors and use different line types to distinguish different curves in the revision to help you better understand these plots.
>
> **Reference**
>
> [1] Online learning with adversarial delays.
>
> [2] Distributed online convex optimization on time-varying directed graphs.
>
> [3] Introduction to online convex optimization.
>
> [4] Asynchronous decentralized parallel stochastic gradient descent.
>
> [5] Asynchronous gradient push.
>
> [6] Projection-free distributed online learning in networks.
>
> [7] Delay-tolerant algorithms for asynchronous distributed online learning.

---

> > ### Comment · Reviewer_ec5A · 2021-09-02
> > **Reply to Authors**
> >
> > Thank you for your comments,
> >
> > According to the first part, that is exactly what I mean, you presented the AD-OCO algorithm and then only described the AD-OGP, and it will be really helpful to se the same ``algoritmic'' for this algorithm to make it easier comparison a-la line-by-line.
> >
> > According to the theoretical result, I cannot see the $O(T^{1/2})$ somewhere. In the corollary I see the linear dependence on $T$.  In the Theorem all the dependences on $T$ are hidden, that what I meant by ``lost in the theoretical result''.
> >
> > You have the sublinear dependence in the remark just under the corollary and I agree that it can be the case if Assumption 1 holds, but it is moved out of the scope of the Theorem.
> >
> > Thank you for explanations of the numerical part, now it became more clear for me. I agree, that the provided algorithm performs better, thank you.

---

### Official Review · Reviewer_S1nK · 2021-07-18

**Rating:** 7
**Confidence:** 4

**Summary:**

This work considers decentralized online learning in the asynchronous setup. Precisely, the learners do not make predictions at the same time and both the local updates and message transmissions incur delays. To account for this situation, the authors propose to index the time by two types of events: predictions and updates. Under this framework, an asynchronous push-sum algorithm is studied and its regret bound is established. The experiments do not only demonstrate the benefit of asynchronous method but also investigate several design choices of the algorithm.

**Limitations And Societal Impact:**

The limitations are discussed in the conclusion. Another limitation that I find about this work is that the learners cannot make predictions when they are occupied by the computation of the gradients. This is slightly different from the usual delayed online learning setup [Quanrud and Khashabi 2015](https://papers.nips.cc/paper/2015/file/72da7fd6d1302c0a159f6436d01e9eb0-Paper.pdf).

**Main Review:**

This work makes an important step towards a better understanding of the asynchronous online optimization problem. Indeed, I agree with the authors that a complete description of an asynchronous online learning framework is inherently more intriguing than its offline couterpart, and the methodology considered in the paper paves the way for future research that tackle similar problems.

The algorithm itself is very similar to the original push-sum method, but the analysis in the new framework also turns out to be challenging. Moreover, I also find the papers to be well-written and easy to follow, despite the complexity of the underlying setup.

One potential concern is that the online learning part seems to be missing from the experimental section. In fact, it is not specified how the loss sequence is generated and how and when a learner is selected for making a prediction. Without the latter information, I do not really know what time measures in Figure 1. Moreover, how can we ensure that this difference is caused by the implementation of the algorithms? As for the losses, I imagine that samples are sampled uniformly at random. Indeed, stochastic optimization is frequently viewed as a special case of online learning (notably through online-to-batch conversion), but these experiments do not justify the choices to focus on the online framework. Similarly, the authors plot the average loss and not the (average) regret for the experiments.

Other remarks:

1. Pointing the readers to appendix B at the bottom at page 5 can be helpful, since, as claimed by the authors, the feasibility of the methods is not that immediate. Similarly, pointing the readers to the appendix for the complete pseudo-code can also be helpful.

2. With Assumption 1, I believe it always holds $D^{proc}\le\Gamma_{d}T$.

3. $\boldsymbol{\hat{g_t}}$
depends on the choice of $j$ and I recommend explicating this in the notation (such as $\hat{\boldsymbol{g}}_{j,t}$).

4. Adding dashed lines in Figure 1 can help illustrate the fact that the two algorithms achieve the same average loss.

5. [Hsieh et al. 2020](https://arxiv.org/abs/2012.11579) also studies asynchronous decentralized online optimization, though through a totally different approach. I suggest the authors include it in related work.

**Time Spent Reviewing:**

5

---

> ### Author Response · Authors · 2021-08-12
> **Author Response**
>
> Thank you for your excellent review!
>
> **1. Questions on experiments:**
>
> **Q1-1.“how the loss sequence is generated”**
>
> The loss sequence is generated in a strict online manner. Specifically, all samples in the dataset form a certain order. Then according to this order, the samples are fed to the learner one after another at each prediction event. Thus, the samples are clearly not sampled at random as in stochastic optimization; it is adequate to justify the choices to focus on the pure online framework. Such treatment is typical in the online learning literature (see reference [1,2]). We're sorry for not explicitly explaining it, and we will add the details in the revision.
>
> **Q1-2. “how and when a learner is selected for making a prediction**
>
> For each learner, we simulate its learning process strictly following the individual view of our framework. Specifically, as stated in line 284-289 and line 332-333, for each learner, we randomly sample three types of time intervals from some certain distributions: (i) the processing time for the learner to calculate the gradient and perform the local update, (ii) the transmission time for the message to reach its gossiping receiver, and (iii) the waiting time for the learner to wait for the next prediction after it finishes each local update.
>
> The above three types of time intervals fully characterize the learning process of a learner, which determines how and when it is selected to predict. Specifically, each learner alternates between processing and waiting; it is selected to predict each time after it finishes the waiting time. Note that, the transmission time does not affect when to predict, because in our asynchronous algorithm the learner can proceed on while the message is still being transmitted. We will add a detailed explanation of the "how and when" issue in the revision.
>
> **Q1-3.“what time measures in Figure 1? how can we ensure that this difference is caused by the implementation of the algorithms”**
>
> In Figure 1 the time metric exactly measures the actual running time of the algorithms. Specifically, the running time of both synchronous and asynchronous algorithms contains the processing time of the samples and the waiting time for the learner to make the next prediction after it finishes each local update. Additionally, the running time of the synchronous algorithm contains the time for the fast learner to wait for the slow learner to finish the processing and message transmission.
>
> Note that, unlike stochastic optimization, in the online setting, the synchronous algorithm also has the waiting time, i.e., the learners need to wait some time to make the next round of predictions after they finish the previous round (i.e., wait for the next samples to come). In our experiment in Figure 1, we set the waiting time of both algorithms to be relatively small compared to the processing time and the transmission time. Now the waiting time has almost negligible effects on the running time difference. In summary, our experimental setup ensures that the time difference indeed illustrates the rationale of the synchronous and asynchronous algorithms. We will add the details in the revision.
>
> **Q1-4.“plot the average loss and not the average regret for the experiments"**
>
> In fact, plotting the average loss is also a very common practice in the experimental parts of online learning research (see reference [1, 2, 3]). Let's elaborate on this. For the $j$-th reference learner, the average loss is defined as $\sum_t f_t(x_{t,j})/T$, which is exactly the first term of the average regret $\sum_t f_t(x_{t,j})/T-\sum_t f_t(x^*)/T$. Then the difference between the average loss and the average regret is $-\sum_t f_t(x^*)/T$, which is irrelevant to the compared algorithms. Hence, plotting the average loss can be regarded as equivalent to plotting the average regret. We can easily translate the average loss plot into the average regret plot in the revision.
>
> **2. Minor remarks**
>
> (1) We will add a remark to refer them to the appendix.
>
> (2) Good observation! It indeed satisfies $D^{proc}\leq\Gamma_dT$. Note that, Assumption 1 is commonly used in the analysis of asynchronous decentralized algorithms (see reference [4, 6]).
>
> (3) We use $\hat g_t$ because the reference learner $j$ is fixed throughout the analysis, and $\hat g_t$ implicitly means $\hat g_{j,t}$. We will explicate this point in the revision.
>
> (4) We will add dash lines in Figure 1.
>
> (5) We have read this paper, and just as you say, their approach is indeed totally different. We will add it to the related work.
>
> **Reference**
>
> [1] Delay-tolerant algorithms for asynchronous distributed online learning.
>
> [2] Projection-free distributed online learning in networks.
>
> [3] Projection-free distributed online convex optimization with sublinear communication complexity.
>
> [4] Asynchronous gradient push.
>
> [5] Decentralized consensus optimization with asynchrony and delays.
>
> [6] Robust asynchronous stochastic gradient-push.
>
> [7] Distributed online convex optimization on time-varying directed graphs.

---

### Author Response · Authors · 2021-08-12
**General Author Response**

We would like to thank all reviewers for the constructive and helpful reviews. We hope our response addresses the concerns. We are very happy to address any additional questions that may arise during the discussion period!

---

### Author Response · Authors · 2021-08-26
**Eagerly look forward to knowing your update**

Respected Reviewers,

The rolling discussion phase is drawing to a close. We eagerly look forward to knowing your update after the initial author response.

We are wondering whether your concerns have been well addressed. If you have any additional questions, it would be great if you could let us know. We are readily prepared to address them.

---

### Decision · Program_Chairs · 2021-09-27

**Decision:**

Accept (Poster)

**Comment:**

The reviewers agree that this generally a good paper although not entirely without (minor) flaws. Please take the reviewers comments in consideration when preparing a revision. The answers provided by the authors were given due consideration.